# ExLLM: Experience-Enhanced LLM Optimization for Molecular Design and Beyond

## Abstract

Molecular design involves an enormous and irregular search space, where traditional optimizers such as Bayesian optimization, genetic algorithms, and generative models struggle to leverage expert knowledge or handle complex feedback. Recently, LLMs have been used as optimizers, achieving promising results on benchmarks such as PMO. However, existing approaches rely only on prompting or extra training, without mechanisms to handle complex feedback or maintain scalable memory. In particular, the common practice of appending or summarizing experiences at every query leads to redundancy, degraded exploration, and ultimately poor final outcomes under large-scale iterative search. We introduce ExLLM (Experience-Enhanced LLM optimization), an LLM-as-optimizer framework with three components: (1) a compact, evolving experience snippet tailored to large discrete spaces that distills non-redundant cues and improves convergence at low cost; (2) a simple yet effective k-offspring scheme that widens exploration per call and reduces orchestration cost; and (3) a lightweight feedback adapter that normalizes objectives for selection while formatting constraints and expert hints for iteration. ExLLM sets new state-of-the-art results on PMO and generalizes strongly—in our setup, it sets records on circle packing and stellarator design, and yields consistent gains across additional domains—requiring only a task-description template and evaluation functions to transfer.

## 1 Introduction

Molecular design underpins drug discovery and materials science, yet the search space is vast and highly discrete, making efficient optimization difficult. Classical machine learning approaches: Bayesian optimization (BO), genetic algorithms (GA), reinforcement learning (RL), multi-objective optimization (MOO), and MCMC, treat the problem largely as black-box search (Shin et al., 2024; Kim et al., 2024; Jensen, 2019; Nigam et al., 2019; Liu et al., 2025; Verhellen, 2022; Xie et al., 2021; Sun et al., 2022; Olivecrona et al., 2017; Jin et al., 2020). Deep generative models improve proposal quality by learning molecular distributions (e.g., JTVAE, VJTNN, DST, diffusion and transformer-based models such as MOOD, MolGPT, MOLGEN) and enable latent-space search (Jin et al., 2018a;b; Fu et al., 2021; Lee et al., 2023; Bagal et al., 2021; Fang et al., 2024; Abeer et al., 2024). However, these lines typically rely on scalarized rewards and fixed pipelines, making it hard to incorporate rich priors (chemist heuristics, textual rules) and to handle heterogeneous feedback (multiple objectives, hard/soft constraints) without task-specific re-engineering; practical protocols such as PMO further highlight the need to optimize under a fixed evaluation budget due to costly oracle calls (Gao et al., 2022).

Large language models (LLMs) offer a complementary opportunity: they encode broad domain knowledge, support reasoning, and can be steered with prompts (Vaswani, 2017; AI4Science & Quantum, 2023; Brown, 2020). Recent work explores LLMs as optimizers or operators within evolutionary loops (e.g., OPRO, LMEA, AlphaEvolve; reasoning–acting with ReAct) and reports encouraging results on numerical, coding, and planning tasks (Yang et al., 2024; Liu et al., 2024b; Novikov et al., 2025a; Yao et al., 2022; Wu et al., 2024a). In molecular design, systems such as ChemCrow, LICO, MolReGPT, Prompt-MolOpt, and MOLLEO demonstrate that pre-trained LLMs or LLM–GA hybrids can guide candidate generation and multi-parameter search (M. Bran et al., 2024; Nguyen & Grover, 2024; Li et al., 2024; Wu et al., 2024b; Wang et al., 2024a). Yet these efforts remain early-stage: most are heavily prompt-dependent or require additional parameter training,

and lack a memory mechanism tailored to molecular optimization which has large, discrete search loops. Existing memory systems were developed primarily for QA, coding, or short-horizon decision making; they append per-step summaries and retrieve them at inference (RAG, RETRO, MemoryBank, MemLLM, A-Mem, Memory-R1), which—when naively reused over long optimization runs—inflate prompts, accumulate redundancy, and bias the search (Lewis et al., 2020; Borgeaud et al., 2022; Zhong et al., 2024; Modarressi et al., 2024; Xu et al., 2025; Yan et al., 2025). Moreover, unified handling of heterogeneous feedback (multi-objective signals, constraints, and expert textual hints) remains limited in practice.

We propose **ExLLM**, an LLM-as-optimizer framework designed for molecular optimization. ExLLM treats the LLM as the optimizer itself and introduces three complementary mechanisms: a *single, evolving* experience distilled from good and bad cases to avoid memory bloat; a *k-offspring* sampling scheme that leverages the autoregressive factorization to widen exploration per query; and a unified *feedback adapter* that integrates objectives, constraints and textual feedback for iterative prompting. The framework is simple to transfer: it works with only a task-description template and custom evaluation functions, and requires no training, as summarized in Figure 1.

**Contributions.** (1) We introduce an **evolving experience** mechanism tailored to large discrete spaces: a compact, low-redundancy snippet updated each generation, which improves convergence and results while controlling cost and avoiding exploration collapse, contrasting with retrieval-style memories (Zhao et al., 2023; Lewis et al., 2020; Borgeaud et al., 2022). (2) We propose a $k$-**offspring** strategy that increases exploratory breadth per LLM call and yields consistent gains under a fixed budget, with an empirical trade-off curve for $k$. (3) We develop a **feedback adapter** that unifies multi-objective signals for selection and incorporates constraints/expert text as concise prompts; promoting critical, variable constraints to explicit objectives improves stability without additional training. (4) We demonstrate strong results on molecular optimization: ExLLM achieves a PMO aggregate score of **19.165** (max 23), ranking **first on 17/23** tasks and improving over the previous SOTA (17.862) by **+7.3%** (Gao et al., 2022; Wang et al., 2024a). We package these contributions into a general optimizer for large discrete spaces: with only a task-description template and evaluation functions, it sets new records in circle packing and stellarator design, and delivers consistent gains across additional domains (e.g., MOTSP/MOCVRP, SACS, NK2R peptide, GCU operator design).

## 2 RELATED WORK

### 2.1 MOLECULAR DESIGN WITH MACHINE LEARNING

Molecular optimization has long been approached as a black-box search problem using Bayesian optimization (BO), genetic algorithms (GA), reinforcement learning (RL), multi-objective optimization (MOO), and MCMC variants. Representative **GA/BO** lines include GB-GA and its extensions—e.g., diversity-aware discriminators and Tanimoto-kernel Gaussian processes—which remain competitive on classic property maximization tasks (Jensen, 2019; Nigam et al., 2019; Tripp et al., 2021; Gómez-Bombarelli et al., 2018). BO-style **MOO** with learned encoders (MLPS) (Liu et al., 2025) and graph-based MOO (Verhellen, 2022) likewise seek Pareto-optimal solutions in latent or combinatorial spaces. Sampling-based methods such as MARS (**MCMC**) and Monte-Carlo tree search further explore chemical spaces probabilistically to identify molecules with desired properties (Xie et al., 2021; Sun et al., 2022). **RL** pipelines train generative policies from reward signals (e.g., REINVENT and rationale-guided GNNs) (Olivecrona et al., 2017; Jin et al., 2020), with recent variants such as DyMol, Augmented Memory and Genetic-GFN improving multi-objective handling, utilize data augmentation and experience replaying, or blending evolutionary operators with flow-based generators (Shin et al., 2024; Guo & Schwaller, 2024; Kim et al., 2024). While these families achieve strong results on several PMO tasks (Gao et al., 2022), they typically optimize only via function evaluations, making it difficult to incorporate domain priors (chemist heuristics, expert constraints) beyond hand-crafted rewards, and requiring nontrivial re-engineering or additional training when objectives or constraints change.

**Deep generative models** advanced the field by learning molecular distributions and enabling higher-quality proposals. Autoencoding and structured-latent approaches (e.g., JTVAE,VJTNN, DST) capture scaffold and substructure regularities (Jin et al., 2018a;b; Fu et al., 2021), and diffusion and transformer-based models (MOOD, MolGPT, MOLGEN) further improve sample fidelity and cross-domain applicability (Lee et al., 2023; Bagal et al., 2021; Fang et al., 2024). Latent-space

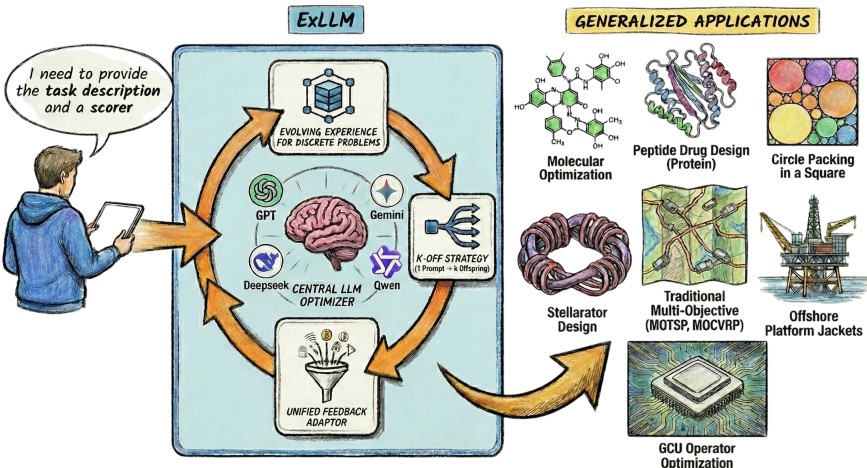

Figure 1: ExLLM requires only two user inputs: a task-description template and evaluation functions, and operates without domain-specific training or tuning. The figure shows how a single unified workflow transfers seamlessly across different discrete problem optimization classes (chemical, geometric, combinatorial, physical, and code-generation tasks), achieving state-of-the-art or record-level performance under domain shift.

optimization (LSO) shows that multi-objective search in the learned space can be effective for deep generators (Abeer et al., 2024). However, limitations exist: (1) Models largely learn from data and scalar rewards; codifying rich textual heuristics, design rules, or exception-heavy lab wisdom into training signals is cumbersome and brittle. (2) Combining multiple objectives, hard/soft constraints, and heuristic rules in a single training loop often requires bespoke reward shaping, delicate weighting, or separate pipelines. Many systems require task-specific fine-tuning or reward engineering when the optimization goal changes, limiting plug-and-play transfer to new objectives or domains (Gao et al., 2022; Liu et al., 2025; Verhellen, 2022).

## 2.2 MOLECULAR DESIGN WITH LLM

LLMs with domain knowledge are increasingly explored for drug and materials discovery (AI4Science & Quantum, 2023). Agentic tool-use systems such as ChemCrow show that LLMs can plan, call external chemistry tools, and iteratively refine candidates (M. Bran et al., 2024). In LICO (Nguyen & Grover, 2024), in-context learning is strengthened by pretraining with separated embedding and prediction layers to improve molecule generation; later Moayedpour et al. (2024) extends this idea to multi-objective settings. MolReGPT targets few-shot molecular optimization with additional parameterization for rapid adaptation (Li et al., 2024). MOLLEO integrates LLMs with a genetic algorithm to guide mutations/edits during search, illustrating a viable LLM-as-optimizer pattern (Wang et al., 2024a), but it lacks an explicit experience-reuse mechanism, explores this framework at a relatively early stage, and is mainly scoped to molecular-design benchmarks. Prompt-MolOpt explicitly leverages the domain knowledge of LLMs via property-specific prompt embeddings and a sequence-to-sequence Transformer trained on substructure-annotated pairs, demonstrating strong zero-/few-shot behavior on property-driven tasks (Wu et al., 2024b). However, existing LLM-based molecular design approaches are still highly prompt-dependent or need additional training, and lack an experience mechanism tailored to vast exploration spaces that can distill and reuse knowledge during optimization. The ability to handle complex feedback is still limited.

## 2.3 LLM-AS-OPTIMIZER AND MEMORY MECHANISM

Recent work tends to use LLM as optimizer to utilize domain knowledge and reasoning ability to guide efficient high-dimensional exploration without training (AI4Science & Quantum, 2023; Wu et al., 2024a; Brown, 2020). OPRO and LMEA cast LLMs as crossover/mutation operators within GA-style loops, balancing exploitation–exploration via prompt design and temperature control (Yang et al., 2024; Liu et al., 2024b); AlphaEvolve further systematizes this LLM-in-the-loop evolutionary pattern,

and ReAct exemplifies reasoning–acting procedures that can guide decision making (Novikov et al., 2025a; Yao et al., 2022). In molecular settings, constrained prompt engineering shows encouraging alignment effects (Wang et al., 2024b), while several studies report that GA+LLM pipelines can be efficient and competitive against standalone LLMs and traditional MOO when coupled with well-structured workflows (Liu et al., 2023; 2024a;c; Huang et al., 2024; Brahmachary et al., 2024), indicating LLM as a promising optimizer for numerical optimizations, coding and planning problems.

External memory is widely used to inject up-to-date knowledge and reduce parametric forgetting—ranging from RAG (retrieve-then-concatenate) (Lewis et al., 2020) and RETRO (cross-attending to retrieved neighbors) (Borgeaud et al., 2022) to persistent, editable stores such as MemoryBank (human-like forgetting/refresh) and MemLLM (explicit read/write) (Zhong et al., 2024; Modarressi et al., 2024). Hybrid controllers like A-Mem and RL-trained Memory-R1 further learn when to add/update/delete/consume memories (Xu et al., 2025; Yan et al., 2025). While effective for QA, code and planning tasks, these designs are not tailored to large discrete optimization: massive iteration counts and huge candidate spaces make append-only or dense-retrieval memories prone to memory bloat and redundancy, drive up per-query prompt cost, and bias exploration when similar histories are repeatedly injected, as shown in table 1. We address this by a compact, low-redundancy experience pool that is both efficient and effective with much lower costs.

## 3 METHODOLOGY

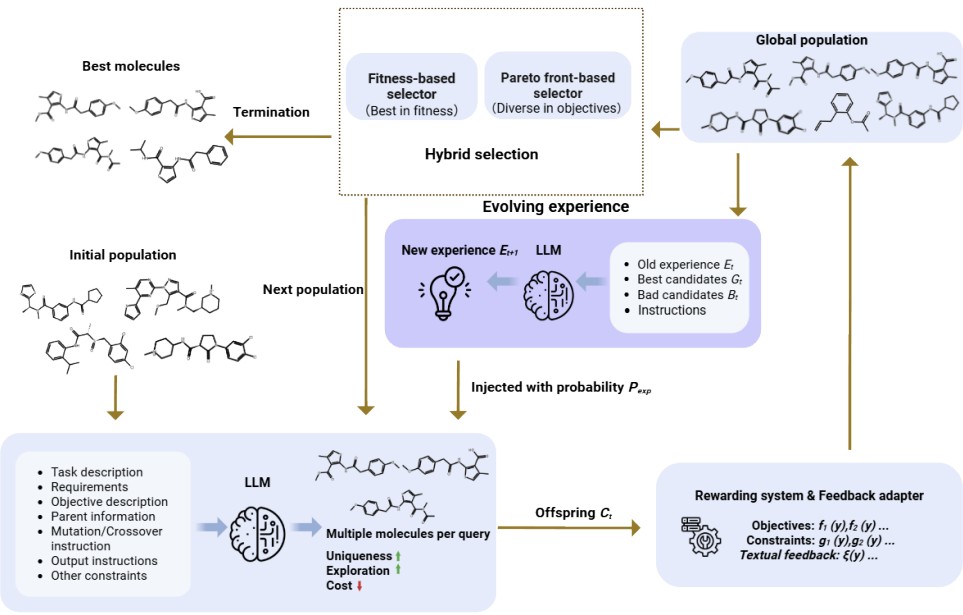

Figure 2: Overall framework of ExLLM. The process begins with an initialized population, followed by LLM-based $k$-offspring generation, evaluation and feedback aggregation, hybrid selection (fitness + Pareto), and experience update. These steps repeat until the evaluation budget is exhausted.

We first provide an overview of our framework, also shown in Figure 2, and then give detailed elaborations of the key components. We cast the LLM as an evolutionary optimizer under a fixed evaluation budget $B$ following PMO benchmark settings (Gao et al., 2022). Let $P_t$ denote the population at generation $t$:

**Initialization.** Construct the initial population $P_0$ (e.g., random seeds, scaffold templates, or domain priors).

**LLM-as-optimizer with $k$-offspring.** For every randomly selected pair of parents $(x_i, x_j) \subseteq P_t$, a proprietary commercial LLM (e.g., GPT, Gemini) proposes $k$ candidate offspring:

$$C_t(x_i, x_j) = \{y^{(1)}, \ldots, y^{(k)}\}, \qquad y^{(i)} \sim p_\theta\big( \cdot \mid (x_i, x_j), \text{ task template, } E_t \big),$$

where $p_\theta$ is the autoregressive distribution and $E_t$ is the distilled experience from the previous generation (Section 3.1). The complete candidate set at iteration $t$ is then

$$C_t = \bigcup_{(x_i, x_j) \in \mathcal{M}(P_t)} C_t(x_i, x_j),$$

where $\mathcal{M}(P_t)$ denotes the set of parent pairs selected from the current population $P_t$. The prompt is constructed using a designed template that includes the task description, requirements, objective specifications, parent information, mutation/crossover instructions, output guidelines, and other constraints.

**Feedback aggregation.** Each $y \in C_t$ is evaluated to obtain: (i) a vector of objective values $\mathbf{f}(y) = [f_1(y), \ldots, f_M(y)]$; the raw objective values are preserved so that they can be explicitly included in the prompt, which facilitates the LLM's understanding of task semantics. All objectives form a unified vector representation that is subsequently employed for Pareto-based selection. (ii) constraint values $\mathbf{g}(y) = [g_1(y), \ldots, g_J(y)]$; (iii) optional expert/textual feedback $\xi(y)$. The feedback adapter (Section 3.3) normalizes objectives into a comparable vector used for selection, and formats $\mathbf{g}(y)$ and $\xi(y)$ into compact, structured text that can be injected back to the LLM in the next generation.

**Selection (Fitness + Pareto).** To construct the next generation, we employ a hybrid strategy: half of the candidates are selected by ranking according to the scalar fitness value

$$F(y) = \sum_{i=1}^{M} w_i \, \widehat{f}_i(y), \qquad \sum_i w_i = 1, \tag{1}$$

where weights are equal in our experiments, the remaining half are drawn from the Pareto front based on dominance relations over the normalized objective vectors. This design balances exploitation of high-performing individuals with preservation of diversity across the multi-objective space. Because some molecules may achieve relatively high fitness values and exhibit structural diversity with good potential, but share similar objective distributions which make them dominated by many other points, fitness-based selection ensures these candidates are retained. Meanwhile, molecules with lower fitness may still excel on specific objectives and remain non-dominated; such molecules are preserved by Pareto-front selection for their potential.

**Experience update.** From all historically evaluated candidates up to generation $t$, we identify a set of *good* examples $G_t$ (top-$r$ by $F$) and sample a set of *bad* examples $B_t$ uniformly from the lower half of the fitness ranking. We then update the experience $E_{t+1}$ by combining $E_t$ with distilled insights from $G_t \cup B_t$ and discarding stale content; see Section 3.1. Then it jumps to LLM-as-optimizer with k-offspring and repeats until the total evaluation calls reach $B$ (PMO protocol) (Gao et al., 2022).

## 3.1 EVOLVING EXPERIENCE

| Memory Method | Hypervolume | Top10 AUC | Uniqueness | LLM queries | Cost (USD) | Running time (h) | Memory (MB) |
|---|---|---|---|---|---|---|---|
| Retrieval-style | $0.427 \pm 0.155$ | $3.904 \pm 0.427$ | $0.139 \pm 0.043$ | $18055 \pm 567$ | $> 100$ | $> 24$ | $\approx 350$ |
| No memory | $0.545 \pm 0.189$ | $3.974 \pm 0.061$ | $0.511 \pm 0.062$ | $3625 \pm 1036$ | $6.246 \pm 0.731$ | $0.262 \pm 0.069$ | $0.000 \pm 0.000$ |
| Our design | $0.750 \pm 0.007$ | $4.070 \pm 0.026$ | $0.615 \pm 0.035$ | $3312 \pm 725$ | $6.938 \pm 0.796$ | $0.393 \pm 0.114$ | $0.002 \pm 0.000$ |

Table 1: Experiments with different memory mechanisms on a 5-objective molecular optimization task, identical to the task in Table 2, using Gemini-2.5-flash without thinking mode.

**Why not a retrieval-style memory.** As discussed in Section 2.3, traditional memories maintain *per-query* summaries and then *re-inject* many of them at every LLM call. We implemented this variant as a control: after each generation, we summarize the new LLM outputs, rank the pool, and insert the top-$K$ ($K \leq 100$) entries into every subsequent prompt (Table 1). In large discrete search, this leads to (i) **memory bloat** and higher prompt cost, (ii) **exploration collapse** with repeated proposals. Some runs of retrieval style memory were terminated early after repeatedly failing to produce novel valid candidates. Accordingly, the reported cost and runtime are lower bounds measured at termination, and memory is the approximate peak usage. We also observed a marked drop in uniqueness (sometimes

even below 10%) and hypervolume, large LLM query costs, significantly longer running time and much larger storage space used compared to no memory and our design.

Prior work such as ReEvo or ExpeL uses summary-style memories for short-horizon reasoning. However, these designs are not suitable for large discrete optimization, where redundancy accumulation and exploration collapse are major problems. Here, we keep one continually updated experience $E_t$ into few hundreds of words. It captures actionable, transferable cues (e.g., frequently binding constraints, robust edit patterns, common invalid cases) without retrieving multiple entries. This avoids prompt bloat and reduces the risk of over-constraining exploration. From all historically evaluated candidates up to generation $t$, form an evidence set $D_t = G_t \cup B_t$, where $G_t$ are the top-$r$ candidates by fitness $F$ and $B_t$ are uniformly sampled from the lower-half of the fitness ranking. Let $S_\theta(\cdot)$ denote the *same LLM* used by the optimizer, prompted to produce a single concise memo (a few hundred words) that merges new evidence with prior experience. We update: $E_{t+1} = S_\theta(E_t, D_t)$ Operationally, $S_\theta$ folds in "good" and "bad" cases to reduce bias, promotes non-redundant micro-insights, and overwrites outdated content so that $E_t$ remains compact, general and up-to-date. At generation $t{+}1$, $E_{t+1}$ is inserted directly into the LLM context without retrievals. To prevent over-exploitation, the experience is included in the prompt with probability $p_{\exp} \in [0,1]$, sampled independently per call:

$$\mathbb{I}\{\text{inject } E_t\} \sim \text{Bernoulli}(p_{\exp}).$$

This mechanism keeps prompts short on average, amortizes summarization cost (one update per generation), and maintains exploration headroom. We ablate $p_{\exp}$ in Section 6.3 and observe that intermediate values yield the best trade-off between sample efficiency and diversity.

## 3.2 UTILIZE AUTOREGRESSIVE EXPLORATION BY $k$-OFFSPRING

How can we increase exploration within a closed-source LLM without finetuning? We exploit their autoregressive factorization to sample $k$ offspring per parent in a single call. Because later proposals can condition on earlier samples within the same context, this yields *diverse-but-plausible* edits with low orchestration overhead. This strategy provides a simple way to strengthen exploratory breadth: under a fixed evaluation budget, generating the same number of candidates requires fewer LLM queries, fewer prompt tokens overall, and less running time than issuing one-off calls. However, overly large $k$ may over-explore a local region and reduce marginal gains. We therefore study the trade-off by varying $k$ and report the resulting gains and generalization under both single- and multi-objective settings in the ablations.

## 3.3 HANDLING COMPLEX FEEBACK AND GENERALIZATION

The adapter normalizes all objectives to $[0, 1]$ to prevent any large-magnitude objective from dominating the fitness; because our fitness follows a "larger-is-better" convention, we convert minimization goals to maximization by taking 1 minus their normalized values. (ii) Constraints and textual feedback are converted into a concise, formatted message that highlights violations, near-feasible margins, and expert hints.When a constraint is critical and variable, we optionally promote it to an explicit objective (e.g., minimize scaffold similarity or stellarator error), improving stability without parameter updates. We empirically validate these design choices and their impact on optimization in Appendix 6.2.2.

Based on the efficient and effective evolving experience designed for large discrete search spaces, as well as the easy-to-scale $k$-offspring mechanism and the unified feedback adapter, our framework can be readily transferred to other problems and domains while maintaining strong performance. To facilitate adoption, we provide a simple template in which users only need to prepare two files: one specifying the task description and another defining the evaluation functions that return complex feedback. We demonstrate the strong performance of our framework across many problems and domains in Appendix 6.2.

## 4 EXPERIMENT

**Task Settings Five-objective optimization.** Following Wang et al. (2024a), we consider a five-objective molecular optimization task with a fixed evaluation budget of $B{=}5{,}000$ oracle calls. The objectives are: minimize SA (Synthetic Accessibility), DRD2 (Dopamine Receptor D2 affinity),

and GSK3$\beta$ (Glycogen Synthase Kinase 3 Beta); and maximize QED (Quantitative Estimate of Drug-likeness) and JNK3 (c-Jun N-terminal kinase 3). Although PMO (Gao et al., 2022) tasks are closely related to GuacaMol (Brown et al., 2019), PMO explicitly emphasizes budgeted evaluation due to the practical cost of property assessment (e.g., simulations or wet-lab proxies). A fixed budget thus encourages efficient exploration and enables fair comparison. While under a fixed budget, the initial population can substantially affect outcomes, yet this factor is often under-specified in evolution-based methods, and complete motivation is elaborated in Appendix 6.8 . To control for it, we compute fitness over ZINC250K (Irwin et al., 2012) and construct three fixed initial populations (size 100 each):**Best-init**: top-100 by fitness; **Worst-init**: bottom-100;**Random-init**: 100 uniformly sampled molecules. All algorithms are run from the same three initial populations, yielding a fair and comprehensive comparison. Best- and random-initialization reflect common practical use, whereas worst-initialization stresses robustness on a harder search. Beyond this five-objective setting, we follow the official PMO protocols and evaluate the full PMO benchmark to assess overall generality. For all experiments unless otherwise noted, ExLLM uses a fixed set of parameters with fixed proprietary LLM, experience injection probability $p_{\text{exp}}$=0.5, $k$=2, crossover probability 0.8, mutation probability 0.2, and population size 50. For the five-objective experiments, we reproduce MOLLEO using their code and use GPT-4o-2024-05-13 for both MOLLEO and our model.

**Metrics** Objectives are normalized and direction-unified as in Sec. 3.3; fitness $F$ follows Eq. equation 1 with equal weights, consistent with Wang et al. (2024a). **Hypervolume.** Multi-objective coverage on the normalized (maximization) objective vectors using the reference point $\mathbf{r} = (1.1, \ldots, 1.1)$; larger values indicate better Pareto coverage. **AUC** Area under the curve of $F$ versus oracle evaluations up to budget $B$; this rewards methods that reach high values with fewer oracle calls (Gao et al., 2022). **Validity**: fraction passing RDKit parsing. **Uniqueness**: fraction of unique molecules proposed. Novelty: fraction absent from ZINC250K. Diversity: average pairwise Tanimoto distance between the Morgan fingerprints within the top-100. **Efficiency** LLM queries, API cost (USD), wall-clock time (hours) under the same budget.

**Baselines** We compare against strong baselines spanning GA, BO, MCMC, RL, DL, and LLM-based methods: GB-GA, GB-BO, JT-VAE, MARS, REINVENT, MOLLEO, DyMol, and Genetic-GFN. For PMO-provided implementations (GB-GA, GB-BO, JT-VAE, MARS, REINVENT), we use the official PMO code and its default hyperparameters. For MOLLEO, DyMol, and Genetic-GFN, we use the authors' public code with their default settings as documented in code/papers. For RL-based baselines (REINVENT, DyMol, Genetic-GFN), we use the scalar fitness $F$ as the reward signal during training.

## 4.1 FIVE-OBJECTIVE EXPERIMENT RESULTS

| Metric | Initial | GB-GA | JT-VAE | GB-BO | MARS | REINVENT | MOLLEO | DyMol | Genetic-GFN | ExLLM(ours) |
|---|---|---|---|---|---|---|---|---|---|---|
| **(Worst initial)** | | | | | | | | | | |
| Top1 F | 2.689 | 4.048±0.114 | 3.838±0.042 | 3.665±0.129 | 3.891±0.018 | N/A | 4.096±0.155 | N/A | N/A | **4.229±0.050** |
| Top10 F | 2.683 | 4.019±0.101 | 3.784±0.027 | 3.647±0.135 | 3.852±0.019 | N/A | 4.044±0.157 | N/A | N/A | **4.186±0.029** |
| AUC-Top10 | N/A | 3.789±0.079 | 3.712±0.027 | 3.489±0.104 | 3.740±0.010 | N/A | 3.825±0.102 | N/A | N/A | **3.915±0.010** |
| Hypervolume | 0.163 | 0.474±0.190 | 0.364±0.075 | 0.167±0.050 | 0.488±0.110 | N/A | 0.720±0.172 | N/A | N/A | **0.737±0.038** |
| Diversity | 0.876 | 0.583±0.032 | 0.847±0.007 | 0.657±0.033 | 0.826±0.011 | N/A | 0.656±0.111 | N/A | N/A | 0.603±0.055 |
| Uniqueness | N/A | 0.786±0.032 | 1.000±0.000 | 1.000±0.000 | 0.488±0.128 | N/A | 0.672±0.032 | N/A | N/A | 0.829±0.021 |
| Validity | N/A | 1.000±0.000 | 1.000±0.000 | 1.000±0.000 | 1.000±0.000 | N/A | 0.930±0.075 | N/A | N/A | 0.937±0.010 |
| **(Random initial)** | | | | | | | | | | |
| Top1 F | 3.804 | 4.017±0.095 | 3.874±0.067 | 4.003±0.121 | 3.931±0.055 | 4.230±0.196 | 4.190±0.076 | 4.232±0.170 | 4.243±0.253 | **4.336±0.246** |
| Top10 F | 3.741 | 3.975±0.095 | 3.831±0.019 | 3.968±0.104 | 3.868±0.025 | 4.136±0.219 | 4.076±0.020 | 4.164±0.132 | 4.202±0.213 | **4.300±0.164** |
| AUC-Top10 | N/A | 3.861±0.052 | 3.771±0.011 | 3.802±0.057 | 3.789±0.011 | 3.930±0.133 | 3.949±0.021 | 4.001±0.054 | 4.078±0.150 | **4.116±0.040** |
| Hypervolume | 0.236 | 0.643±0.268 | 0.428±0.127 | 0.507±0.287 | 0.409±0.111 | 0.742±0.259 | 0.860±0.088 | 0.868±0.146 | 0.871±0.288 | **0.905±0.200** |
| Diversity | 0.884 | 0.623±0.047 | 0.778±0.012 | 0.717±0.017 | 0.819±0.015 | 0.640±0.111 | 0.670±0.015 | 0.581±0.069 | 0.633±0.066 | 0.494±0.032 |
| Uniqueness | N/A | 0.821±0.032 | 0.956±0.005 | 1.000±0.000 | 0.477±0.120 | 0.690±0.132 | 0.575±0.075 | 0.986±0.005 | 0.349±0.004 | 0.872±0.015 |
| Validity | N/A | 1.000±0.000 | 1.000±0.000 | 1.000±0.000 | 0.999±0.000 | 0.979±0.002 | 0.938±0.007 | 1.000±0.000 | 0.998±0.000 | 0.908±0.019 |
| **(Best initial)** | | | | | | | | | | |
| Top1 F | 4.329 | 4.583±0.154 | 4.329±0.000 | 4.605±0.047 | 4.419±0.074 | N/A | **4.699±0.000** | N/A | N/A | **4.699±0.000** |
| Top10 F | 4.132 | 4.582±0.167 | 4.132±0.000 | 4.467±0.066 | 4.181±0.029 | N/A | 4.564±0.064 | N/A | N/A | **4.628±0.043** |
| AUC-Top10 | N/A | 4.130±0.088 | 4.091±0.000 | 4.237±0.112 | 4.137±0.028 | N/A | 4.362±0.075 | N/A | N/A | **4.481±0.055** |
| Hypervolume | 0.917 | 0.968±0.183 | 0.917±0.000 | **1.275±0.027** | 0.975±0.041 | N/A | 1.168±0.106 | N/A | N/A | 1.175±0.067 |
| Diversity | 0.793 | 0.424±0.070 | 0.792±0.001 | 0.630±0.024 | 0.788±0.002 | N/A | 0.600±0.052 | N/A | N/A | 0.491±0.057 |
| Uniqueness | N/A | 0.729±0.041 | 1.000±0.000 | 1.000±0.000 | 0.432±0.053 | N/A | 0.678±0.000 | N/A | N/A | 0.942±0.006 |
| Validity | N/A | 1.000±0.000 | 1.000±0.000 | 1.000±0.000 | 0.999±0.000 | N/A | 0.913±0.022 | N/A | N/A | 0.790±0.024 |

Table 2: Unconstrained molecular design results, objectives: QED↑ + SA↓ + DRD2↓ + GSK3$\beta$ ↓ + JNK3↑

Table 2 reports mean $\pm$ std over five seeds; best scores are **bold** and second best are underlined. RL-based baselines (REINVENT, DyMol, Genetic-GFN) do not expose a mechanism to fix the initial population, so they are included only in the random-init condition. ExLLM attains the best $F$ and AUC across all settings. For hypervolume, ExLLM is best under **worst-init** and **random-init**, and second best under **best-init**. In both random- and worst-initialization, the **mean Top-10** $F$ of ExLLM exceeds the **Top-1** $F$ of the second-best method, indicating strong improvement under the same budget. Novelty w.r.t. ZINC 250K is near 100% across models and settings; since this offers little differentiation for our objectives, we omit it from the table for space. Under best-init, MOLLEO matches ExLLM on Top-1 $F$, but ExLLM's mean Top-10 $F$ and AUC remain higher than all other methods, suggesting more reliable batch-level gains. ExLLM maintains **80–95% uniqueness** across three settings, while MOLLEO is typically **55–70%**, because they use the same LLM, this indicates our framework has higher exploration ability. Validity is slightly lower than some baselines because we generate SMILES and discard invalid strings rather than post-hoc repairing them; this has negligible impact on final outcomes, and PMO notes SMILES is not necessarily inferior to 100%-valid representations (Gao et al., 2022). The diversity of the final top-100 set is somewhat lower, which reflects a common fitness–diversity trade-off: tighter, high-value exploration can reduce spread. Notably, ExLLM delivers substantial gains over the initial populations in all three init schemes, while trading some diversity for finer exploitation; coverage remains competitive as evidenced by strong hypervolume.

## 4.2 RESULTS ON PMO

| Task type | Objective(↑) | REINVENT | Augmented Memory | Graph GA | GP BO | MOLLEO | Genetic GFN | ExLLM (Ours) |
|---|---|---|---|---|---|---|---|---|
| Property optimization | QED | 0.941 ± 0.000 | 0.941 ± 0.000 | 0.940 ± 0.000 | 0.937 ± 0.000 | **0.948 ± 0.000** | 0.942 ± 0.000 | 0.943 ± 0.000 |
| | JNK3 | 0.783 ± 0.023 | 0.773 ± 0.073 | 0.564 ± 0.155 | 0.564 ± 0.155 | **0.790 ± 0.027** | 0.764 ± 0.069 | 0.732 ± 0.078 |
| | DRD2 | 0.945 ± 0.007 | 0.962 ± 0.005 | 0.964 ± 0.012 | 0.923 ± 0.017 | 0.968 ± 0.012 | 0.974 ± 0.006 | **0.980 ± 0.003** |
| | GSK3$\beta$ | 0.865 ± 0.043 | **0.889 ± 0.027** | 0.788 ± 0.070 | 0.851 ± 0.041 | 0.863 ± 0.047 | 0.881 ± 0.042 | 0.818 ± 0.050 |
| Name-based optimization | mestranol_similarity | 0.618 ± 0.048 | 0.764 ± 0.035 | 0.579 ± 0.022 | 0.627 ± 0.089 | 0.972 ± 0.009 | 0.708 ± 0.057 | **0.980 ± 0.005** |
| | albuterol_similarity | 0.896 ± 0.008 | 0.918 ± 0.026 | 0.874 ± 0.020 | 0.902 ± 0.019 | 0.985 ± 0.024 | 0.949 ± 0.010 | **0.989 ± 0.000** |
| | thiothixene_rediscovery | 0.534 ± 0.013 | 0.562 ± 0.028 | 0.479 ± 0.025 | 0.559 ± 0.027 | 0.727 ± 0.052 | 0.583 ± 0.034 | **0.910 ± 0.004** |
| | celecoxib_rediscovery | 0.716 ± 0.084 | 0.784 ± 0.011 | 0.582 ± 0.057 | 0.728 ± 0.048 | 0.864 ± 0.034 | **0.891 ± 0.033** | **0.891 ± 0.033** |
| | troglitazone_rediscovery | 0.452 ± 0.048 | 0.556 ± 0.052 | 0.377 ± 0.010 | 0.405 ± 0.007 | 0.562 ± 0.019 | 0.511 ± 0.054 | **0.726 ± 0.111** |
| | perindopril_mpo | 0.537 ± 0.016 | 0.598 ± 0.008 | 0.538 ± 0.009 | 0.493 ± 0.011 | 0.600 ± 0.031 | 0.595 ± 0.014 | **0.797 ± 0.016** |
| | ranolazine_mpo | 0.760 ± 0.009 | 0.802 ± 0.003 | 0.728 ± 0.012 | 0.735 ± 0.013 | 0.769 ± 0.022 | 0.819 ± 0.018 | **0.855 ± 0.021** |
| | sitagliptin_mpo | 0.021 ± 0.003 | 0.479 ± 0.039 | 0.433 ± 0.075 | 0.186 ± 0.055 | 0.584 ± 0.067 | **0.634 ± 0.039** | 0.555 ± 0.048 |
| | amlodipine_mpo | 0.642 ± 0.044 | 0.686 ± 0.046 | 0.625 ± 0.040 | 0.552 ± 0.025 | 0.773 ± 0.037 | 0.761 ± 0.019 | **0.874 ± 0.010** |
| | fexofenadine_mpo | 0.769 ± 0.009 | 0.686 ± 0.010 | 0.779 ± 0.025 | 0.745 ± 0.009 | 0.847 ± 0.018 | 0.856 ± 0.039 | **0.984 ± 0.006** |
| | osimertinib_mpo | 0.834 ± 0.046 | 0.856 ± 0.013 | 0.808 ± 0.012 | 0.762 ± 0.029 | 0.835 ± 0.024 | 0.860 ± 0.008 | **0.902 ± 0.018** |
| | zaleplon_mpo | 0.347 ± 0.049 | 0.438 ± 0.082 | 0.456 ± 0.007 | 0.272 ± 0.026 | 0.510 ± 0.031 | 0.552 ± 0.033 | **0.723 ± 0.007** |
| | median1 | 0.372 ± 0.015 | 0.335 ± 0.012 | 0.287 ± 0.008 | 0.325 ± 0.012 | 0.352 ± 0.024 | 0.379 ± 0.010 | **0.384 ± 0.007** |
| | median2 | 0.294 ± 0.006 | 0.290 ± 0.006 | 0.229 ± 0.017 | 0.308 ± 0.034 | 0.275 ± 0.045 | 0.294 ± 0.007 | **0.475 ± 0.002** |
| Structure-based optimization | isomers_c7h8n2o2 | 0.842 ± 0.029 | 0.954 ± 0.033 | 0.949 ± 0.036 | 0.662 ± 0.071 | **0.984 ± 0.008** | 0.969 ± 0.003 | **0.984 ± 0.001** |
| | isomers_c9h10n2o2pf2cl | 0.642 ± 0.054 | 0.830 ± 0.016 | 0.719 ± 0.047 | 0.469 ± 0.180 | 0.874 ± 0.053 | 0.897 ± 0.007 | **0.961 ± 0.028** |
| | deco_hop | 0.666 ± 0.044 | 0.688 ± 0.060 | 0.619 ± 0.004 | 0.629 ± 0.018 | 0.942 ± 0.013 | 0.733 ± 0.109 | **0.956 ± 0.014** |
| | scaffold_hop | 0.560 ± 0.019 | 0.565 ± 0.008 | 0.517 ± 0.007 | 0.548 ± 0.019 | **0.971 ± 0.004** | 0.615 ± 0.100 | 0.916 ± 0.127 |
| | valsartan_smarts | 0.000 ± 0.000 | 0.000 ± 0.000 | 0.000 ± 0.000 | 0.000 ± 0.000 | **0.867 ± 0.092** | 0.135 ± 0.271 | 0.831 ± 0.043 |
| | **Total (↑)** | 14.036 | 15.356 | 13.823 | 13.182 | 17.862 | 16.213 | **19.165** |
| | **Rank (↓)** | 7 | 5 | 8 | 9 | 2 | 4 | **1** |

Table 3: Top-10 AUC of tasks in PMO (Gao et al., 2022) benchmark, including single-objective optimization and multi-objective optimization for 3 task types. ExLLM attains a total score of **19.165** (**+7.3%** improvement compared to the previous SOTA by MOLLEO), ranking **first in 17/23 tasks** and achieving the highest overall ranking. The models with 3rd and 6th totao score are in the additional LLM-based results in Appendix 6.4.

We evaluate ExLLM on the full PMO suite covering property, name-based, and structure-based optimization (Table 3). Best values are **bold**; the second and third best are underlined. Following the official scoring, we report an aggregate leaderboard score with a maximum of 23 points. ExLLM attains a total score of 19.165, ranking first in 17/23 tasks and achieving the highest overall ranking. Compared to the previous SOTA (17.862), this corresponds to a +7.3% improvement in the aggregate score. To assess the contribution of the experience module, we also report an ablation without the evolving experience (ExLLM w/o experience), which still reaches **18.165**. This result suggests that the k-offspring exploration is a strong contributor on its own, while the evolving experience provides consistent additional gains overall.

Table 4: MOCPOP benchmark. ExLLM achieves SOTA on MOCVRP and ranks highly on MOTSP.

| Method | MOTSP↑ | MOCVRP↑ |
|---|---|---|
| Pymoo | 0.983 | 0.956 |
| ReEvo | 1.029 | 1.035 |
| AIDE | 1.021 | 1.006 |
| FunSearch | 1.023 | 1.032 |
| AlphaEvolve | **1.0293** | 1.0318 |
| **ExLLM** | 1.0273 | **1.0704** |

Table 5: Offshore platform jackets optimization. ExLLM finds the lightest feasible design.

| Method | Weight↓ | Stress↓ |
|---|---|---|
| Human | 218.0 | 0.024 |
| GA | 32.24 | 0.093 |
| MOEAD | 37.76 | 0.137 |
| RS | 49.33 | 0.435 |
| Ablation | 14.90 | 0.814 |
| **ExLLM** | **13.60** | **0.508** |

Table 6: Stellarator optimization (ConStellaration). ExLLM exceeds official SOTA on P2/P3.

| Method | P2↑ | P3↑ |
|---|---|---|
| trust-constr | Fail | Fail |
| COBYQA | Fail | Fail |
| ALM–NGOpt | 0.431 | 129.796 (97) |
| **ExLLM** | **0.505** | **133.634 (103)** |

Table 7: Circle packing (n=26–32), rotated for compact layout. ExLLM improves all best-known results.

| n | Record | ExLLM |
|---|---|---|
| 26 | 2.635977 (AlphaEvolve (Novikov et al., 2025a)) | **2.635983** |
| 27 | 2.685+ (Friedman, 2025) | **2.68598** |
| 28 | 2.737+ (Friedman, 2025) | 2.73740 |
| 29 | 2.790+ (Friedman, 2025) | 2.79034 |
| 30 | 2.842+ (Friedman, 2025) | 2.84267 |
| 31 | 2.889+ (Friedman, 2025) | 2.88997 |
| 32 | 2.937+ (AlphaEvolve (Novikov et al., 2025a)) | 2.939+ |

Table 8: NK2R peptide design (AlphaFold3 ipTM). ExLLM peptides outperform natural ligand NKA.

| Peptide | ipTM(NKA) | ipTM(ours) | Δ |
|---|---|---|---|
| P1 | 0.09 | 0.87 | 0.78 |
| P2 | 0.10 | 0.88 | 0.78 |
| P3 | 0.12 | 0.88 | 0.76 |
| P4 | 0.12 | 0.88 | 0.76 |
| P5 | 0.11 | 0.86 | 0.75 |
| P6 | 0.10 | 0.84 | 0.74 |
| P7 | 0.13 | 0.87 | 0.74 |
| P8 | 0.14 | 0.88 | 0.74 |
| P9 | 0.11 | 0.85 | 0.74 |
| P10 | 0.13 | 0.87 | 0.74 |

## 4.3 EXTENDED EXPERIMENTS

**Cross-domain transfer.** To fully demonstrate the generalizability of ExLLM, we conduct experiements in multiple domains, including circle packing (mathematics), stellarator coil optimization (physics), and offshore jacket structural analysis (offshore structural engineering), MOTSP and MOCVRP (classical optimization problems), NK2R peptide design (peptide-drug discovery), GCU operator optimization (high performance computing kernel design). ExLLM achieves new records on circle packing and stellarator design benchmark (ConStellaration Cadena et al. (2025)), and achieves SOTA performance on MOCVRP and offshore jacket structural optimization, and competitive performance on MOTSP, peptide deisgn and kernel optimziation.

**Circle packing in a unit square** Given an integer $n$, the task is to place $n$ congruent circles inside the unit square while maximizing the common radius $r$ subject to two strict feasibility constraints: (i) no overlaps, $\|c_i - c_j\|_2 \geq 2r$, and (ii) all circle centers must satisfy $r \leq x_i, y_i \leq 1 - r$. The optimization landscape is very unstable, sub-millimeter changes in coordinates easily break feasibility, and the input state $(x_i, y_i, r)$ provides no structural cues for the model. Only a single scalar objective (the sum of radius) is returned, making gradient-free global search difficult. As shown in Table 7, ExLLM attains **new best-known** radii for $n = 26$ and $n = 32$ and **matches** all published records for $n = 27$–31. Existing records for these middle cases are available only to three decimal places, so it is unclear whether ExLLM further improves beyond that precision. These results are achieved with the same hyperparameters used in all other domains, demonstrating ExLLM's ability to solve highly sensitive geometric–continuous problems under strict feasibility constraints.

**Stellarator Design** The ConStellaration benchmark defines two quasi-isodynamic (QI) stellarator optimization tasks: (P2) maximizing a coil-friendly QI score under five physics constraints, and (P3) jointly optimizing MHD stability and coil simplicity under the five physics constraints as well. A design is scored only if *all* constraints are satisfied; otherwise, its score is zero. The feasible region is extremely narrow, and small geometric deviations can simultaneously violate multiple plasma-physics constraints. Standard baselines (trust-constr, COBYQA) fail to find any feasible point. As shown in Table 6, ExLLM consistently identifies feasible solutions and achieves new SOTA on both tasks: a **17%** improvement on P2 (0.505 vs. 0.431), a **3%** improvement on P3 (133.634

vs. 129.796) and a **6%** improvement on the single point(97 vs. 103) on P3. This demonstrates that ExLLM can navigate fragile multi-constraint physics landscapes without any domain-specific tuning.

**Offshore jacket optimization** The SACS benchmark requires optimizing offshore jacket structures under three competing engineering goals: minimizing total weight, while satisfying axial-load and bending/torsional stress limits. Each candidate is evaluated by commercial SACS finite-element analysis, which provides only black-box outputs (weight and stress ratios). The search space mixes discrete structural-section choices with continuous dimensions, and SACS evaluations are costly, making sample efficiency critical. Table 5 shows that ExLLM finds a feasible design weighing only **13.6 tons**, more than **93%** lighter than the 218-ton human baseline, while satisfying all stress constraints. It also outperforms GA, MOEAD, and random search. This highlights ExLLM's capacity to handle mixed-variable engineering optimization with expensive, black-box oracles.

**MOCPOP** The Multi-Objective Traveling Salesman Problem (MOTSP) and the Multi-Objective Capacitated Vehicle Routing Problem (MOCVRP) tasks involve large-scale ($n = 100$) routing under strict permutation and capacity constraints. Both problems exhibit strong multi-objective conflict: small changes in coordinates or routes can drastically alter feasibility or the relative objective ordering, and scalarized heuristics often collapse the Pareto front. As reported in Table 4, ExLLM achieves **new SOTA hypervolume** on MOCVRP and ranks second only to AlphaEvolve on MOTSP, outperforming strong baselines such as Pymoo, ReEvo, AIDE, and FunSearch. These results show that ExLLM effectively reasons over discrete combinatorial structures and generates high-quality, feasible offspring without relying on brittle crossover heuristics.

**NK2R peptide design** The NK2R task requires designing peptides that (i) share $< 30\%$ sequence similarity with the native ligand NKA and (ii) achieve higher AlphaFold3 complex ipTM with NK2R. Because similarity rules, formatting constraints, and ipTM comparison to NKA jointly determine success, most candidates score zero and the landscape is extremely sparse. Table 8 shows that ExLLM produces many peptides surpassing NKA within 1,000 evaluations, with best margins around **+0.78**. All reported peptides satisfy the similarity and length constraints. This indicates that ExLLM can perform constrained molecular design using only AF3 black-box feedback.

**GCU operator design** This domain evaluates whether ExLLM can generate high-performance GPU kernels for Tencent's GCU across three operators (`Var`, `SiLU`, `GEMM v2`). The task is challenging due to a new programming model, scarce public examples, strict compiler requirements, and 5–10 minute compile–run cycles where naive LLM code fails $> 90\%$ of the time. With only 300 evaluations per operator, ExLLM raises compile success from $< 10\%$ to **about 85%**, produces fully correct kernels, and significantly improves execution latency. These submissions placed in the **top-10** of the Tencent Kaiwu 2025 competition qualifier and earned a **second prize award**, with all submitted kernels fully LLM-generated and no manual code edits. This demonstrates ExLLM's ability to operate in real-world domain-shifted, performance-critical code-generation settings.

## 5 CONCLUSION

We presented **ExLLM**, an LLM-as-optimizer framework for large discrete optimization, instantiated for multi-objective molecular design. The method couples three simple components: a *single, evolving experience* that distills non-redundant cues at low cost; a *k-offspring* sampling scheme that widens exploration per query; and a *feedback adapter* that normalizes objectives for selection while formatting constraints and expert hints for iteration. Under a fixed evaluation budget, ExLLM attains state-of-the-art results on PMO (total score **19.165**, ranking first on **17/23** tasks, **+7.3%** over prior SOTA) and generalizes beyond chemistry, setting new records on circle packing and stellarator design while delivering strong performance across additional domains including offshore jacket optimization, MOCPOP benchmark, peptide design and GCU kernel optimization. The probabilistic experience injection improves efficiency without over-constraining exploration. Looking forward, we plan to (i) adapt $k$ and the experience-injection rate online, (ii) incorporate oracle uncertainty and 3D/physics-informed feedback, and (iii) broaden evaluation to additional scientific design tasks. We release templates for rapid transfer, lowering the barrier to applying ExLLM as a general optimizer for large discrete spaces.

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

# 6 APPENDIX

## 6.1 REPRODUCIBILITY

We commit to releasing the full implementation upon acceptance of the paper. The public repository will include all code to reproduce the results reported in this work, covering not only the experiments in the main text but also the extended templates and evaluation functions for other domains. All prompt templates used in our optimizer will also be released to ensure transparency and reproducibility.

## 6.2 MORE APPLICATIONS

### 6.2.1 CIRCLE PACKING IN A UNIT SQUARE

**Task.** Place $n$ circles inside a unit square to maximize the common radius without overlap or boundary violations; this classic geometric packing problem has long-standing community records and curated best-known configurations (Friedman, 2025; Novikov et al., 2025a).

**Challenges for our optimizer.** (1) *End-to-end instability*: tiny coordinate nudges can flip feasibility, making naive prompt-only updates easily to voliate the constraints. (2) *Variable abstraction*: the raw state of the problem is described only by a list of circle centers and their individual radii $(x_i, y_i, r_i)$. This representation is highly abstract and purely numerical, making it difficult for an LLM to derive intuitive geometric insights directly from the prompt. And the feedback is only the total radius.

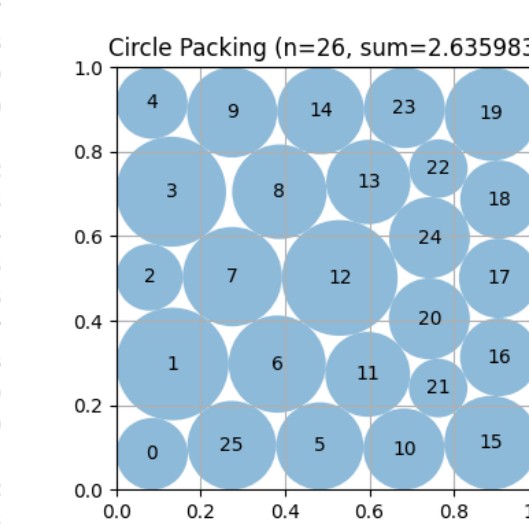
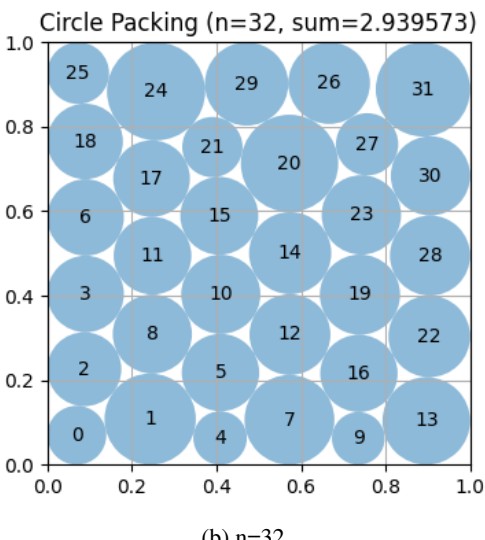

(a) n=26

(b) n=32

Figure 3: Best layouts found by our optimizer: (left) $n = 26$, (right) $n = 32$.

**Our approach.** We keep the ExLLM optimizer fixed (same hyperparameters as in the main text) and modify only the task template and the evaluation function. The template succinctly describes the target (pack $n$ circles in a unit square) and specifies a simple output format (center coordinates and radii). To stabilize end-to-end outcomes, we post-process ExLLM's proposals with a *fixed* off-the-shelf solver—SLSQP from *SciPy*—which enforces non-overlap and boundary constraints and locally increases the radii given the LLM's initial centers/radii. In contrast to methods that retrain or tune a bespoke solver (e.g., evolving the solver itself), we hold the optimizer constant and use a fixed solver purely for feasibility/finetuning, while ExLLM focuses on generating strong initializations.

**Results and visualization.** All results are shown in Figure 3 and table 7. In our setup, we obtain new best-known results for $n = 26$ and $n = 32$. For $n = 27$–31, our solutions match the publicly reported records up to the available three-decimal precision; because reference values beyond three decimals are not published, we cannot determine whether we strictly surpass them. We include figures of our best layouts and a table of achieved radii $n$. Notably, our algorithm often discovers multiple

distinct arrangements that attain the same maximal score. The experiments were conducted with 2500 evaluation budget, and it takes about only 3 hours to complete and find achieved new records.

### 6.2.2 STELLARATOR DESIGN

**Task.** Optimize quasi-isodynamic (QI) stellarator plasma boundaries under strict physics and engineering constraints using the ConStellaration benchmark (Cadena et al., 2025): (P2) *Simple-to-build QI* (favor coil simplicity subject to QI/geometry constraints) and (P3) *MHD-stable QI* (trade off compactness vs. coil simplicity under MHD stability and turbulence-proxy constraints). We adopt the official scoring and evaluation protocol of the benchmark release.

**Challenges for our optimizer.** (1) Constraint hardness & multiplicity: Problem 2 is single objective optimization with 5 constraints, and problem 3 is even harder with 2 objective optimization and 5 constraints including edge rotational transform, quasi isodynamicity residual, vacuum well etc. (2) Sparse nonzero solution: setting the number of field periods to 3 according to official code, sweeping the about 180k released designs yields no nonzero scores under the strict benchmark tolerances. (3) The official gradient-based (`scipy-trust-constr`) and derivative-free (`COBYQA`) baselines fail to produce feasible points on P2/P3; only the ALM–NGOpt pipeline attains nonzero scores, and sparsely so.

**Our approach.** We keep the ExLLM optimizer fixed (same hyperparameters as in the main text) and change only the task template and evaluation to match P2/P3. Because feasibility is *hard*—any violation yields a zero score—and the key feasibility indicator is a measurable scalar (the feasibility score must be <0.01), we treat this constraint as an **explicit objective**. Concretely, besides using the official targets, we add a feasibility term that rewards minimizing feasibility score; this follows our adapter's rule for promoting critical, variable constraints into objectives. In practice, this substantially accelerates convergence to nonzero scores and stabilizes progress, the proportion of nonzero scores largely increases.

**Results and visualization.** On both benchmarks we surpass the official SOTA (ALM–NGOpt), obtaining many feasible points in a single run while tightly respecting constraints, 17% improvement on problem 2 and 3% improvement on problem 3. Figure 4 shows visualization of our best solutions; Table 6 compares scores.

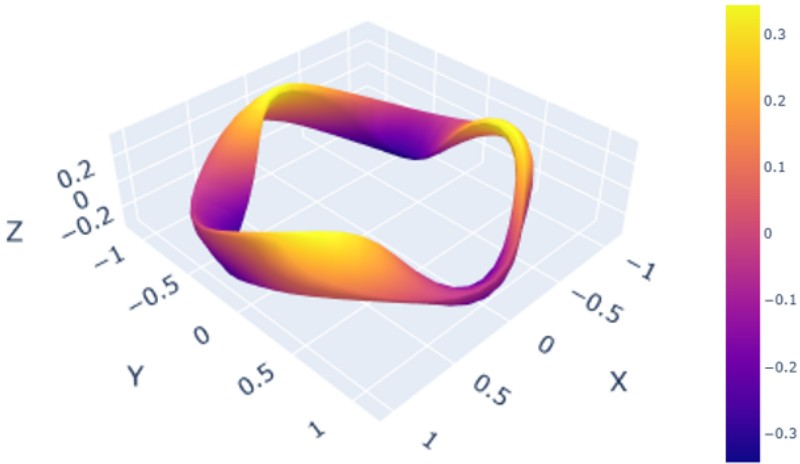

Figure 4: A visualization of our best solution for problem 2.

### 6.2.3 MOCPOP (MULTI-OBJECTIVE COMBINATORIAL PATH OPTIMIZATION PROBLEMS)

**Task.** The Multi-Objective Traveling Salesman Problem (MOTSP) and the Multi-Objective Capacitated Vehicle Routing Problem (MOCVRP) are both classical NP-hard combinatorial optimization tasks. MOTSP seeks a single Hamiltonian circuit that, starting and ending at a given depot, visits every city exactly once while simultaneously minimizing multiple conflicting objectives. MOCVRP

designs a set of capacitated vehicle routes that originate from a common depot, serve all customer demands, and jointly minimize the total travel distance and the makespan. All benchmark instances used in this study were generated with the scalable generator proposed by Lin et al. (2022)

**Challenges for our optimizer.** (1) Implicit dimensional coupling: MOTSP exhibits a scale–sensitive embedding, a microscopic rescaling of one coordinate can flip the relative distances of a city pair from the global minimum to the global maximum. Consequently, the gradient direction that appears Pareto-improving in one scalarization step may become strongly misleading after an infinitesimal $\delta$-shift, rendering classical directional updates ineffective and causing severe oscillations on the Pareto front. (2) Pareto-conflict trap: In MOCVRP, the total travel distance and the makespan form a deeply antagonistic pair, minimizing the former tends to squeeze all routes into a few "spokes," whereas minimizing the latter forces a balanced, yet longer, set of tours. A single scalarized surrogate—no matter how sophisticated the weight schedule—collapses the true Pareto front into a narrow ridge and inevitably misses the knee regions that dominate most practical trade-offs. (3) Destructive heuristic search bias: Offspring generated by canonical route-based crossover (e.g., OX, MPX) inherit disconnected fragments or violate capacity constraints with $> 90\,\%$ probability. The resulting infeasible individuals are rejected outright, so the search wastes the vast majority of evaluations and the effective population size collapses—an effect we term "lethal recombination drag." Advanced repair mechanisms partially alleviate the issue, yet they simultaneously erode the schema that parental high-fitness routes originally encoded, decelerating convergence toward the Pareto set

**Our approach.** For MOCPOP, our ExLLM employs deliberately fuzzy prompts that fully exploit the intrinsic experience and reasoning capacity of the large language model. The model proposes inference-based, feasible offspring instead of rigidly applying heuristics such as OX or PMX, which would otherwise produce a high proportion of invalid children. We also integrate an LLM + Solver scheme: after the large model generates high-quality feasible solutions, a solver—guided by a heuristic also proposed by the LLM—further refines these solutions to produce a superior population. This hybrid strategy guarantees both the quality and the speed of ExLLM's search.

**Results.** Across both benchmarks, ExLLM attains highly competitive performance (Table 5). MOTSP instances comprise $n = 100$ cities, and MOCVRP instances comprise $n = 100$ customers and $m = 20$ vehicles. We adopt hyper-volume as the universal performance indicator. Baselines include the human-designed solver Pymoo (Blank & Deb, 2020), as well as recent search-based algorithms ReEvo (Ye et al., 2024), AlphaEvolve (Novikov et al., 2025b), and other representative model. ExLLM achieves new SOTA hyper-volume on MOCVRP and ranks second only to AlphaEvolve on MOTSP, demonstrating the effectiveness of the LLM-guided hybrid paradigm.

### 6.2.4 SACS (STRUCTURAL ANALYSIS COMPUTER SYSTEM)

**Task.** To validate the cross-domain transferability of our framework, we apply ExLLM to a challenging real-world engineering problem: the structural optimization of offshore platform jackets. This system intelligently adjusts the dimensions of structural components to simultaneously achieve three conflicting goals: 1) Minimize the structural weight to reduce material and construction costs; 2) Ensure structural strength is sufficient to withstand extreme axial load conditions; and 3) Ensure structural strength is sufficient to resist extreme bending and torsional loads. The optimization is performed using the SACS finite element analysis software as the evaluation oracle.

**Challenges for our optimizer.** (1) *Complex mixed-variable parameter space*: The optimization involves a hybrid of discrete and continuous variables. It requires selecting standard cross-sections from a catalog using discrete string representations, while simultaneously optimizing other continuous design parameters. This mixed-variable nature presents a significant challenge for many optimization algorithms. (2) *Computationally expensive feedback loop*: Each proposed design must be evaluated using the SACS finite element software, which involves computationally intensive simulations. This places a strict limit on the number of evaluations possible, demanding high sample efficiency from the optimizer. (3) *Balancing competing objectives*: The goals of minimizing weight while maximizing strength against two different types of critical loads are inherently contradictory. Finding a balanced solution on the Pareto front requires sophisticated exploration and exploitation strategies. (4) *Black-box feedback*: The optimizer receives performance metrics (weight, stress, displacement) from SACS without direct access to the software's internal gradients, treating it as a black box.

**Our approach.** Demonstrating the framework's 'plug-and-play' capability, we kept its hyperparameters identical to those used in the molecular design experiments. The transfer to this new domain required only the creation of a task-specific template—which instructs the ExLLM to propose modifications to a vector of design parameters—and a corresponding evaluation function that interfaces with the SACS software. The feedback adapter then processes the results from SACS—structural weight and stress safety factors—and formats them for the next optimization iteration. The evolving experience mechanism is particularly valuable for identifying promising design patterns (e.g., which members are most sensitive to change) and avoiding repeated evaluation of inferior designs, which is critical given the high cost of SACS simulations.

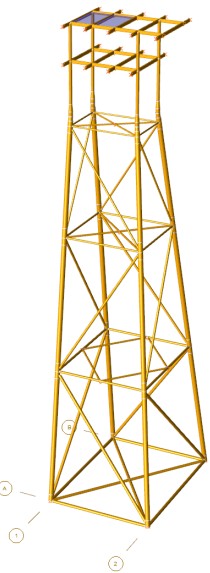 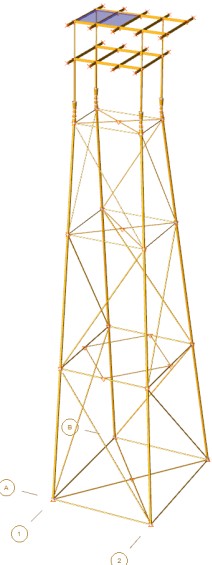

Figure 5: Comparison of the offshore jacket structure before and after optimization. Through structural optimization, the member dimensions were adjusted to satisfy all safety and performance constraints (e.g., stress, displacement), resulting in a total weight reduction of 93%.

**Results.** By applying ExLLM, we successfully automated the iterative design process for the offshore jacket structure. As shown in Table 5, our LLM-based approach discovered a design with a final weight of only 13.6 tons, a reduction of over 93% compared to the 218.0-ton human-designed baseline, while maintaining the maximum stress ratio within safe limits (<1.0). This result significantly outperforms traditional optimization algorithms like Genetic Algorithm (GA), MOEAD, and Random Search (RS) in terms of final structural weight. The convergence plots in Figure 6 further illustrate the efficiency of our method. The ExLLM optimizer demonstrates substantially faster convergence and achieves superior final values for both the Top 10 Fitness (F) and hypervolume metrics compared to other baselines. This indicates that the ExLLM framework not only finds better solutions but also does so with significantly fewer evaluation calls, a critical advantage for computationally expensive, real-world engineering problems.

### 6.2.5 PEPTIDE DESIGN FOR NK2R

**Task and background.** This task comes from the peptide design competition organized by Tsinghua University's FBS(Frontiers In Biological Structures) lab, aiming to discover anti-obesity therapeutics via the Neurokinin-2 receptor (NK2R)[1]. NK2R (UniProt P21452) is a promising target because activating it can reduce food intake via central mechanisms and increase energy expenditure peripherally; thus, efficacious NK2R agonists could provide dual-action weight-loss benefits (Sass et al., 2024). It requires designing short peptide agonists that can outcompete the native ligand Neurokinin A (NKA) at NK2R under the competition's evaluation protocol (AlphaFold3 complex modeling with NK2R, G$\alpha$, NKA and designed peptide).

---

[1]https://www.fbs.frcbs.tsinghua.edu.cn/competition/2025Peptide

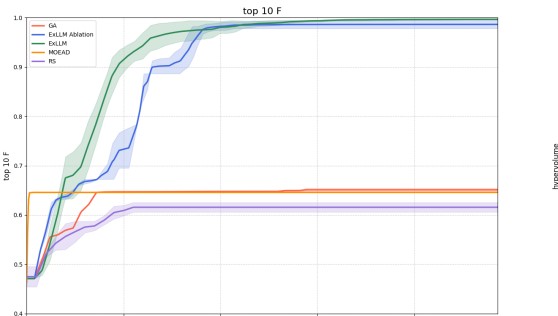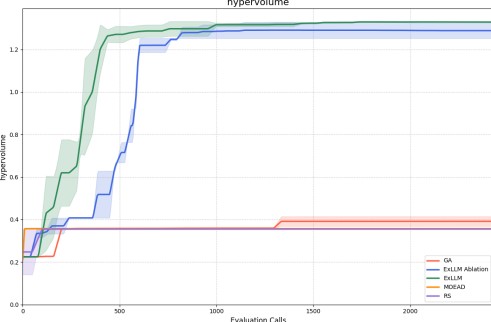

Figure 6: Performance comparison on the SACS benchmark. (Left) Top 10 F metric vs. evaluation calls. Our ExLLM methods show significantly faster convergence and achieve a higher final fitness score. (Right) Hypervolume evolution vs. evaluation calls. The ExLLM approaches achieve a substantially larger and more stable hypervolume, demonstrating superior performance in balancing the multi-objective trade-offs.

**Challenge specification.** Candidates must satisfy two hard success criteria: (i) sequence similarity to NKA (`HKTDSFVGLM`) must be $< 30\%$ by `mmseqs2`; (ii) under the same AlphaFold3 setup, the peptide's complex ipTM with NK2R must exceed the ipTM of NKA (i.e., $\text{ipTM}_{\text{ours}} > \text{ipTM}_{\text{NKA}}$). Higher ipTM is better, and successful designs will proceed to wet-lab synthesis and activity testing in later rounds.

**Challenges for our optimizer.** (1) *Hard constraints dominate success*: without meeting the similarity/length filters and the "beat-NKA" ipTM criterion, candidates effectively score zero. (2) *Sparse, coupled feedback*: AlphaFold3 provides a small set of structural scores (e.g., ipTM), and the success condition is relative to NKA rather than an absolute threshold, coupling objectives and constraints tightly.

**Our approach.** We keep the ExLLM optimizer fixed (same hyperparameters as in the main text) and provide the task template and evaluation function using AlphaFold3. The task template compresses all rules (length, similarity, formatting) and objectives into a compact, structured description; the feedback adapter enforces filters and exposes per-iteration diagnostics. We optimize a two-term objective: maximize $\text{ipTM}_{\text{ours}}$ (peptide–NK2R complex) and minimize $\text{ipTM}_{\text{NKA}}$ under the same NK2R target, subject to the length and `mmseqs2` similarity constraints. We start from 25 random UniProt peptides ($\leq 40$ aa); none beat NKA initially. Under a budget of 1,000 evaluations, ExLLM designs many candidates with positive margins.

**Results.** Table 8 lists representative top candidates (sorted by $\Delta$); all reported sequences satisfy the competition's length and similarity filters in our screen. Notably, the best margin reaches $+0.67$ ipTM, and we routinely observe dozens of non-zero successes in a single run.

### 6.2.6 GCU OPERATOR DESIGN

**Task.** This track targets high-performance kernel development on Tencent's GCU using the official operator SDK and `TopsCC` toolchain; submissions are auto-graded for both correctness and latency across 10 test cases per operator. The qualifier includes three operators (`Var`, `SiLU`, `GEMM v2`). Competition page: `https://aiarena.tencent.com/aiarena/zh/match/open-competition-2025`.

**Challenges for our optimizer.** (1) *Limited public expertise.* GCU is a new accelerator; kernels are built on a new framework, so CUDA-centric patterns from LLM pretraining transfer poorly. (2) *Low naïve compile success.* Direct LLM-generated kernels compile successfully in $< 10\%$ of attempts. (3) *Long iteration latency.* Each compile–run test takes 5–10 minutes, making failed trials especially costly and throttling exploration.

**Our approach.** We keep **ExLLM** fixed (same hyperparameters as in the main text) and adapt only the task template and evaluation harness. First, we distill the official SDK documentation into a few-hundred-word *prior* (key APIs, vectorization rules, memory/resource limits) and inject

it into the prompt. Second, the feedback adapter returns compact, structured diagnostics from `TopsCC`—compiler errors, resource overuse (register/SMEM), numeric checks, and latency. These messages are surfaced verbatim to ExLLM and summarized into the evolving experience so that subsequent generations see parents' error traces plus accumulated general rules summarized by evolving experience.

**Results.** With only **300 evaluations per operator**, ExLLM-produced kernels pass all platform accuracy checks and substantially reduce latency for `Var`, `SiLU`, and `GEMM v2`. Our compile success rate rises from $< 10\%$ (naïve prompting) to **85%** with error-aware feedback and evolving experience, enabling rapid iteration despite 5–10 minute test cycles. The submission ranked in the **top-10** of the qualifier and was close to the top score (exact rank withheld for anonymity). All submitted kernels were authored by ExLLM under the above template and harness, with **no human-written lines of code**—every kernel was fully model-generated without manual edits.

## 6.3 ABLATION STUDY

To fully characterize the framework, we ask: (1) how to determine $k$ and what's the trade-off?; (2) how frequently should evolving experience be injected; (3) is the hybrid selector better using only one of them? (4) what is performance of different proprietary and open-source LLM in ExLLM? (5) how does performance change with number of objectives; (6) how should constraints be handled for stable, budgeted gains? We hence quantify budgeted gains vs. local over-exploration by varying $k \in \{1, \ldots, 6\}$ in single-/multi-objective settings with two LLM backbones. And then identify when experience helps vs. hinders exploration by sweeping $p_{\exp} \in [0, 1]$. We finally Tests scalability from 1 to 6 objectives including comparison with MOLLEO. Detailed tables and the $4^{th}$ ablation study are in the appendix.

### 6.3.1 $k$-OFF STRATEGY

| Method | Top10 F | Top10 AUC | Uniqueness | Validity | Top10 F | Top10 AUC | Uniqueness | Validity |
|---|---|---|---|---|---|---|---|---|
| | ExLLM (Gemini-2.5-Flash) | | | | ExLLM (GPT-4o) | | | |
| 1 offspring | $0.643 \pm 0.105$ | $0.501 \pm 0.057$ | $0.376 \pm 0.079$ | $0.984 \pm 0.006$ | $0.861 \pm 0.044$ | $0.644 \pm 0.025$ | $0.895 \pm 0.004$ | $0.822 \pm 0.035$ |
| 2 offsprings | $\mathbf{0.764 \pm 0.033}$ | $\mathbf{0.558 \pm 0.014}$ | $\mathbf{0.686 \pm 0.068}$ | $0.978 \pm 0.009$ | $0.877 \pm 0.043$ | $0.698 \pm 0.038$ | $0.927 \pm 0.018$ | $\mathbf{0.835 \pm 0.030}$ |
| 3 offsprings | $0.757 \pm 0.105$ | $0.603 \pm 0.094$ | $0.646 \pm 0.057$ | $0.979 \pm 0.008$ | $\mathbf{0.908 \pm 0.033}$ | $0.709 \pm 0.068$ | $0.940 \pm 0.046$ | $0.817 \pm 0.026$ |
| 4 offsprings | $0.618 \pm 0.101$ | $0.505 \pm 0.111$ | $0.542 \pm 0.046$ | $0.986 \pm 0.002$ | $0.834 \pm 0.127$ | $0.674 \pm 0.109$ | $\mathbf{0.972 \pm 0.018}$ | $0.751 \pm 0.062$ |
| 5 offsprings | $0.650 \pm 0.100$ | $0.451 \pm 0.077$ | $0.629 \pm 0.073$ | $\mathbf{0.987 \pm 0.001}$ | $0.859 \pm 0.046$ | $0.664 \pm 0.009$ | $0.946 \pm 0.051$ | $0.746 \pm 0.029$ |
| 6 offsprings | $0.597 \pm 0.183$ | $0.468 \pm 0.119$ | $0.569 \pm 0.132$ | $0.987 \pm 0.004$ | $0.899 \pm 0.053$ | $\mathbf{0.710 \pm 0.058}$ | $0.891 \pm 0.056$ | $0.822 \pm 0.012$ |
| | ExLLM (DeepSeek-V3.1) | | | | ExLLM(Qwen3-Max) | | | |
| 1 offsprings | $0.803 \pm 0.056$ | $0.629 \pm 0.052$ | $0.750 \pm 0.047$ | $0.869 \pm 0.015$ | $0.167 \pm 0.019$ | $0.240 \pm 0.125$ | $0.282 \pm 0.045$ | $0.916 \pm 0.004$ |
| 2 offsprings | $0.792 \pm 0.147$ | $0.618 \pm 0.103$ | $0.818 \pm 0.035$ | $0.881 \pm 0.021$ | $\mathbf{0.650 \pm 0.136}$ | $\mathbf{0.523 \pm 0.088}$ | $0.384 \pm 0.059$ | $\mathbf{0.920 \pm 0.003}$ |
| 3 offsprings | $0.784 \pm 0.116$ | $0.577 \pm 0.093$ | $\mathbf{0.871 \pm 0.025}$ | $0.929 \pm 0.006$ | $0.500 \pm 0.075$ | $0.406 \pm 0.031$ | $0.459 \pm 0.061$ | $0.871 \pm 0.044$ |
| 4 offsprings | $0.835 \pm 0.032$ | $\mathbf{0.675 \pm 0.005}$ | $0.851 \pm 0.051$ | $0.929 \pm 0.006$ | $0.575 \pm 0.041$ | $0.343 \pm 0.132$ | $0.454 \pm 0.081$ | $0.850 \pm 0.059$ |
| 5 offsprings | $\mathbf{0.841 \pm 0.026}$ | $0.672 \pm 0.019$ | $0.774 \pm 0.048$ | $0.933 \pm 0.026$ | $0.558 \pm 0.032$ | $0.459 \pm 0.027$ | $\mathbf{0.479 \pm 0.009}$ | $0.849 \pm 0.025$ |
| 6 offsprings | $0.654 \pm 0.111$ | $0.496 \pm 0.092$ | $0.773 \pm 0.016$ | $\mathbf{0.938 \pm 0.018}$ | $0.483 \pm 0.065$ | $0.318 \pm 0.108$ | $0.401 \pm 0.055$ | $0.913 \pm 0.044$ |

Table 9: Experiments of k-offspring per prompt. The objective is maximizing JNK3. We add two open-source LLM for more comprehensive ablation studies here.

$k$**-offspring trade-offs** Figure 7 shows the results of varying $k$ without using experience. For the five-objective plot, we subtract 3.5 from both AUC and $F$ before plotting to improve readability. Increasing $k$ improves exploratory breadth, especially in the single-objective JNK3 task, yet overly large $k$ tends to over-explore locally and limits global search under a fixed budget. Across tasks, $k$=2 yields the largest and most stable gains in our sweep; $k$=3 is less stable, and larger $k$ becomes increasingly unstable or degrades improvement. From Figure 7, Table 9, and Table 10, we observe a consistent trend as the number of offspring $k$(i.e., the number of molecules generated per LLM query) increases. Performance metrics such as AUC and Top-F typically peak at $k = 2$ or $3$ indicating that generating multiple candidates per call improves both search quality and stability. In most settings, uniqueness also increases once $k > 1$, showing that the $k$-offspring strategy broadens the exploration space and yields more diverse solutions. Given these consistent gains over the standard $k = 1$ setting used in prior LLM-as-optimizer methods, we adopt $k = 2$ as our default: it provides the most stable improvements while also reducing the total number of LLM queries and overall runtime.

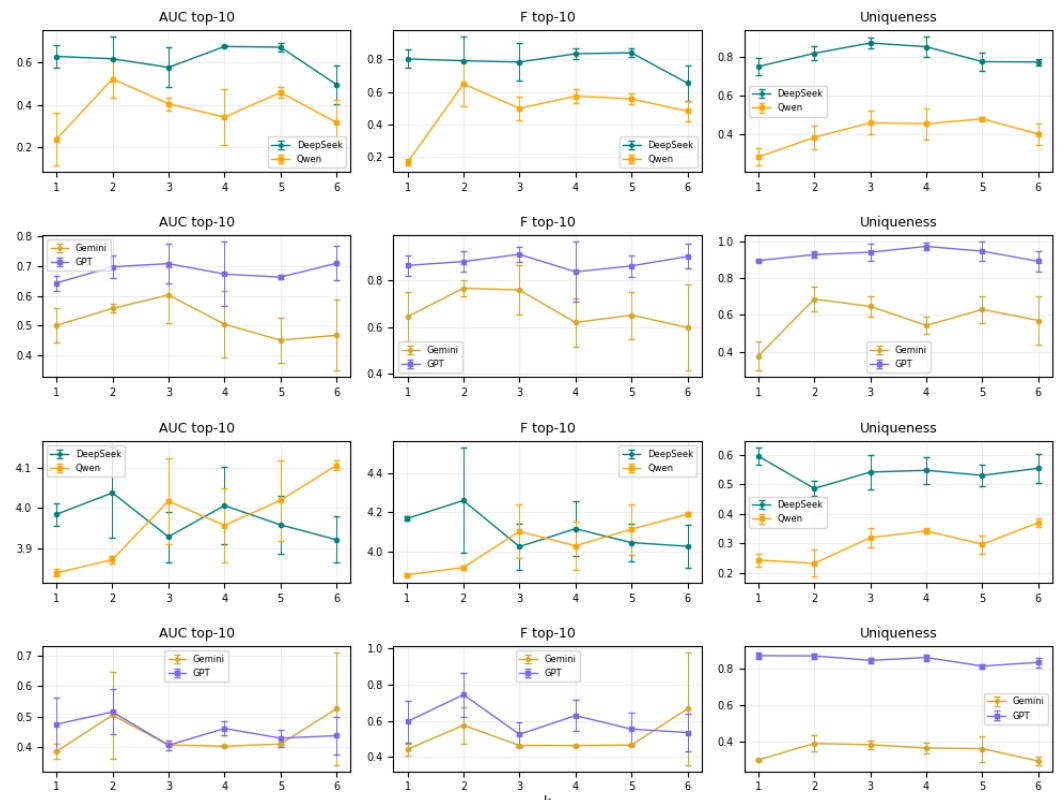

Figure 7: $k$-offspring ablation studies. Top two rows: single-objective optimization (JNK3). Bottom two rows: five-objective optimization. The most consistent improvement on results and exploration occurs at about $k$=2 and 3, while higher $k$ may lead to local over-exploration.

| Method | Top10 F | Top10 AUC | Uniqueness | Validity | Top10 F | Top10 AUC | Uniqueness | Validity |
|---|---|---|---|---|---|---|---|---|
| | | ExLLM (Gemini-2.5-Flash) | | | | ExLLM (GPT-4o) | | |
| 1 offspring | $3.943\pm0.032$ | $3.886\pm0.025$ | $0.300\pm0.009$ | $0.990\pm0.002$ | $4.097\pm0.116$ | $3.975\pm0.088$ | $\mathbf{0.872\pm0.020}$ | $\mathbf{0.920\pm0.012}$ |
| 2 offsprings | $4.076\pm0.100$ | $4.005\pm0.143$ | $\mathbf{0.390\pm0.045}$ | $\mathbf{0.992\pm0.001}$ | $\mathbf{4.245\pm0.121}$ | $\mathbf{4.015\pm0.074}$ | $0.870\pm0.014$ | $0.913\pm0.007$ |
| 3 offsprings | $3.964\pm0.003$ | $3.907\pm0.003$ | $0.383\pm0.026$ | $0.990\pm0.000$ | $4.026\pm0.068$ | $3.906\pm0.016$ | $0.846\pm0.015$ | $0.917\pm0.012$ |
| 4 offsprings | $3.934\pm0.004$ | $3.903\pm0.003$ | $0.366\pm0.030$ | $0.983\pm0.010$ | $4.129\pm0.085$ | $3.961\pm0.024$ | $0.862\pm0.018$ | $0.906\pm0.005$ |
| 5 offsprings | $3.967\pm0.007$ | $3.910\pm0.002$ | $0.362\pm0.071$ | $0.989\pm0.001$ | $4.055\pm0.094$ | $3.929\pm0.028$ | $0.814\pm0.012$ | $0.890\pm0.019$ |
| 6 offsprings | $\mathbf{4.167\pm0.314}$ | $\mathbf{4.025\pm0.185}$ | $0.294\pm0.024$ | $0.987\pm0.005$ | $4.036\pm0.104$ | $3.937\pm0.061$ | $0.835\pm0.026$ | $0.862\pm0.029$ |
| | | ExLLM (DeepSeek-V3.1) | | | | ExLLM(Qwen3-Max) | | |
| 1 offsprings | $4.168\pm0.012$ | $3.984\pm0.027$ | $\mathbf{0.595\pm0.028}$ | $0.971\pm0.002$ | $3.881\pm0.009$ | $3.839\pm0.009$ | $0.244\pm0.023$ | $\mathbf{0.980\pm0.008}$ |
| 2 offsprings | $\mathbf{4.259\pm0.268}$ | $\mathbf{4.037\pm0.111}$ | $0.486\pm0.026$ | $\mathbf{0.974\pm0.009}$ | $3.919\pm0.009$ | $3.872\pm0.008$ | $0.233\pm0.045$ | $0.978\pm0.009$ |
| 3 offsprings | $4.025\pm0.119$ | $3.928\pm0.063$ | $0.541\pm0.057$ | $0.970\pm0.013$ | $4.103\pm0.135$ | $4.017\pm0.107$ | $0.319\pm0.034$ | $0.948\pm0.028$ |
| 4 offsprings | $4.116\pm0.139$ | $4.006\pm0.096$ | $0.547\pm0.046$ | $0.958\pm0.017$ | $4.029\pm0.125$ | $3.957\pm0.093$ | $0.342\pm0.010$ | $0.940\pm0.022$ |
| 5 offsprings | $4.045\pm0.096$ | $3.958\pm0.073$ | $0.529\pm0.363$ | $0.968\pm0.002$ | $4.113\pm0.129$ | $4.019\pm0.099$ | $0.297\pm0.031$ | $0.933\pm0.024$ |
| 6 offsprings | $4.026\pm0.109$ | $3.921\pm0.057$ | $0.554\pm0.048$ | $0.971\pm0.010$ | $\mathbf{4.189\pm0.010}$ | $\mathbf{4.106\pm0.013}$ | $\mathbf{0.370\pm0.015}$ | $0.902\pm0.013$ |

Table 10: Experiments of k-offspring per prompt. The objective is optimizting 5 objectives same as in table 2. We add two open-source LLM for more comprehensive ablation studies here.

### 6.3.2 EXPERIENCE INJECTION

Our evolving experience is lightweight and general, yielding steady performance gains with only a modest impact on diversity. That said, in molecular optimization which has large discrete spaces, $P_{exp}$ should be properly controlled to prevent over-conditioning the search and restricting exploration.

### 6.3.3 HYBRID SELECTOR

| $P_{exp}$ | Top10 F | AUC-Top10 | Hypervolume | Uniqueness | Validity | Diversity |
|---|---|---|---|---|---|---|
| 0.0 | $4.245 \pm 0.121$ | $4.015 \pm 0.074$ | $0.850 \pm 0.224$ | $0.870 \pm 0.014$ | $0.900 \pm 0.003$ | $0.556 \pm 0.106$ |
| 0.3 | $4.260 \pm 0.145$ | $4.004 \pm 0.077$ | $0.881 \pm 0.201$ | $0.879 \pm 0.014$ | $0.905 \pm 0.010$ | $0.509 \pm 0.057$ |
| 0.5 | $\mathbf{4.301 \pm 0.164}$ | $\mathbf{4.116 \pm 0.040}$ | $\mathbf{0.905 \pm 0.200}$ | $0.872 \pm 0.015$ | $0.908 \pm 0.019$ | $0.494 \pm 0.032$ |
| 0.7 | $3.986 \pm 0.032$ | $3.892 \pm 0.007$ | $0.555 \pm 0.188$ | $0.843 \pm 0.016$ | $0.930 \pm 0.015$ | $0.586 \pm 0.070$ |
| 0.9 | $4.030 \pm 0.098$ | $3.923 \pm 0.051$ | $0.571 \pm 0.228$ | $0.856 \pm 0.020$ | $0.926 \pm 0.002$ | $0.496 \pm 0.021$ |

Table 11: The ablation study of $P_{exp}$ shows that incorporating experience can notably improve performance and convergence, but it must be properly controlled to avoid restricting the exploration direction.

| Selector | Top1 F | Top10 F | AUC-Top10 | Hypervolume | Uniqueness | Validity | Diversity |
|---|---|---|---|---|---|---|---|
| Only Pareto front-based | $4.170 \pm 0.055$ | $4.092 \pm 0.042$ | $3.957 \pm 0.037$ | $0.857 \pm 0.061$ | $0.870 \pm 0.025$ | $0.909 \pm 0.012$ | $0.667 \pm 0.049$ |
| Only Fitness-based | $4.163 \pm 0.190$ | $4.055 \pm 0.094$ | $3.938 \pm 0.048$ | $0.680 \pm 0.274$ | $0.844 \pm 0.015$ | $0.916 \pm 0.005$ | $0.517 \pm 0.031$ |
| Hybrid (pareto front and fitness) selector | $4.336 \pm 0.246$ | $4.300 \pm 0.164$ | $4.116 \pm 0.040$ | $0.905 \pm 0.200$ | $0.872 \pm 0.015$ | $0.908 \pm 0.019$ | $0.494 \pm 0.032$ |

Table 12: The ablation study of pareto front-based select and fitness-based selector.

The ablation study shows that neither pure Pareto-based selection nor pure fitness-based selection performs well in isolation, while the hybrid selector achieves the most balanced and consistently strong performance. The hybrid strategy attains the highest Top-1 F, Top-10 F, AUC-Top10, and hypervolume scores, indicating more stable optimization and better convergence across objectives. In contrast, fitness-only selection exhibits instability and reduced hypervolume due to over-exploitation, whereas Pareto-only selection maintains higher diversity but converges less effectively. By combining global exploration from the Pareto front with the local refinement provided by fitness-based ranking, the hybrid selector offers a more reliable optimization trajectory and produces the best overall results in multi-objective molecular optimization.

In addition, using both components together is necessary because each of the single selectors systematically removes valuable molecules. If only Pareto-based selection is applied, any molecule that is dominated, even slightly, will be discarded, even when its structure is very different from the dominating molecule and contains meaningful information that should be preserved for exploration. In practice, this means that two molecules with similarly high objective values may end up having very different chemistries, yet only one will survive under a strict Pareto filter, causing the search process to lose structurally diverse and informative candidates. Conversely, if only fitness-based selection is used, the algorithm tends to keep only the highest-scoring individuals and gradually collapses onto a narrow region of chemical space. Molecules that are not dominated in the multi-objective sense, but happen to have slightly lower fitness, are removed despite being valuable for maintaining coverage of the objective distribution. Such molecules help prevent mode collapse and preserve alternative optimization pathways. By combining both selectors, the hybrid strategy retains globally competitive and diverse candidates while still promoting effective local refinement, leading to a more stable and informative population across the entire optimization trajectory.

### 6.3.4 DIFFERENT LLM IN EXLLM

| LLM in ExLLM | Top1 F | Top10 F | AUC-Top10 | Hypervolume | Uniqueness | Validity | Diversity |
|---|---|---|---|---|---|---|---|
| GPT-4o | $\mathbf{4.336 \pm 0.246}$ | $\mathbf{4.300 \pm 0.164}$ | $\mathbf{4.116 \pm 0.040}$ | $\mathbf{0.905 \pm 0.200}$ | $\mathbf{0.872 \pm 0.015}$ | $0.908 \pm 0.019$ | $0.494 \pm 0.032$ |
| Gemini-2.5-Flash | $4.208 \pm 0.011$ | $4.191 \pm 0.009$ | $4.070 \pm 0.026$ | $0.750 \pm 0.007$ | $0.615 \pm 0.035$ | $0.950 \pm 0.006$ | $\mathbf{0.521 \pm 0.013}$ |
| Deepseek-V3.1 | $4.175 \pm 0.039$ | $4.152 \pm 0.040$ | $3.994 \pm 0.039$ | $0.763 \pm 0.100$ | $0.544 \pm 0.072$ | $0.959 \pm 0.009$ | $0.496 \pm 0.020$ |
| Qwen3-Max | $4.075 \pm 0.123$ | $4.056 \pm 0.115$ | $3.968 \pm 0.085$ | $0.587 \pm 0.215$ | $0.316 \pm 0.041$ | $\mathbf{0.974 \pm 0.011}$ | $0.500 \pm 0.045$ |

Table 13: The ablation study of using different LLM in ExLLM, including both proprietary LLM (GPT and Gemini) and open-source LLM (Qwen and Deepseek).

Across all alternative LLMs, ExLLM maintains strong overall performance on the five-objective molecular optimization task, demonstrating that the framework is not tied to a particular backbone. GPT-4o delivers the best results across almost all metrics—including Top-1/Top-10 fitness, AUC-Top10, and hypervolume—indicating that stronger reasoning and generation abilities directly translate into higher optimization quality. Gemini-2.5-Flash performs competitively and achieves the highest

diversity, while the open-source models (DeepSeek-V3.1 and Qwen3-Max) also produce valid and diverse candidates despite their architectural and pretraining differences. Notably, validity remains consistently high for all models, and uniqueness/diversity stay within a strong range. These results show that although model capacity affects peak performance, ExLLM remains robust and effective regardless of whether the backbone is proprietary or open-source.

### 6.3.5 INCREASING THE NUMBER OF OBJECTIVES

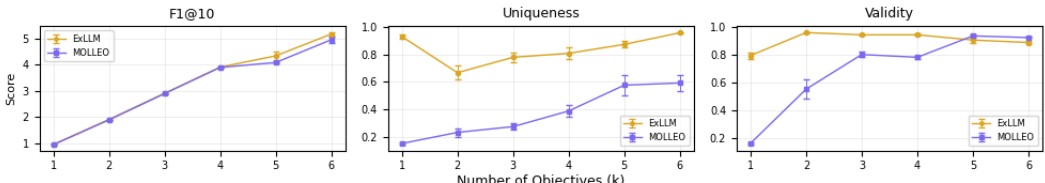

Figure 8: Experiment of different number of objectives.

| Metric | 1 Objective | | 2 Objectives | | 3 Objectives | | 4 Objectives | | 5 Objectives | | 6 Objectives | |
|---|---|---|---|---|---|---|---|---|---|---|---|---|
| | ExLLM | MOLLEO | ExLLM | MOLLEO | ExLLM | MOLLEO | ExLLM | MOLLEO | ExLLM | MOLLEO | ExLLM | MOLLEO |
| Top1 F | **0.948** | 0.941 | **1.901** | 1.887 | **2.901** | 2.891 | **3.901** | 3.890 | **4.336** | 4.098 | **5.183** | 4.964 |
| Top10 F | **0.948** | 0.936 | **1.901** | 1.882 | **2.901** | 2.886 | **3.901** | 3.887 | **4.300** | 4.076 | **5.164** | 4.948 |
| Uniqueness | **0.929** | 0.150 | **0.666** | 0.231 | **0.778** | 0.273 | **0.807** | 0.387 | **0.872** | 0.575 | **0.957** | 0.591 |
| Validity | **0.796** | 0.159 | **0.962** | 0.552 | **0.946** | 0.803 | **0.946** | 0.783 | 0.908 | **0.938** | 0.890 | **0.926** |
| Diversity | 0.538 | **0.865** | 0.450 | **0.646** | 0.510 | **0.627** | 0.375 | **0.614** | 0.494 | **0.573** | 0.529 | **0.611** |

Table 14: Unconstrained molecular design results with 1 to 6 objectives. The sixth objective is BBBP.

We conduct experiments using random initialization across scenarios with one to six objectives, starting with only QED↑ to QED↑ + SA↓ + DRD2↓ + GSK3$\beta$↓ + JNK3↑ + BBBP↑ (Blood-Brain Barrier Permeability) as shown in Figure 8 and Table 14. BBBP (Blood-Brain Barrier Permeability) as a sixth objective, as it is a more complex and less predictable property with limited domain knowledge. As the number of objectives increases, the performance gap between ExLLM and MOLLEO widens, particularly when optimizing more than four objectives, showing the gains of increasing complexity of the task. In MOLLEO uniqueness and validity tend to degrade significantly when optimizing fewer objectives, this indicates the exploration ability is seriously affected. ExLLM consistently achieves high uniqueness and validity, showing its stable exploration ability across different task complexity. Notably, ExLLM achieves consistently higher Top-1 and Top-10 fitness scores across all scenarios, representing a greater search efficiency. Overall, these results highlight the scalability and stability of ExLLM across varying task complexities, confirming its superior ability to maintain exploration ability and optimization efficiency in multi-objective molecular design.

### 6.3.6 PROMOTING A CONSTRAINT TO AN OBJECTIVE

| Experiment | GPT-4o(5 candidates) | GPT-4o(20 candidates) | ExLLM first generation | ExLLM with similarity in prompt | ExLLM with similarity in selection and prompt |
|---|---|---|---|---|---|
| QED | 0.173 | 0.513 | 0.543 | 0.729 | **0.929** |
| LogP | 0.224 | 0.502 | 0.472 | 0.512 | **0.613** |
| Donor | 0.403 | 0.873 | 0.804 | 0.504 | **0.949** |
| QED+LogP | 0.069 | 0.185 | 0.218 | 0.456 | **0.504** |
| QED+Donor | 0.117 | 0.356 | 0.401 | 0.528 | **0.699** |
| LogP+Donor | 0.168 | 0.404 | 0.351 | 0.411 | **0.505** |
| QED+Logp+Donor | 0.032 | 0.115 | 0.136 | 0.265 | **0.324** |

Table 15: Target molecular optimization with similarity threshold 0.4.

To evaluate the importance of promoting critical constraints into optimization objectives, we conducted a target molecular optimization experiment. For each task, a reference molecule was given, and the model was required to modify it to improve or decrease specific properties (e.g., QED, LogP, or Donor count) while maintaining a structural similarity threshold of at least 0.4. Each experiment involved 150 molecular edits, with ExLLM running for only 20 generations and a population size of 20 without experience. The reported numbers in Table 15 represent the success rate among the final 20 generated molecules that satisfied both the similarity and target property criteria.

As baselines, GPT-4o was asked to directly generate 5 or 20 molecular candidates, and the success rates from ExLLM's first generation were also included. We further compared two variants of ExLLM: one that includes the similarity constraint only in the prompt and another that integrates similarity both in the prompt and in the multi-objective (MO) selection step.

Across all target combinations, incorporating the similarity constraint directly into the MO selection consistently and substantially improves success rates (e.g., QED: $0.729 \rightarrow 0.929$; QED + Donor: $0.528 \rightarrow 0.699$). This demonstrates that when a constraint plays a critical role, such as structural similarity, it is beneficial to promote it into the optimization objectives rather than treating it merely as a filtering condition. Such integration leads to more stable and efficient optimization performance.

## 6.4 ADDITIONAL RESULTS IN PMO

| Task type | Objective(↑) | LICO | Chemma 2B | ExLLM (Ours) | LICO (PMO-1K) | ExLLM (PMO-1K) |
|---|---|---|---|---|---|---|
| Property optimization | QED | $0.936 \pm 0.001$ | $0.941 \pm 0.000$ | $\mathbf{0.943 \pm 0.000}$ | $\mathbf{0.925 \pm 0.005}$ | $0.896 \pm 0.000$ |
| | JNK3 | $0.731 \pm 0.037$ | $\mathbf{0.891 \pm 0.032}$ | $0.732 \pm 0.078$ | $\mathbf{0.336 \pm 0.051}$ | $0.308 \pm 0.066$ |
| | DRD2 | $0.928 \pm 0.018$ | $0.972 \pm 0.006$ | $\mathbf{0.980 \pm 0.003}$ | $\mathbf{0.859 \pm 0.066}$ | $0.797 \pm 0.033$ |
| | GSK3$\beta$ | $0.876 \pm 0.045$ | $\mathbf{0.928 \pm 0.021}$ | $0.818 \pm 0.050$ | $\mathbf{0.617 \pm 0.063}$ | $0.449 \pm 0.027$ |
| Name-based optimization | mestranol_similarity | $0.614 \pm 0.064$ | $0.926 \pm 0.023$ | $\mathbf{0.980 \pm 0.005}$ | $0.423 \pm 0.016$ | $\mathbf{0.798 \pm 0.048}$ |
| | albuterol_similarity | $0.885 \pm 0.019$ | $0.951 \pm 0.009$ | $\mathbf{0.989 \pm 0.000}$ | $0.656 \pm 0.125$ | $\mathbf{0.886 \pm 0.001}$ |
| | thiothixene_rediscovery | $0.514 \pm 0.037$ | $0.698 \pm 0.121$ | $\mathbf{0.910 \pm 0.004}$ | $0.343 \pm 0.035$ | $\mathbf{0.718 \pm 0.045}$ |
| | celecoxib_rediscovery | $0.664 \pm 0.122$ | $\mathbf{0.920 \pm 0.011}$ | $0.891 \pm 0.033$ | $0.447 \pm 0.073$ | $\mathbf{0.634 \pm 0.048}$ |
| | troglitazone_rediscovery | $0.380 \pm 0.026$ | $\mathbf{0.824 \pm 0.049}$ | $0.726 \pm 0.111$ | $0.292 \pm 0.028$ | $\mathbf{0.348 \pm 0.026}$ |
| | perindopril_mpo | $0.557 \pm 0.028$ | $0.711 \pm 0.062$ | $\mathbf{0.797 \pm 0.016}$ | $0.473 \pm 0.009$ | $\mathbf{0.660 \pm 0.040}$ |
| | ranolazine_mpo | $0.774 \pm 0.008$ | $\mathbf{0.868 \pm 0.015}$ | $0.855 \pm 0.021$ | $\mathbf{0.687 \pm 0.029}$ | $0.662 \pm 0.011$ |
| | sitagliptin_mpo | $0.567 \pm 0.034$ | $\mathbf{0.613 \pm 0.018}$ | $0.555 \pm 0.048$ | $\mathbf{0.315 \pm 0.097}$ | $0.290 \pm 0.025$ |
| | amlodipine_mpo | $0.679 \pm 0.027$ | $0.766 \pm 0.107$ | $\mathbf{0.874 \pm 0.010}$ | $0.541 \pm 0.026$ | $\mathbf{0.784 \pm 0.004}$ |
| | fexofenadine_mpo | $0.772 \pm 0.023$ | $0.931 \pm 0.014$ | $\mathbf{0.984 \pm 0.006}$ | $0.700 \pm 0.023$ | $\mathbf{0.893 \pm 0.006}$ |
| | osimertinib_mpo | $0.820 \pm 0.012$ | $0.879 \pm 0.016$ | $\mathbf{0.902 \pm 0.018}$ | $0.759 \pm 0.008$ | $\mathbf{0.789 \pm 0.013}$ |
| | zaleplon_mpo | $0.515 \pm 0.017$ | $0.608 \pm 0.055$ | $\mathbf{0.723 \pm 0.007}$ | $0.404 \pm 0.022$ | $\mathbf{0.622 \pm 0.010}$ |
| | median1 | $0.291 \pm 0.016$ | $0.382 \pm 0.022$ | $\mathbf{0.384 \pm 0.007}$ | $0.217 \pm 0.019$ | $\mathbf{0.329 \pm 0.005}$ |
| | median2 | $0.280 \pm 0.019$ | $0.366 \pm 0.018$ | $\mathbf{0.475 \pm 0.002}$ | $0.193 \pm 0.009$ | $\mathbf{0.370 \pm 0.014}$ |
| Structure-based optimization | isomers_c7h8n2o2 | $0.939 \pm 0.022$ | $0.947 \pm 0.009$ | $\mathbf{0.984 \pm 0.001}$ | $0.779 \pm 0.099$ | $\mathbf{0.842 \pm 0.013}$ |
| | isomers_c9h10n2o2pf2cl | $0.819 \pm 0.039$ | $0.914 \pm 0.017$ | $\mathbf{0.961 \pm 0.028}$ | $0.672 \pm 0.075$ | $\mathbf{0.746 \pm 0.018}$ |
| | deco_hop | $0.619 \pm 0.015$ | $0.831 \pm 0.123$ | $\mathbf{0.956 \pm 0.014}$ | $0.596 \pm 0.010$ | $\mathbf{0.855 \pm 0.015}$ |
| | scaffold_hop | $0.547 \pm 0.026$ | $0.669 \pm 0.110$ | $\mathbf{0.916 \pm 0.127}$ | $0.480 \pm 0.008$ | $\mathbf{0.786 \pm 0.089}$ |
| | valsartan_smarts | $0.000 \pm 0.000$ | $0.000 \pm 0.000$ | $\mathbf{0.831 \pm 0.043}$ | $0.000 \pm 0.000$ | $\mathbf{0.069 \pm 0.076}$ |
| | **Total (↑)** | 14.708 | 17.534 | **19.165** | 11.71 | **14.533** |
| | **Rank (↓)** | 6 | 3 | **1** | 2 | **1** |

Table 16: Top-10 AUC on the PMO benchmark and the PMO benchmark with a 1000-oracle budget, including additional baseline models. ExLLM still performs the best. The 7.3% improvement by ExLLM remains unchanged compared the second best model MOLLEO in table 3. On the PMO benchmark under a 1000-oracle budget, LICO achieved state-of-the-art performance (Nguyen & Grover, 2024), while our ExLLM improves the total score by 24.1% over LICO, demonstrating substantially better sample efficiency.

In Table 16, we include additional LLM-based molecular optimization models on the PMO benchmark for comparison. The first is Chemma (Bedrosian et al.), which finetunes large language models on large-scale molecular corpora; the second is LICO, which relies on in-context learning augmented with domain-specific embedding layers. On the standard PMO benchmark, our ExLLM continues to achieve the best overall performance, maintaining a substantial lead over all other models. The PMO-1K setting, introduced by Nguyen & Grover (2024), is a low-budget variant of PMO with an oracle budget of only 1000 evaluations. This setting is designed to stress-test sample efficiency and to compare how effectively different models perform under highly constrained evaluation budgets. ExLLM obtains a total score of 14.533, marking a +24.1% improvement over LICO and establishing a new state-of-the-art under the strict 1000-oracle budget.

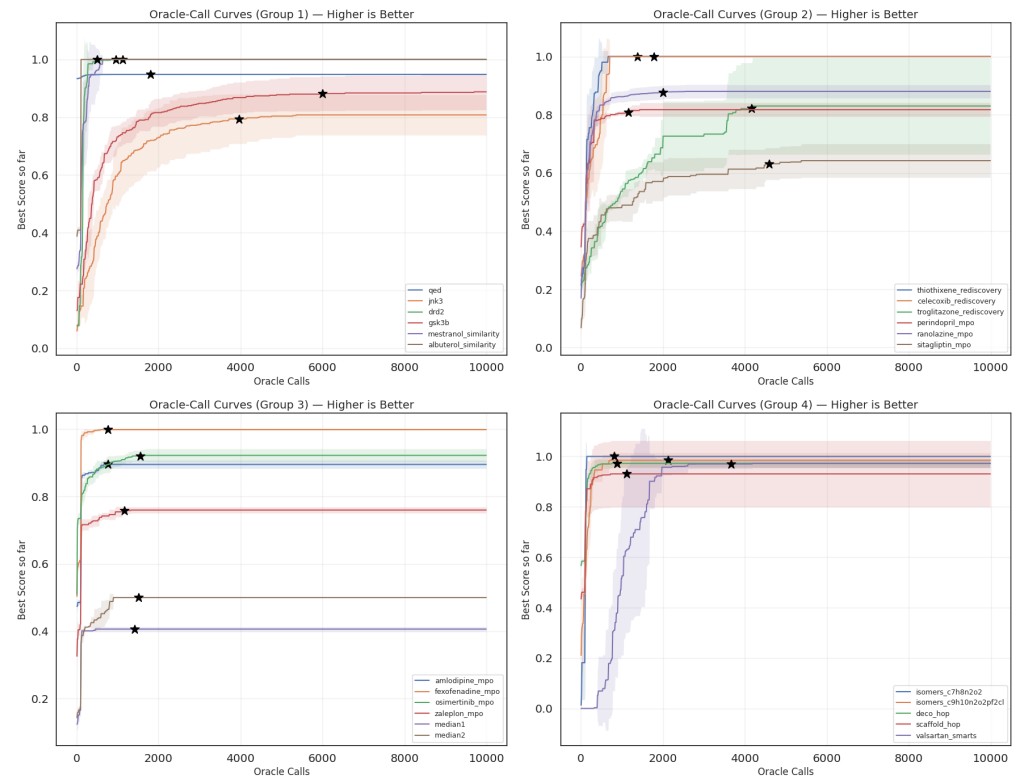

Figure 9: The convergence curve of ExLLM on the PMO benchmark, with the convergence point indicated by a star averaged over 5 seeds.

## 6.5 EFFICIENCY COMPARISON

| Method | ExLLM (GPT-4o) | MOLLEO (GPT-4o) | Graph GA | Gp-BO | Genetic-GFN | MARS | JT-VAE | DyMol | REINVENT |
|---|---|---|---|---|---|---|---|---|---|
| Running time(hours) | $0.393 \pm 0.114$ | $6.029 \pm 1.281$ | $1.041 \pm 0.156$ | $0.683 \pm 0.100$ | $0.453 \pm 0.071$ | $0.692 \pm 0.129$ | $3.082 \pm 0.362$ | $0.289 \pm 0.037$ | $0.522 \pm 0.088$ |

Table 17: Comparison of runnning time on the five objective molecular optimization.

| Method | Total input tokens | Total output tokens | Input tokens for experience | Output tokens for experience | Total cost(US dollar) | Running time(h) |
|---|---|---|---|---|---|---|
| MOLLEO(GPT-4o) | $2.991 \pm 0.246$ | $2.191 \pm 0.201$ | N/A | N/A | $19.126 \pm 1.698$ | $6.029 \pm 1.281$ |
| ExLLM(GPT-4o) | $2.857 \pm 0.338$ | $0.204 \pm 0.020$ | $0.253 \pm 0.031$ | $0.031 \pm 0.005$ | $6.938 \pm 0.796$ | $0.393 \pm 0.114$ |
| ExLLM(Gemini-2.5-Flash) | $3.799 \pm 0.456$ | $0.279 \pm 0.036$ | $0.365 \pm 0.029$ | $0.027 \pm 0.004$ | $1.102 \pm 0.136$ | $0.495 \pm 0.133$ |
| ExLLM(DeepSeek-V3.1) | $4.145 \pm 0.364$ | $0.283 \pm 0.024$ | $0.372 \pm 0.033$ | $0.037 \pm 0.004$ | $2.816 \pm 0.246$ | $1.703 \pm 0.341$ |
| ExLLM(Qwen3-Max) | $4.316 \pm 1.687$ | $0.285 \pm 0.110$ | $0.408 \pm 0.160$ | $0.051 \pm 0.020$ | $4.604 \pm 0.389$ | $1.575 \pm 0.796$ |

Table 18: Running cost comparison, the tokens are all in millions.

We record the running time of each model presented in Table 2, a five objective molecular optimization with 5000 oracle calls limit. With the exactly same API model, ExLLM consumes much less API costs compared to MOLLEO. Apart from that, without early stopping, ExLLM is more than even 15x faster than MOLLEO in run time to achieve better results, as shown in Table 17 and Table 2. The running time is also competitive to other methods that are not based on LLMs. We also report the costs of running ExLLM with different proprietary and open-source LLMs on the five-objective molecular optimization task. The evolving-experience module contributes only a small fraction of the total tokens, leading to only a slight increase in overall cost while demonstrating that the experience mechanism is highly efficient. Pricing details. Costs are computed using the following

rates (per million tokens): GPT-4o-2024-05-13 — $2.00 for input and $6.00 for output tokens; Gemini-2.5-flash-nothinking — $0.18 for input and $1.50 for output tokens; DeepSeek-V3.1 — $0.564 for input and $1.691 for output tokens; Qwen3-Max — $0.845 for input and $3.382 for output tokens. Pricing data were accessed on 25 November 2025.

## 6.6 PER-OBJECTIVE IMPROVEMENT AND PARETO FRONT COVERAGE

Table 19 provides a quantitative view of per-objective improvement relative to the randomly initialized population. Across all five objectives, ExLLM shifts the distribution toward substantially better regions: SA, DRD2, and GSK3$\beta$ decrease notably, while QED and JNK3 increase by large margins. Compared to the broad and noisy initial populations, the top-100 ExLLM molecules (aggregated over 5 seeds) exhibit lower variance, tighter ranges, and consistently stronger minima and maxima across objectives. These results confirm that ExLLM improves each objective individually rather than relying on easier targets, aligning with the balanced Pareto-front coverage observed in Figures 10 and 11.

| Objective | Initial Mean ± Std | Min | Max | ExLLM Mean ± Std | Min | Max |
|---|---|---|---|---|---|---|
| SA ↓ | 3.021 ± 0.792 | 1.652 | 5.074 | 2.037 ± 0.157 | 1.533 | 2.441 |
| DRD2 ↓ | 0.009 ± 0.033 | 0.000 | 0.283 | 0.001 ± 0.001 | 0.000 | 0.007 |
| GSK3$\beta$ ↓ | 0.026 ± 0.040 | 0.000 | 0.270 | 0.034 ± 0.048 | 0.000 | 0.200 |
| QED ↑ | 0.730 ± 0.143 | 0.303 | 0.936 | 0.879 ± 0.034 | 0.756 | 0.944 |
| JNK3 ↑ | 0.015 ± 0.022 | 0.000 | 0.130 | 0.490 ± 0.234 | 0.110 | 0.950 |

Table 19: Per-objective improvement of the five objective optimization, the results of ExLLM are computed from top 100 molecules with 5 seeds.

---

**Prompt Example for Multi-Objective Molecular Optimization**

**Task Description.** This task proposes improved molecular candidates according to multiple optimization objectives.

**Output Format.** Each generated candidate must begin with `<candidate>` and end with `</candidate>` in SMILES format. An example output is:

`<candidate>c1ccccc1</candidate>`

**Mutation Instruction.** Example operations include: 1. modifying functional groups; 2. replacing atoms or bonds; 3. adding or removing small substituents; 4. ring modifications; 5. stereochemistry changes; 6. property-specific structural adjustments.

**Crossover Instruction.** (No crossover instruction provided for this task.)

**Additional Requirements.** Generated molecules must be chemically valid.

**Objectives.**

`qed.` The QED (Quantitative Estimate of Drug-likeness) quantifies how "drug-like" a molecule is based on molecular weight, solubility, and hydrogen-bonding properties. Small, balanced functional groups typically increase QED, whereas large or highly polar substituents often decrease it.

`sa.` SA measures synthetic accessibility. Simplifying complex rings or functional groups reduces SA (easier synthesis), while introducing complex or heavily constrained structures increases SA.

`drd2.` The DRD2 objective scores predicted activity against the dopamine receptor D2. Adding hydroxyl, halogen, or other activity-enhancing substituents to aromatic systems may increase affinity, whereas removing aromaticity or adding bulky groups near key binding regions often lowers DRD2 activity.

`gsk3b.` GSK3$\beta$ is a kinase implicated in bipolar disorder. The objective is computed by a random-forest classifier using ECFP6 fingerprints.

`jnk3.` JNK3 is a kinase targeted in neuroprotective drug design. Introducing electronegative or small polar groups can enhance predicted activity, while removing polarity or adding bulky substituents tends to reduce activity.

---

## 6.7 PROMPT EXAMPLES

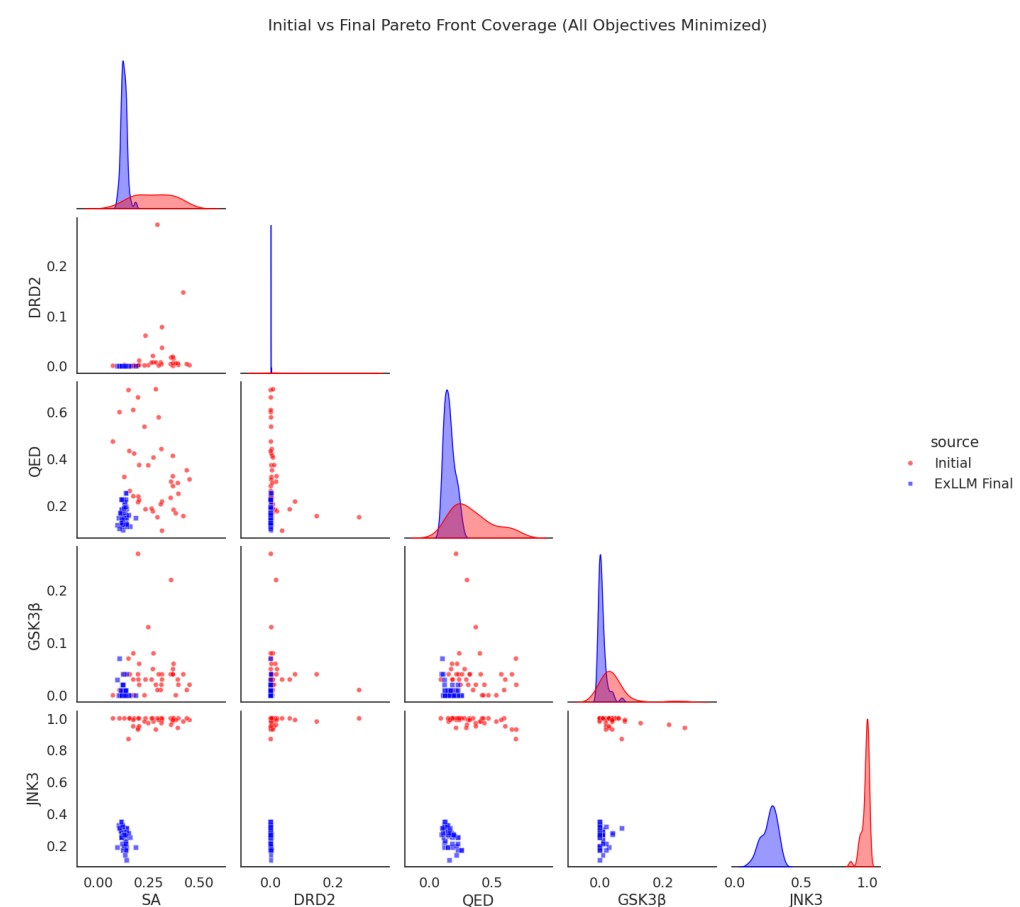

Figure 10: The pairwise scatter matrix for the 5-objective molecular design task shows that ExLLM produces solutions that improve substantially across all objectives relative to the randomly initialized population. The points do not collapse onto any single objective; instead, they occupy a broad, well-distributed region of the Pareto front, indicating balanced multi-objective progress. All objectives are normalized and converted into minimization form.

In this section, we provide prompt examples for the extended-domain experiments, including molecular optimization, circle packing, and peptide design. Each prompt follows a unified template consisting of five components: task description, output format, mutation/crossover instructions, additional requirements, and objective definitions. Filling in these components is straightforward and does not require complex prompt engineering. The task description simply describe the task. The output format offers an explicit example to ensure that the LLM produces results in a consistent and correctly parsed structure. The mutation and crossover instructions are optional; simple high-level suggestions are sufficient, and the system performs well even when these fields are left blank. Additional requirements serve to impose any necessary constraints on valid outputs. For the objectives, we directly use the standard definitions provided by each domain's official evaluation or oracle.

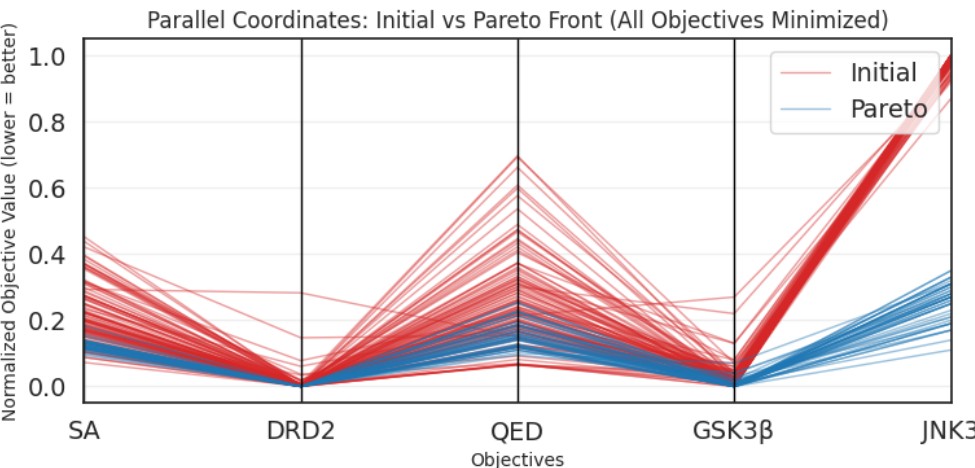

Figure 11: The parallel-coordinate visualization further confirms that ExLLM consistently improves the randomly initialized population on all five objectives. Each objective shows clear, uniform improvement, demonstrating the reliability and stability of ExLLM's multi-objective optimization. All objectives are normalized and converted into minimization form.

---

**Prompt Example for Circle Packing Optimization**

**Task Description.** This task proposes improved solutions for the `circle_packing` problem, where 26 circles must be placed within a unit square.

**Output Format.** Each generated candidate must begin with `<candidate>` and end with `</candidate>`. An example output is:

```
<candidate>
centers = np.array([
    [0.50, 0.50], [0.30, 0.50], ...
])
radii = np.array([
    0.12, 0.10, ...
])
</candidate>
```

**Mutation Instruction.** Example operations include: – modifying circle coordinates or radii.

**Crossover Instruction.** Example operations include: – swapping coordinate or radius values between two parent candidates.

**Additional Requirements.** Keep all circle centers inside the unit square. Large changes to coordinates, ordering, or radii are allowed and encouraged. You do not need to ensure validity; overlaps and boundary violations will be corrected automatically.

**Objective: radii.** The objective `radii` corresponds to the sum of all circle radii.

> **Prompt Example for NK2R Peptide Agonist Design**
>
> **Task Description.** This task aims to design highly selective, high-affinity peptide agonists for the Neurokinin-2 Receptor (NK2R, UniProt ID: P21452). The objective is to maximize `ipTM` (interface predicted TM-score) computed by AlphaFold3. `ipTM` ranges from 0 to 1 and measures the predicted structural accuracy and reliability of the peptide–receptor binding interface in AlphaFold-Multimer and AlphaFold3.
>
> **Output Format.** Each generated candidate must begin with `<candidate>` and end with `</candidate>` in peptide sequence format (natural amino acids only). Example:
>
> `<candidate>ACDEFGH</candidate>`
>
> **Mutation Instruction.** Example operations include: – increasing or decreasing peptide length; – altering secondary-structure–related residue patterns; – modifying local sequence motifs to adjust predicted interface geometry.
>
> **Crossover Instruction.** (No crossover instruction provided.)
>
> **Additional Requirements.** 1. Candidates must be valid peptide sequences composed of natural amino acids. 2. Maximum sequence length must not exceed 40 residues. 3. Sequence similarity to the natural ligand NKA (HKTDSFVGLM) must be below 30% according to `mmseqs2` (indicated by `pass_mmseqs`). 4. Only peptides satisfying both constraints—`pass_mmseqs` = true and length $\leq 40$—will receive a non-zero `ipTM` value. 5. Designs should avoid unintended activation of NK1R or NK3R.
>
> **Objectives.**
>
> `iptm_ours`. The predicted `ipTM` of the designed peptide with NK2R; higher is better.
>
> `iptm_nka`. The predicted `ipTM` of the natural peptide NKA; lower is preferred so that the designed peptide competes more effectively with the native ligand.
>
> `similarity`. Sequence similarity to NKA computed via Biopython. When `pass_mmseqs` is false, lower similarity is preferred to encourage novelty. Note that this similarity differs from that computed by `mmseqs2`; a similarity $> 30\%$ by Biopython does *not* imply $> 30\%$ similarity by `mmseqs2`.

## 6.8 DETAILED MOTIVATION OF 3 INITIALIZATION SCHEMES

The motivation for the three initialization schemes is to study how sensitive different optimization methods are to the quality of the starting population under a fixed oracle budget. In practice, the initial set of molecules in a real molecular discovery campaign can vary widely depending on how much prior information is available, and evolutionary optimization methods are known to be sensitive to such variability. Best-init and random-init reflect the two most common real-world scenarios: **Best-init** models a setting where researchers have prior predictors, literature candidates, or earlier high-quality hits to initialize the search. **Random-init** corresponds to the common case in de novo tasks where no prior structural biases exist and the initial library is obtained by uniform sampling or scaffold enumeration. The **worst-init** setting is not intended to represent a realistic choice made by practitioners, but rather serves as a stress test that evaluates *robustness to poor initialization*. In realistic campaigns, poorly chosen starting points can arise unintentionally due to limitations of surrogate predictors, scaffold biases, or dataset-specific artifacts, especially when working with noisy or unreliable scores. Evolutionary pipelines can fail catastrophically in these cases, collapsing early and wasting the entire evaluation budget. Including worst-init therefore reveals whether a method can recover from unfavorable starting points and still navigate toward high-quality regions of chemical space.

## 6.9 EXPERIENCE EXAMPLES

This section demonstrates experience from two tasks, the 5-objective optimization and single-objective optimization. The experience of 5-objective optimization task with random initilization is extracted from the experiments in Table 1 in the main content. And the single-objective task is optimizing JNK3 on PMO benchmark.

## Initial experience of 5 objectives

**1. Excellent Molecules:**

- **Balanced Substituents:** Molecules with balanced, non-bulky substituents tend to have lower SA values.
- **Heterocycles & Aromatic Rings:** Incorporation of heterocyclic systems and aromatic rings contributes to favorable DRD2 and QED values.
- **Hydrophobic and Polar Groups:** Presence of hydrophobic aromatic systems along with polar functional groups like amides, ethers, and amines enhances GSK3$\beta$ and JNK3 selectivity.
- **Stereochemistry:** Utilization of chiral centers often aids in achieving higher QED and specificity for GSK3$\beta$ and JNK3.

**2. Poor-Performing Molecules:**

- **High SA Scores:** Often due to bulky, complex substituents and extensive branching.
- **Low QED Values:** Simplicity or lack of functional diversity can result in lower QED scores.
- **High DRD2 Values:** Overly hydrophobic or basic molecules tend to have higher DRD2 values, possibly leading to off-target effects.

**Strategies to Optimize New Molecules**

**1. Decrease SA Value:**

- Favor linear or moderately branched structures with controlled stereochemistry.
- Avoid excessive bulky groups and complex fused rings.

**2. Decrease DRD2 Value:**

- Integrate balanced hydrophobic and hydrophilic groups to avoid nonspecific binding.
- Use heterocycles to enhance specificity.

**3. Increase QED Value:**

- Aim for a balance in molecular weight, lipophilicity, and aromatic character.
- Incorporate functional groups that enhance drug-likeness, such as amides, esters, and ethers.

**4. Decrease GSK3$\beta$ Value:**

- Select functional groups known for specific enzyme interactions, like amides and imides.
- Leverage computational tools to tailor interactions for GSK3$\beta$.

**5. Increase JNK3 Value:**

- Incorporate chiral centers to improve selectivity.
- Include moieties known for JNK3 interactions, such as specific aromatic or heterocyclic systems.

**Avoiding Suboptimal Properties**

- **Reduce Molecular Complexity:** Avoid overly complex molecules with high SA values.
- **Enhance Functional Diversity:** Ensure a good mix of polar and non-polar groups to avoid low QED.
- **Modulate Hydrophobicity:** Avoid excessive hydrophobicity, which increases DRD2 values and off-target effects.

**Final experience of 5 objectives**

- **Aromatic Cores:** Utilize benzene and thiophene rings for stability and enhanced bioactivity.
- **Functional Groups:** Prefer amides, carbamates, esters, and ethers to enhance QED and JNK3 while reducing GSK3$\beta$.
- **Halogen Substitution:** Introduce halogens, especially monosubstitution in aromatic rings, to improve SA and bioactivity.
- **Structural Simplicity:** Favor simpler, smaller structures to achieve lower SA values.
- **Selective Substitution:** Utilize monosubstitution in aromatic rings to balance low SA and high bioactivity.
- **Bioactivity Optimization:** Enhance QED and JNK3 values while minimizing DRD2 and GSK3$\beta$ values.
- **Avoid Bulky Groups:** Minimize bulky groups to maintain simplicity, lower SA, and sustained bioactivity.
- **Functional Integration:** Combine hydrophobic and polar groups strategically to optimize bioactivity and maintain low SA.
- **Linear and Compact Structures:** Avoid complex branching; favor linear and compact molecules to minimize DRD2 and GSK3$\beta$.

**Initial experience of JNK3**

**Summary of Molecular Optimization Insights**
**1. Characteristics of High-Performing Molecules**

- The presence of aromatic rings and heterocycles is prevalent.
- Functional groups such as amines, ethers, and sulfoxides are common.
- Chiral centers and stereochemistry play a significant role.
- Substitution on aromatic rings with electron-withdrawing groups like chlorine and fluorine enhances performance.
- Alkyl side chains and central amine linkages contribute to activity.

**2. Strategies for Designing New Molecules**

- Incorporate aromatic systems and heterocyclic structures to increase stability and specificity.
- Introduce functional groups like amines, ethers, and sulfoxides to enhance binding interactions.
- Leverage chiral centers and stereochemistry to improve efficacy and selectivity.
- Utilize electron-withdrawing groups for aromatic ring substitutions to enhance activity.
- Ensure balanced lipophilicity and solubility through careful side chain selection.

**3. Reasons for Poor Performance in Low-Scoring Molecules**

- Lack of aromatic or heterocyclic components reduces stability and binding efficacy.
- Absence of key functional groups like amines and ethers diminishes interaction potential.
- Insufficient stereochemistry and chirality lead to lower specificity and activity.
- Overly simple molecular structures lack necessary complexity and interaction sites.

**4. Avoidance of Suboptimal Properties**

- Prioritize the inclusion of aromatic systems and heterocyclic compounds.
- Add diverse functional groups to create better binding and interaction profiles.
- Design molecules with defined stereochemistry to enhance specificity.
- Maintain a balance of molecular complexity to ensure both efficacy and manageable synthesis.

Following these insights can guide the design of new molecules with enhanced JNK3 values and overall better performance. You can take advantage of these experiences to propose better molecules aligned with the optimization objectives.

---

**Final experience of JNK3**

**Integrated Experience for Molecular Optimization**
**High-Performing Molecule Characteristics**

- **Key Functional Groups:** Sulfonamide, sulfonyl, N-substituents (amines, alcohols).
- **Structural Features:** Aromatic and heterocyclic rings with flexible or cyclic linkers.
- **Hydrophobic/Hydrophilic Balance:** Achieved through diverse functional groups.
- **Electron-Withdrawing Groups:** Nitrogen-based groups on aromatic or heterocyclic rings.

**Design Strategies**

1. **Functional Groups:** Incorporate sulfonamide or sulfonyl moieties to enhance binding affinity and solubility.
2. **N-Substituents:** Employ functionalized side chains to improve molecular flexibility and target specificity.
3. **Electron-Withdrawing Groups:** Optimize electron interactions to strengthen binding affinity.
4. **Structural Flexibility:** Utilize cyclic and flexible linkers to promote favorable binding dynamics.

**Reasons for Poor Performance in Low-Scoring Molecules**

- **Lack of Key Groups:** Absence of critical functional groups reduces binding capacity and solubility.
- **Steric Hindrance:** Presence of bulky substituents hinders effective binding.
- **Suboptimal Electron Density:** Insufficient electron interactions weaken molecular efficacy.
- **Poor Functionalization:** Ineffective ring substitutions compromise performance.

**Actionable Insights**

- Incorporate diverse N-substituents and electron-withdrawing groups.
- Maintain an appropriate hydrophobic/hydrophilic balance.
- Avoid excessive steric hindrance in functional group placement.
- Design flexible or cyclic linkers to enhance dynamic binding interactions.

These insights can guide the development of improved molecules that better satisfy multi-objective optimization criteria.

---

## 6.10 VISUALIZATION OF MOLECULES

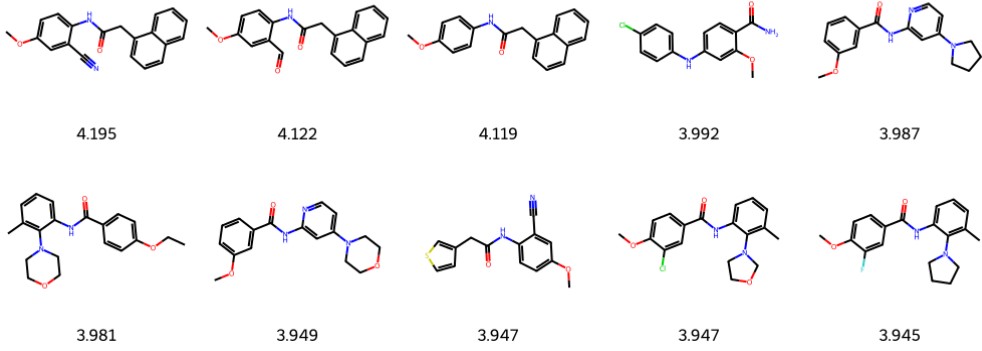

Figure 12: MOLLEO

The number under each molecule is the F value of it.

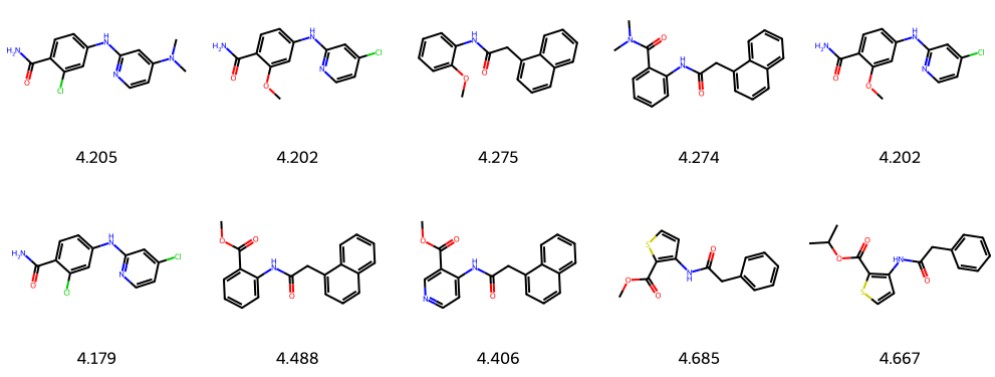

Figure 13: ExLLM (ours)

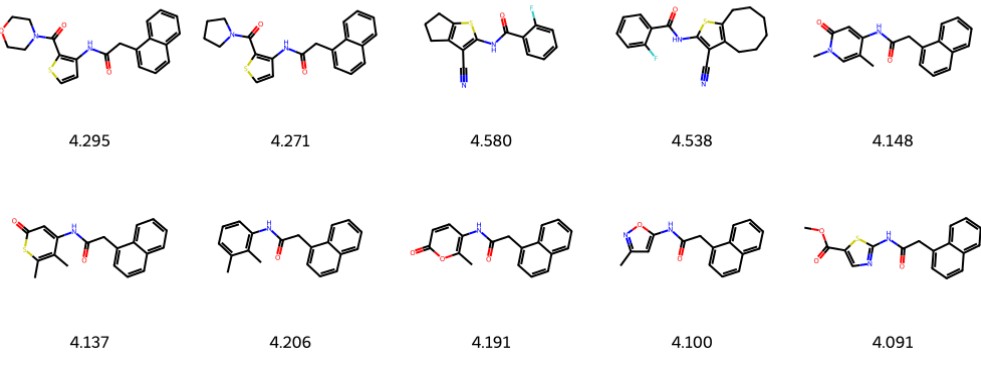

Figure 14: DyMol

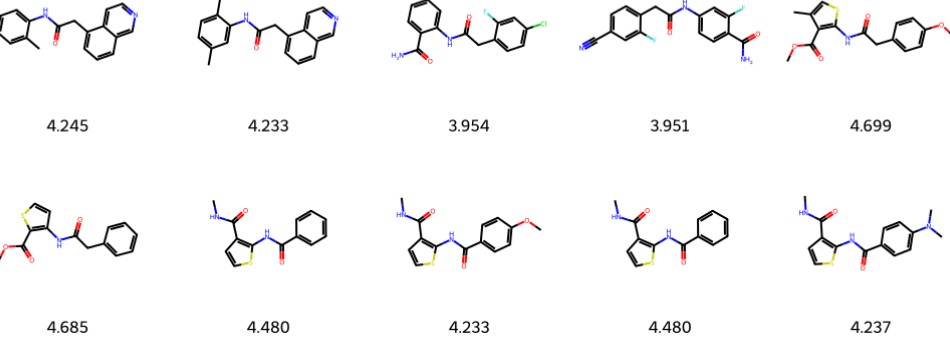

Figure 15: Genetic-GFN

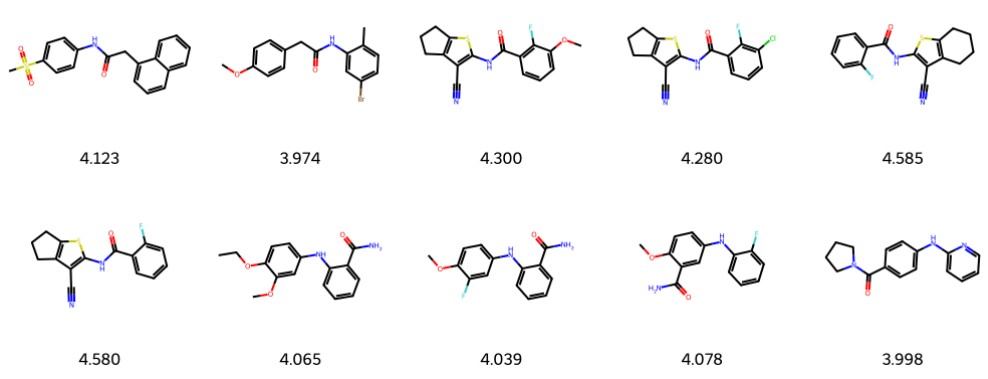

Figure 16: REINVENT

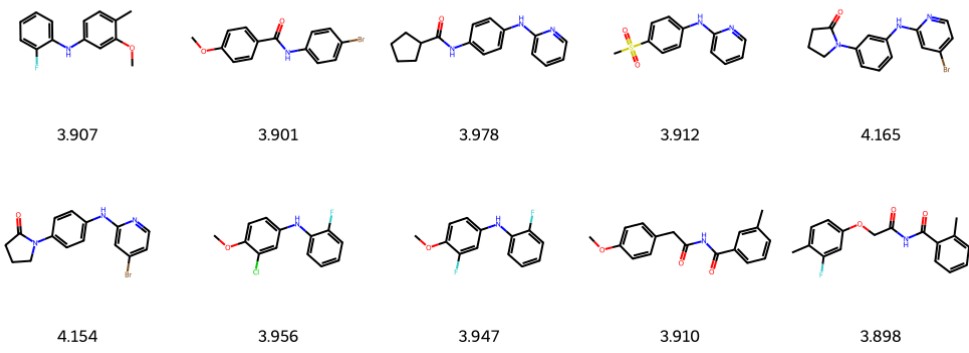

Figure 17: GB-GA

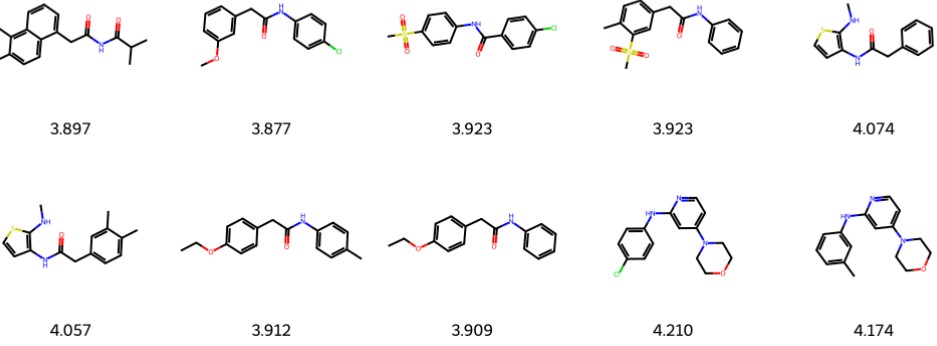

Figure 18: GB-BO

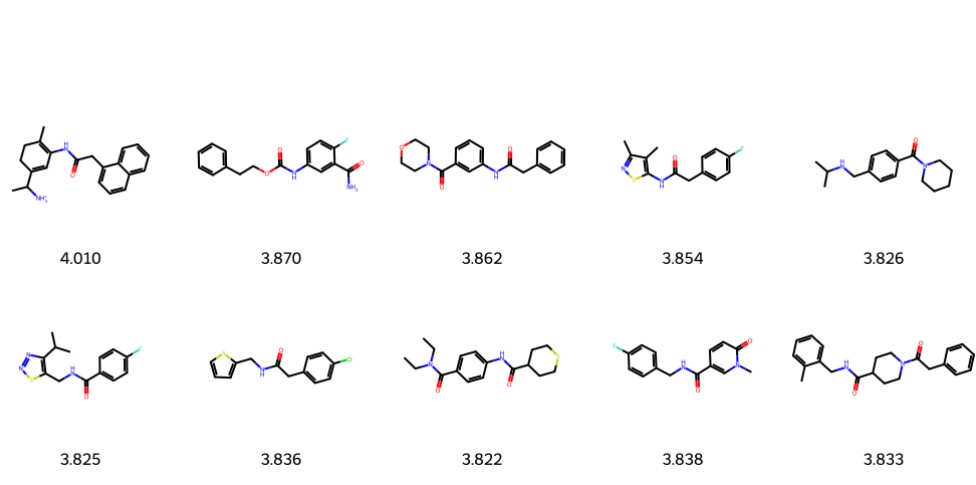

Figure 19: JT-VAE

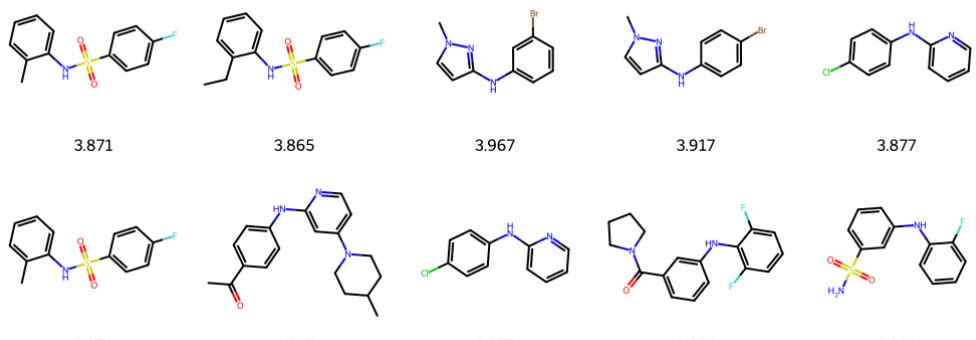

Figure 20: MARS

