# OpenReview forum: "ExLLM: Experience-Enhanced LLM Optimization for Molecular Design and Beyond"
_ICLR.cc/2026/Conference — Submitted to ICLR 2026_

### Official Review · Reviewer_CwLi · 2025-10-29

**Soundness:** 3
**Presentation:** 2
**Contribution:** 3
**Rating:** 4
**Confidence:** 4

**Summary:**

The authors propose ExLLM, a framework that enables the effective utilization of LLMs for molecular design without re-engineering or additional training. ExLLM employs a k-offspring and an evolving experience mechanism to achieve efficient exploration and maintain a non-redundant population. Moreover, the proposed feedback adapter allows the framework to perform consistent and stable optimization in multi-objective settings

**Strengths:**

- The paper presents a compelling framework that leverages pretrained LLMs as optimizer without requiring any additional training. Also, ExLLMs showed improved performance gains in both single-objective and multi-objective optimization tasks

- The k-offspring and evolving experience effectively enhance the exploration capability of pretrained LLMs while reducing the number of API calls and costs. This design choice represents a novel contribution that improves efficiency without sacrificing optimization quality

**Weaknesses:**

- In the paper, the comparison with other LLM-as-optimizer models is somewhat limited. While approaches such as OPRO, LMEA, and AlphaEvolve are mentioned, the paper does not provide experimental results against them in the molecular design. Moreover, other relevant molecular design-aware LLM frameworks (e.g., ChemCrow, LICO, MolReGPT, Prompt-MolOpt) are discussed but not empirically compared.

- The paper primarily relies on GPT-4o or Gemini, without evaluating open-source LLMs such as Llama[1] or Qwen[2].

- Although the work is positioned as an LLM optimizer for molecular design, the paper provides limited discussion on molecular design.

- Figure 1 could be improved for clarity. Some key components(e.g., k-offspring) are not visually highlighted, making it somewhat difficult for readers.

[1] Dubey, Abhimanyu, et al. "The llama 3 herd of models." arXiv e-prints (2024): arXiv-2407.

[2] Yang, An, et al. "Qwen3 technical report." arXiv preprint arXiv:2505.09388 (2025).

**Questions:**

- (w1)The authors do not present experimental comparisons with other LLM-as-optimizer models in molecular design. While existing LLM optimizer studies (e.g., OPRO, LMEA, AlphaEvolve) may not be specifically designed for molecular design tasks, ExLLM also operates without explicit in-context learning, prompt engineering, or domain-specific feedback. Therefore, it seems feasible to evaluate these LLM optimizer frameworks on molecular design. In addition, could you clarify why MOLLEO was selected as the only molecular-design baseline(e.g., ChemCrow, LICO, MolReGPT, Prompt-MolOpt)? If there are practical constraints in evaluating agent-based frameworks or using external chemical tools, it would be helpful to mention that explicitly, and include comparisons with any executable models if possible.

-  (w2)The paper only presents results using GPT-4o and Gemini. However, it seems that other open-source LLMs such as Llama[1], Qwen[2] could also be applied to ExLLM. In particular, Qwen[2], which adopts a Mixture-of-Experts (MoE) architecture, is expected to offer advantages in terms of API calls and inference time. It would be helpful to include additional ablation experiments comparing their cost and performance.

-  (w3)Although the paper is positioned as an LLM optimizer for molecular design, there is little discussion related to the generated molecules. In particular, since k-offspring is introduced only in the ablation study, it would be helpful to include a brief analysis of the sampled offspring themselves.  Moreover, a pareto front plot and an oracle call curve (for PMO single-objective optimization) are not presented.

- In Table 2(and Table 10), the reported diversity appears to be somewhat lower, even though uniqueness remains high. Could you elaborate on this observation? Is the lower diversity mainly due to differences in how the metric is defined, or should it be understood as a trade-off between fitness and diversity?

- Could you provide more details on the selection process? It could be viewed as a form of sampling, but it currently appears to be implemented solely as a weighted-sum operation. Are there any experiments comparing this approach with other selection methods? For example, methods such as Chebyshev scalarization sampling[3], Tchebycheff Scalarization[4], or Dirichlet distribution–based sampling[5] could also be considered. Are these approaches incorporated into the pareto front–based selector? Additional explanation of the fitness and pareto selectors would be helpful.

- In the appendix ‘7.9 NUMBER OF OUTPUT MOLECULES’, “Table??” appears to be a LaTeX mapping error. Please correct it.

[3] Chugh, Tinkle. "Scalarizing functions in Bayesian multiobjective optimization." 2020 IEEE Congress on Evolutionary Computation (CEC). IEEE, 2020.

[4] Lin, Xi, et al. "Smooth tchebycheff scalarization for multi-objective optimization." arXiv preprint arXiv:2402.19078 (2024).

[5] Shin, Dong-Hee, et al. "Offline Model-based Optimization for Real-World Molecular Discovery." Forty-second International Conference on Machine Learning.

---

> ### Author Response · Authors · 2025-11-28
> **Response to Reviewer CwLi (Part 1)**
>
> **Reviewer Question:** *"In the paper, the comparison with other LLM-as-optimizer models is somewhat limited ... it seems feasible to evaluate these LLM optimizer frameworks ((e.g., OPRO, LMEA, AlphaEvolve)) on molecular design."*
>
> **Response:**
>
> AlphaEvolve, OPRO, and LMEA are not suitable baselines for molecular optimization. These frameworks are designed for program synthesis, prompt optimization, and combinatorial sequence optimization (e.g., TSP), respectively, and cannot operate on molecular structures or multi-objective chemical objectives without substantial modifications to their core modules. ChemCrow is also not an appropriate baseline: although it is a capable tool-using agent, it does not generate or optimize molecular structures and therefore cannot be evaluated on PMO-style optimization tasks.  Prompt-MolOpt and MolReGPT, while relevant to molecular generation, are not general-purpose optimizers, they rely on task-specific supervised training or finetuning and cannot optimize arbitrary multi-objective molecular functions in a zero-shot manner as required in our setting.
> To ensure fair comparison, we include two recent LLM-based molecular optimization systems that are directly applicable to PMO: **LICO (2025)** and **Chemma (2024)**. These represent the current SOTA LLM optimizers in this domain. Their results are added in Sec. 6.4, and a summary is provided below.
> | Task type & Objective (↑) | LICO | Chemma 2B | **ExLLM (Ours)** | LICO (PMO-1K) | ExLLM (PMO-1K) |
> |---------------------------|------|-----------|------------------|----------------|----------------|
> | **Total (↑)**             | 14.708 | 17.534 | **19.165** | 11.71 (Prior SOTA on PMO-1K) | **14.533** |
> | **Rank (↓)**              | 6 | 3 | **1** | 2 | **1** |
>
> The results show that our ExLLM still performs the best compared to other SOTA LLM-based methods, as well as chemistry-specific LLM Chemma- 2B[1], and even show a significant improvement of 24.1% in PMO-1K introduce by LICO.
>
> [2]Guevorguian, Philipp, et al. "Small molecule optimization with large language models." arXiv preprint arXiv:2407.18897 (2024).
>
> **Revision:** The additional results and discussion are added in Sec 6.4
>
> ---
>
> **Reviewer Question:** *"The paper primarily relies on GPT-4o or Gemini, without evaluating open-source LLMs such as Llama[1] or Qwen[2]."*
>
> **Response:**
> Thank you for your suggestions. We have added a new ablation in Sec 6.3.4 evaluating ExLLM with both proprietary LLMs (GPT-4o and Gemini-2.5-Flash) and open-source LLMs (DeepSeek-V3.1 and Qwen3-Max). The results show that ExLLM consistently maintains strong optimization performance across all models, demonstrating that the framework is not tied to a particular backbone and generalizes well to open-source LLMs.
> The full comparison is provided below:
> | LLM in ExLLM | Top1 F | Top10 F | AUC-Top10 | Hypervolume | Uniqueness | Validity | Diversity |
> |--------------|--------|---------|-----------|-------------|------------|----------|-----------|
> | GPT-4o | **4.336 ± 0.246** | **4.300 ± 0.164** | **4.116 ± 0.040** | **0.905 ± 0.200** | **0.872 ± 0.015** | 0.908 ± 0.019 | 0.494 ± 0.032 |
> | Gemini-2.5-Flash | 4.208 ± 0.011 | 4.191 ± 0.009 | 4.070 ± 0.026 | 0.750 ± 0.007 | 0.615 ± 0.035 | 0.950 ± 0.006 | **0.521 ± 0.013** |
> | DeepSeek-V3.1 | 4.175 ± 0.039 | 4.152 ± 0.040 | 3.994 ± 0.039 | 0.763 ± 0.100 | 0.544 ± 0.072 | 0.959 ± 0.009 | 0.496 ± 0.020 |
> | Qwen3-Max | 4.075 ± 0.123 | 4.056 ± 0.115 | 3.968 ± 0.085 | 0.587 ± 0.215 | 0.316 ± 0.041 | **0.974 ± 0.011** | 0.500 ± 0.045 |
>
> **Revision:** These results and discussion are added in Sect 6.3.4
>
> ---
>
> **Reviewer Question:** *"the paper provides limited discussion on molecular design"*
>
> **Response:**
> We thank the reviewer for raising this concern. We already have visualized generated molecules in Sec 6.10, experience examples for molecular optimization in Sec 6.9. On top of that, we have also added the pareto-front plots, parallel coordinate plots and per-objective improvement of molecular optimization Sec 6.6, showing the good coverage of optimized molecules and consistent improvement.
>
> ---
>
> **Reviewer Question:** *"Moreover, a pareto front plot and an oracle call curve (for PMO single-objective optimization) are not presented."*
>
> **Response:**
> Thank you for the suggestion. We have added full Pareto-front plots and parallel coordinate visualizations in Sec. 6.6. We also include objective-value vs. oracle-call curves for ExLLM on the PMO benchmark in Sec. 6.4, which illustrate its stable behavior and rapid convergence across tasks. These additions provide a clearer picture of ExLLM’s optimization dynamics and further demonstrate its effectiveness.
>
> **Revision:** Plots are added in Sec 6.6 and 6.4

---

> ### Author Response · Authors · 2025-11-28
> **Response to Reviewer CwLi (Part 2)**
>
> **Reviewer Question:** *"Is the lower diversity mainly due to differences in how the metric is defined, or should it be understood as a trade-off between fitness and diversity?"*
>
> **Response:**
> Thank you for the question. The lower diversity is not caused by differences in how the metric is defined. Instead, it reflects a **deliberate fitness–diversity trade-off** that emerges in the later stages of optimization. As discussed in the paper (end of Sec 4.1), ExLLM gradually shifts toward tighter, high-value exploration when the search approaches high-fitness regions. This naturally reduces spread in the top-100 set, but it is a **strategic behavior** rather than a methodological limitation: focusing on more fine-grained exploitation in the later stage of optimization enables ExLLM to consistently discover high-quality molecules and maintain SOTA performance across initialization schemes.
>
> Importantly, overall coverage remains competitive, as indicated by the strong hypervolume results.
>
> ---
>
> **Reviewer Question:** *"Could you provide more details on the selection process? It could be viewed as a form of sampling, but it currently appears to be implemented solely as a weighted-sum operation. Are there any experiments comparing this approach with other selection methods?"*
>
> **Response:**
> Thank you for the insightful question regarding the selection strategy. We clarify that ExLLM does not rely solely on a weighted-sum selector. Instead, it uses a **hybrid selector** that combines (i) a fitness-based selector and (ii) a Pareto-front selector. This is implemented as a 50–50 split at each generation, ensuring both exploitation and diverse multi-objective exploration. To address the reviewer’s concerns, we have added a dedicated ablation study in Sec. 6.3.3. The results are summarized below:
> | Selector | Top1 F | Top10 F | AUC-Top10 | Hypervolume | Uniqueness | Validity | Diversity |
> |----------|--------|---------|-----------|-------------|------------|----------|-----------|
> | Only Pareto front-based | 4.170 ± 0.055 | 4.092 ± 0.042 | 3.957 ± 0.037 | 0.857 ± 0.061 | 0.870 ± 0.025 | 0.909 ± 0.012 | **0.667 ± 0.049** |
> | Only Fitness-based | 4.163 ± 0.190 | 4.055 ± 0.094 | 3.938 ± 0.048 | 0.680 ± 0.274 | 0.844 ± 0.015 | **0.916 ± 0.005** | 0.517 ± 0.031 |
> | Hybrid selector | **4.336 ± 0.246** | **4.300 ± 0.164** | **4.116 ± 0.040** | **0.905 ± 0.200** | **0.872 ± 0.015** | 0.908 ± 0.019 | 0.494 ± 0.032 |
>
> The hybrid selector achieves the highest values on Top-1 F, Top-10 F, AUC-Top10, and hypervolume, demonstrating more stable optimization and better convergence across objectives. Fitness-only selection tends to over-exploit and shows unstable hypervolume, while Pareto-only selection maintains higher diversity but converges more slowly. Combining the two strategies avoids the weaknesses of each individual selector and consistently produces the best overall multi-objective optimization performance.
> Regarding the reviewer’s suggestion on alternative sampling strategies (e.g., Tchebycheff scalarization): these methods are not incorporated in our current framework. Extending ExLLM with additional scalarization-based selectors is an interesting direction, but is beyond the scope of this paper and will be explored in future work.
>
> ---
>
> **Reviewer Question:** *"Editorial issues: Table??"*
>
> **Response:**
> We thank the reviewer for finding these issues, we have fixed all editorial issues.

---

### Official Review · Reviewer_ds92 · 2025-10-31

**Soundness:** 2
**Presentation:** 2
**Contribution:** 2
**Rating:** 2
**Confidence:** 4

**Summary:**

This work introduces an LLM-as-optimizer framework with three contributions: an evolving experience snippet, a k-offspring scheme, and a feedback adapter for molecular design and optimization. Under fixed evaluation budgets this approach improves on PMO benchmark, across single- and multi-objective molecular optimization settings.

**Strengths:**

The manuscript reads mostly clearly.
Strong results on PMO benchmark.
The experiments included extra in addition to small molecules such as peptides.
Six-objective experiments with good results are novel.

**Weaknesses:**

The three main contributions seem to focus on “low cost” but discussion on cost perspective significantly lacks.
The framework looks close to MOLLEO, as LLMs handle mutation and crossover on the parent molecules and the evaluation and update pools rely on conventional metrics, even if the templates looked carefully crafted in their overview.

Figure 1 and the method section do not provide a clear end-to-end overview. Subsections in Section 3 Method do not fully match the figure. The paper should clarify how the next population is selected in single vs multi objective cases, given that Pareto selection applies only to multi objective settings.

Editorial issues. Please look through the whole manuscript and fix indexing. Below are some wrong referencing I have found.
-line 141 says Figure 8 for the framework overview, but I assume they refer to Figure 1.
-Table 2 is missing some boldface for some best results.
-Around lines 316-317, the text points to Table 10 which appears to mean Table 2.

More concerns and questions will be placed in Questions section below.

**Questions:**

This lies with Weakness 1 regarding the main contributions. Could you compare cost directly and show how the framework reduces cost, for example with ablations or matched cost vs quality analysis in detail?

Reading through the method and the overview figure, I assume the task-specific templates and the prompts seem very important, yet details are limited. Please describe design choices.

The LLM-as-optimizer idea is emphasized, but the final decision makers are a fitness-based selector and a Pareto front-based selector for next populations and choosing the best molecules. Have you done some ablations that replaces LLM proposals with molecules that are generated in rule-based way and optimizes with the fitness/pareto-based selectors?

In Figure 2 in Ablation study, GPT-based variants are consistently better in both single and five objective settings regardless of k. The curves over k do not show a stable pattern, especially in multi-objective setting. Clear evidence that k  offspring scheme itself actually add meaningful contribution to the framework should be included.

In multi objective molecular optimization, some objectives like QED and SA are easier than others. In the five-objective setting, can you provide evidence that the framework balances the objectives rather than leaning on easy objectives? For example, through per-objective improvements and coverage of the Pareto front.

---

> ### Author Response · Authors · 2025-11-28
> **Response to Reviewer ds92 (Part 1)**
>
> **Reviewer Question:** *"The three main contributions seem to focus on “low cost” but discussion on cost perspective significantly lacks. ... The framework looks close to MOLLEO ..."*
>
> **Response:**
> We thank the reviewer for the feedback and wish to clarify a central misunderstanding.
> **ExLLM does not follow the MOLLEO workflow**, and our main contributions do not focus on “low cost”, but on a powerful optimizer for large discrete space search with significant generalizability.  Our method introduces **novel algorithmic components not present in MOLLEO or prior LLM-based molecular optimization pipelines**, and these components are essential for the performance and cross-domain generalization demonstrated in the paper. We also added a new method overview diagram in **Figure 1**, which also shows our contributions.
>
> ## 1. ExLLM is not based on MOLLEO
> LLM-as-optimizer has become a well-established direction after OPRO, LMEA, and related work. Our method is developed within this broader paradigm, but it is **not** derived from MOLLEO. We start from the challenges of large discrete optimization and design an end-to-end framework from scratch. ExLLM introduces two core algorithmic components: **the first experience memory designed specifically for large discrete search** and an **autoregressive k-offspring exploration mechanism**. These enable ExLLM to achieve new state-of-the-art performance in multi-objective molecular optimization, while reducing runtime and cost by an order of magnitude, and generalizing across domains with a unified complex feedback adaptor.
> In contrast, MOLLEO is a GA–LLM hybrid tailored to molecular optimization, **with no memory mechanism, no k-offspring strategy, and no cross-domain generalizability**.
> Our contributions are algorithmic rather than “practical tips.” Each component has a clear theoretical motivation and demonstrates measurable impact:
> ### (a) First experience memory for large discrete optimization (Sec. 3.1)
> While memory is important in LLM-as-optimizer frameworks, conventional retrieval-based memory causes **information and storage explosion**, and leads to **significant runtime and cost increases** (Table 1).
> We propose an evolving experience pool with a curriculum-style update scheme, enabling a **lightweight memory** to remain consistently effective. On PMO, removing the experience module yields 18.165 (already state-of-the-art); adding it improves performance to 19.165 (Sec. 4.2), with full ablations in Sec. 6.3.2.
> We further incorporate probabilistic memory injection to balance exploration and exploitation (Sec. 6.3.2).
> ### (b) Autoregressive k-offspring exploration (Sec. 3.2)
> We leverage the autoregressive factorization of LLMs to generate k diverse offspring per call, increasing exploratory breadth while reducing cost and runtime.
> Comprehensive ablations (Sec. 6.3.1) show consistent gains when k>1.
> Runtime and cost comparisons are added now in Sec. 6.5 and summarized below:
>
> | Method            | ExLLM (GPT-4o) | MOLLEO (GPT-4o) | Graph GA      | Gp-BO         | Genetic-GFN   | MARS          | JT-VAE        | DyMol         | REINVENT      |
> |-------------------|----------------|------------------|----------------|----------------|----------------|----------------|----------------|----------------|----------------|
> | Running time (hours) | 0.393 ± 0.114 | 6.029 ± 1.281    | 1.041 ± 0.156 | 0.683 ± 0.100 | 0.453 ± 0.071 | 0.692 ± 0.129 | 3.082 ± 0.362 | 0.289 ± 0.037 | 0.522 ± 0.088 |
>
> ---
>
> | Method                   | Total input tokens | Total output tokens | Input tokens for experience | Output tokens for experience | Total cost (USD) | Running time (h) |
> |--------------------------|--------------------|----------------------|------------------------------|-------------------------------|-------------------|------------------|
> | MOLLEO (GPT-4o)          | 2.991 ± 0.246      | 2.191 ± 0.201        | N/A                          | N/A                         | 19.126 ± 1.698    | 6.029 ± 1.281    |
> | ExLLM (GPT-4o)           | 2.857 ± 0.338      | 0.204 ± 0.020        | 0.253 ± 0.031                | 0.031 ± 0.005                 | 6.938 ± 0.796     | 0.393 ± 0.114    |
> | ExLLM (Gemini-2.5-Flash) | 3.799 ± 0.456      | 0.279 ± 0.036        | 0.365 ± 0.029                | 0.027 ± 0.004                 | 1.102 ± 0.136     | 0.495 ± 0.133    |
> | ExLLM (DeepSeek-V3.1)    | 4.145 ± 0.364      | 0.283 ± 0.024        | 0.372 ± 0.033                | 0.037 ± 0.004                 | 2.816 ± 0.246     | 1.703 ± 0.341    |
> | ExLLM (Qwen3-Max)        | 4.316 ± 1.687      | 0.285 ± 0.110        | 0.408 ± 0.160                | 0.051 ± 0.020                 | 4.604 ± 0.389     | 1.575 ± 0.796    |

---

> ### Author Response · Authors · 2025-11-28
> **Response to Reviewer ds92 (Part 2)**
>
> ## 2. Strong evidence of novelty and impact
> If the method were merely incremental, it would not achieve:
> - **New SOTA on PMO** (19.165 total score, +7.3% over prior best),
> - **1st place on 17/23 PMO tasks**,
> - **Robust cross-domain generalization** across physics (stellarator design), geometry (circle packing), engineering (offshore jackets), routing problems (MOCVRP), peptide design, and GPU kernel optimization. We have moved the results to Sec 4.3 in the main text instead of only being in appendix.
> These domains differ fundamentally from MOLLEO’s scope.
>
> ## 3. Summary
> ExLLM provides a **new LLM-as-optimizer framework for optimization problems with large discrete search space**, in terms of:
> - evolving experience memory specifically for optimization with large discrete search space,
> - autoregressive multi-offspring exploration,
> - unified multi-objective & constraint-aware feedback handling,
> - and demonstrated generalization across 7 diverse domains.
> Therefore, the method is **not derived from MOLLEO**, and the contributions go significantly beyond prompt engineering or cost reduction.
> We hope this resolves the misunderstanding and accurately reflects the novelty of our work.
>
> **Revision**: Clarification and cost analysis provided in 6.5
>
> ---
>
> **Reviewer Question:** *"Figure 1 and the method section do not provide a clear end-to-end overview."*
>
> **Response:**
> We appreciate the reviewer’s concern about Figure 1. We have added a new Figure 1 which clearly provide a clear end-to-end overview, and the old Figure 1 now become Figure 2 describing the details of ExLLM.
>
> ---
>
> **Reviewer Question:** *" The paper should clarify how the next population is selected in single vs multi objective cases"*
>
> **Response:**
> Thank you for your problem, we just set two situation: when the number of objectives > 1, we use the hybrid selector, otherwise we just use the fitness-based selector (because pareto-based is not applicable).
>
> ---
>
> **Reviewer Question:** *"Editorial issues"*
> **Response:**
>
> We thank the reviewer for finding these errors, we have completely fixed all of them in the revised version.
>
> ---
>
> **Reviewer Question:** *"Could you compare cost directly and show how the framework reduces cost"*
>
> **Response:**
> Thank you for your suggestions. The cost table is listed above and also added to Sec 6.5. From the main results from Table 2 and ablation results of 1-6 objectives in Table 14, we can observe that ExLLM always maintain a high Uniqueness, which enhances its exploration abilities therefore enables it to complete the fixed budget with less costs and runtime. And the improved uniqueness is mainly because of k-off strategy as shown in ablation study Sec 6.3.1. Because of the k>1, it further reduces LLM queries and runtime while producing improved results.
>
> ---
>
> **Reviewer Question:** *"I assume the task-specific templates and the prompts seem very important, yet details are limited. Please describe design choices."*
>
> **Response:**
> We appreciate the review’s concern about task-specific templates. In fact, the template is straightforward and does not need to be carefully crafted as long as it is correct and complete. Users can directly use the official definition of the tasks and descriptions of objectives. We have provided the prompt template examples of multi-objective molecular design, circle packing and peptide design as well as a discussion in Sec 6.7.
>
> **Revision:** Example task templates added in Sec 6.7
>
> ---
>
> **Reviewer Question:** *" Have you done some ablations that replaces LLM proposals with molecules that are generated in rule-based way and optimizes with the fitness/pareto-based selectors?"*
>
> **Response:**
> Ablating the LLM by replacing its proposals with rule-based mutations would effectively convert ExLLM into a standard GA (e.g., the classic GB-GA). However, this removes all three core components of our method: the evolving experience, the k-offspring autoregressive exploration, and the unified feedback adapter, which are only meaningful when the generator is LLM. The resulting system would not be an ablation of ExLLM, but simply a **different baseline** that is already well-studied.
> Moreover, prior work has already evaluated this hybrid direction. MOLLEO’s Appendix Table A2 (Row 1 and 2) shows that **LLM-based crossover + rule-based mutation (~6.7%)** provides the best performance, while replacing LLM edits with rule-based generations leads to a clear drop in optimization quality. For these reasons, a “rule-based generator" ablation would not isolate any mechanism of ExLLM and would instead reduce the method to a known GA baseline whose behavior and limitations are already established in the literature.

---

> ### Author Response · Authors · 2025-11-28
> **Response to Reviewer ds92 (Part 3)**
>
> **Reviewer Question:** *"Clear evidence that k offspring scheme itself actually add meaningful contribution to the framework should be included"*
>
> **Response:**
> We thank the reviewer for this suggestion. In the revised version, we have added **new ablations using two additional open-source LLMs** (DeepSeek and Qwen), complementing the existing GPT and Gemini results. These experiments isolate only the value of the k-offspring mechanism.
> Across all four models, the results in **Sec. 6.3.1 (Tables 9–10 and Fig. 7)** show a consistent and substantial improvement when **k > 1**, for both single-objective and multi-objective settings. The gains are especially pronounced for Qwen and Gemini(single-objective), where k-offspring significantly accelerates convergence and improves final Top-10/AUC performance. We also observe a consistent trend across models: when k=1 leads to weaker exploratory behavior (e.g., very low uniqueness), the improvement from using k>1 becomes even more pronounced. This suggests that the k-offspring strategy is especially beneficial in alleviating exploration bottlenecks in settings where the base model struggles.
>
> **Revision** The discussion and additional ablation studies are added in Sec 6.3.1
>
> ---
>
> **Reviewer Question:** *"can you provide evidence that the framework balances the objectives rather than leaning on easy objectives?"*
>
> **Response:**
> Thank you for your suggestion, we have provided comprehensive plots and per-objective improvement table in Sec6.6. The pairwise Scatter plot of the pareto-front coverage and parallel coordinate plot of the 5 objective molecular optimization shows that our results balances the objectives and consistently improve each objective rather than leaning easy objectives. We also provide the table in the following for your convenience:
> | Objective | Initial Mean ± Std | Min | Max | ExLLM Mean ± Std | Min | Max |
> |----------|--------------------|-----|-----|-------------------|-----|-----|
> | SA ↓     | 3.021 ± 0.792      | 1.652 | 5.074 | 2.037 ± 0.157 | 1.533 | 2.441 |
> | DRD2 ↓   | 0.009 ± 0.033      | 0.000 | 0.283 | 0.001 ± 0.001 | 0.000 | 0.007 |
> | GSK3β ↓  | 0.026 ± 0.040      | 0.000 | 0.270 | 0.034 ± 0.048 | 0.000 | 0.200 |
> | QED ↑    | 0.730 ± 0.143      | 0.303 | 0.936 | 0.879 ± 0.034 | 0.756 | 0.944 |
> | JNK3 ↑   | 0.015 ± 0.022      | 0.000 | 0.130 | 0.490 ± 0.234 | 0.110 | 0.950 |
>
> **Revision:** The comprehensive pareto-front converage plots, per-objective improvement table and its discussion is added in Sec 6.6

---

### Official Review · Reviewer_Kohi · 2025-10-31

**Soundness:** 2
**Presentation:** 2
**Contribution:** 2
**Rating:** 2
**Confidence:** 4

**Summary:**

This paper presents ExLLM (Experience-Enhanced LLM Optimization), which is a framework that uses LLM as optimizers for molecular design and other large discrete optimization problems. The proposed framework addresses key limitations of existing LLM-based methods for molecular design, which are often heavily prompt-dependent, require additional training, and lack memory mechanisms suited for large-scale iterative search. ExLLM consists of three main components: (1) an evolving experience mechanism that maintains a single, compact memory snippet updated each generation to distill non-redundant insights from good and bad examples, avoiding the memory bloat and exploration collapse seen in retrieval-style memories; (2) a k-offspring sampling scheme that generates k candidate molecules per LLM call by exploiting autoregressive factorization, widening exploration while reducing the number of LLM queries needed; and (3) a feedback adapter that normalizes multiple objectives into comparable vectors for Pareto-based selection and formats constraints/expert feedback into structured text for the next iteration. The framework uses a hybrid selection strategy, choosing half the population by scalar fitness (weighted sum of normalized objectives) and half from the Pareto front. The experience is injected into prompts with probability p_exp to balance exploitation and exploration.

**Strengths:**

1. I appreciate that the authors address multi-objective optimization, which many molecular generation works either omit or treat superficially. I personally believe that multi-objective optimization is critically important for real-world molecular discovery.

2. The training-free approach requires no additional model training, thereby reducing computational costs compared to other existing methods that require retraining for each new task or property.

3. Generating multiple offspring per LLM call reduces the number of required queries and thus improves computational efficiency.

**Weaknesses:**

**1. Limited Technical Novelty**
- Using LLMs for molecular design is no longer novel and has become a well-established research direction. The prior works mentioned in the introduction (ChemCrow, LICO, MolReGPT, Prompt-MolOpt, MOLLEO) already demonstrate that LLMs can be effectively applied to molecular generation/optimization tasks.

- The core contribution appears to be incremental tweaking of existing components rather than methodological innovation. The evolving experience mechanism is adapted from prior work (ReEvo, ExpeL as acknowledged by authors), the k-offspring strategy is a straightforward application of autoregressive sampling of LLMs, and the feedback adapter is essentially normalization plus text formatting. Each component individually represents a relatively minor modification to existing techniques.

**2. Insufficient Baseline Comparisons for LLM-based Methods**
- The authors mention multiple LLM-based molecular design methods in the introduction (ChemCrow, LICO, MolReGPT, Prompt-MolOpt) but only include MOLLEO as a competing LLM-based baseline in the experiments. This creates an incomplete picture of how ExLLM compares to the broader landscape of LLM-based molecular optimization approaches.

**3. Poor Paper Organization and Presentation**
- Section 2.3 contains only the text "We have put this part to appendix 7.2," which is highly unusual and unprofessional. While moving supplementary details to an appendix is acceptable practice, completely omitting a main-text section and relegating all content to the appendix is inappropriate formatting.

- This organizational choice disrupts the flow of the paper and suggests either careless preparation or an attempt to circumvent page limits. If the LLM-as-optimizer and memory mechanism background is important enough to warrant a section number, it deserves at least a brief summary in the main text with details deferred to the appendix.

**4. Insufficient ablation studies and analysis**
- No ablation on the hybrid selection strategy (50% fitness-based, 50% Pareto-based). Why this specific ratio? How sensitive is performance to this choice?

- I am still confused and not fully convinced about the clear beneficial effects of the hybrid selection strategy.

- The paper claims the experience mechanism is "lightweight" and "low-redundancy," but provides no quantitative analysis of memory consumption, prompt token counts over time, or computational overhead compared to the retrieval-style baseline beyond Table 1.

**Questions:**

1. The paper exclusively uses two proprietary LLMs (GPT-4o and Gemini) without justifying this choice or exploring alternatives. Why were only these specific models selected? Would the proposed framework work with open-source general-purpose LLMs such as Llama, Mistral, Qwen, or DeepSeek?
2. Also, would chemistry-specific LLMs such as ChemLLM or Galactica potentially perform better given their domain-specific pretraining?

3. The paper extensively evaluates three initialization schemes (worst-init, random-init, best-init) in Table 2, but provides no justification for why these specific schemes are relevant to real-world molecular discovery. In practical drug discovery scenarios, what situation would correspond to "worst-init" where researchers deliberately start with the 100 worst-performing molecules? The paper should clarify what real-world molecular discovery processes these initialization schemes are meant to simulate, and why demonstrating robustness across all three is important

4. The paper acknowledges in Table 2 that "ExLLM delivers substantial gains over the initial populations in all three init schemes, while trading some diversity for finer exploitation" and that "the diversity of the final top-100 set is somewhat lower." However, molecular diversity is a crucial consideration in real-world drug discovery for several reasons: (1) diverse chemical scaffolds provide multiple starting points for lead optimization; (2) diversity helps hedge against failures in later stages (e.g., toxicity, synthesis issues); (3) intellectual property strategies often require exploring diverse chemical matter. ExLLM's diversity scores are substantially lower than several baselines. How do the authors justify sacrificing molecular diversity for higher fitness in the context of real-world applications where diversity is often explicitly required? Would the framework be unsuitable for scaffold-hopping or exploring novel chemical space?

5. The evolving experience mechanism maintains a single, continually updated snippet that is overwritten each generation. This unidirectional update raises concerns about catastrophic forgetting. For example, early in optimization, certain structural patterns might seem unimportant and get discarded, but they could become critical after the search moves to a different region of chemical space. The paper provides no mechanism to prevent or detect such forgetting.

---

> ### Author Response · Authors · 2025-11-28
> **Response to Reviewer Kohi (Part 1)**
>
> **Reviewer Question:** *"The core contribution appears to be incremental tweaking of existing components rather than methodological innovation. The evolving experience mechanism is adapted from prior work (ReEvo, ExpeL as acknowledged by authors) "*
>
> ---
>
> **Response:**
> We thank the reviewer for raising this concern. We believe this impression arises from a misunderstanding of *what* ExLLM contributes and *why* these contributions are necessary for large-scale discrete optimization. Below we clarify the novelty and importance of each component.
>
>
> ## **1. ExLLM introduces a new framework for LLM-as-optimizer in large discrete spaces**
> While prior work has shown that LLMs can propose molecules or act as mating operators, **none provides an end-to-end optimizer architecture tailored for large discrete space search**, where issues such as memory redundancy (Table 1), exploration collapse, and heterogeneous feedback integration become critical.
> ExLLM introduces three algorithmic components that are not present in prior frameworks and are designed specifically for large discrete space search:
> ### **(a) Compact evolving experience memory for long-horizon discrete optimization** (Sec 3.1)
> Retrieval-style or append-based memories used in QA/coding tasks degrade severely in discrete optimization due to (Table 1):
> - prompt bloat
> - redundant history accumulation
> - exploration collapse over long runs
> To address these optimization-specific failure modes, we introduce:
> The first experience mechanism specifically for large discrete optimization problems in LLM-as-Optimizer.
> - Use **old experience, top good cases and uniformly sampled bad cases** to learn summarized and generalized knowledge to avoid information explosion and redundant history accumulation,
> - injected **probabilistically** to avoid over-conditioning which limits the exploration.
>
> This design is **not present in ReEvo or ExpeL**, whose memories were not designed for tens of thousands of in-loop proposals nor for preventing exploration collapse in discrete combinatorial spaces. Moreover, our evolving experience is not a modification of their mechanisms: it is a new memory architecture developed specifically for long-horizon discrete optimization. The objectives, update rules, and failure modes we address (redundancy accumulation, long-run bias, exploration collapse) do not arise in their settings, and therefore required a fundamentally different design. We have also revised this statement in the middle of Sec 3.1.
> Our ablations show that the experience mechanism provides :
> - **+1.0 improvement** in PMO total score (Sec 4.2)
> - **≈200× smaller memory footprint and costs and runtime** than retrieval-style memory (Table 1)
> - Investigation of appropriate experience injection probability (Sec 6.3.2)
>
> ### **(b) Autoregressive k-offspring exploration for efficient breadth expansion**
> Though autoregressive sampling itself is simple, **using k-offspring as an explicit exploration operator within an optimizer**, characterizing its trade-offs, and showing consistent gains under fixed evaluation budgets is novel in LLM-based molecular optimization (Sec 6.3.1).
> We show that:
> - k>1 consistently improves AUC and Fitness.
> - exploration ability increases while the cost and runtime decrease.
>
> ### **(c) Unified feedback adapter for objectives, constraints, and expert hints**
>
> Existing methods lack a general mechanism to integrate:
> - multi-objective signals,
> - hard and soft constraints,
> - optional expert textual hints,
>
> into a single structured prompt.
> Our adapter provides exactly this, enabling **plug-and-play transfer** across domains without additional training, making our system very easy to transfer while maintaining the SOTA performance across domains. This unified adapter is critical for ExLLM’s cross-domain generalization. (Results in Sec4.3 and Sec6.2)
>
> ## **2. The architectural novelty is validated by performance and generalization**
> If ExLLM were merely incremental, it would not achieve:
> - **New SOTA on PMO** (19.165 total score, +7.3% over prior SOTA),
> - **1st place in 17/23 PMO tasks**,
> - **record-breaking results on circle packing**,
> - **new SOTA on stellarator optimization**,
> - strong gains on MOCVRP, offshore jacket design, peptide design, and GPU kernel generation.
> All obtained **with the same hyperparameters across domains** and **no training**, demonstrating the strength of the architecture itself.
> ---
> We hope this clarifies the methodological innovation and resolves the misunderstanding.

---

> ### Author Response · Authors · 2025-11-28
> **Response to Reviewer Kohi (Part 2)**
>
> **Reviewer Question:** *"Insufficient Baseline Comparisons for LLM-based Methods. Would chemistry-specific LLMs such as ChemLLM or Galactica potentially perform better given their domain-specific pretraining? "*
>
> **Response:**
> ChemCrow cannot be used as a molecular optimization baseline because it does not generate or optimize molecules in any form. It is a tool-using agent, not a generative optimizer. Prompt-MolOpt and MolReGPT cannot be included as baselines because they are not general-purpose optimizers: they require additional task-specific training and cannot optimize arbitrary molecular objectives. We add LICO (2025) and Chemma (2024) [1], which represent the current SOTA performance of LLM in molecular optimization (Sec 6.4). A summary of the results are in the following:
> | Task type & Objective (↑) | LICO | Chemma 2B | **ExLLM (Ours)** | LICO (PMO-1K) | ExLLM (PMO-1K) |
> |---------------------------|------|-----------|------------------|----------------|----------------|
> | **Total (↑)**             | 14.708 | 17.534 | **19.165** | 11.71 (Prior SOTA on PMO-1K) | **14.533** |
> | **Rank (↓)**              | 6 | 3 | **1** | 2 | **1** |
>
> The results show that our ExLLM still performs the best compared to other SOTA LLM-based methods, as well as chemistry-specific LLM Chemma- 2B[1], and even show a significant improvement of 24.1% in PMO-1K introduce by LICO.
>
> [1]Guevorguian, Philipp, et al. "Small molecule optimization with large language models." arXiv preprint arXiv:2407.18897 (2024).
> **Revision:** The additional results and discussion are added in Sec 6.4
>
> ---
>
> **Reviewer Question:** *"Section 2.3 contains only the text "We have put this part to appendix 7.2, "*
>
> **Response:**
> The issue with Sec 2.3 was purely a formatting decision. In an earlier draft we moved the detailed LLM-as-optimizer background and memory mechanism discussion to Appendix 7.2 to ensure that all experimental results fit within the page constraints. Now we have restored the related work in Sec 2.3 in the main text, while keeping extended details in the appendix. The revised structure improves readability and restores the intended flow of Sec 2.
>
>
> **Reviewer Question:** *"No ablation on the hybrid selection strategy "*
> **Response:**
> We thank the reviewer for pointing this out. We have now added a dedicated ablation study of the hybrid selector in Section 6.3.3. The table below summarizes the results:
>
> | Selector | Top1 F | Top10 F | AUC-Top10 | Hypervolume | Uniqueness | Validity | Diversity |
> |----------|--------|---------|-----------|-------------|------------|----------|-----------|
> | Only Pareto front-based | 4.170 ± 0.055 | 4.092 ± 0.042 | 3.957 ± 0.037 | 0.857 ± 0.061 | 0.870 ± 0.025 | 0.909 ± 0.012 | 0.667 ± 0.049 |
> | Only Fitness-based | 4.163 ± 0.190 | 4.055 ± 0.094 | 3.938 ± 0.048 | 0.680 ± 0.274 | 0.844 ± 0.015 | 0.916 ± 0.005 | 0.517 ± 0.031 |
> | Hybrid selector | 4.336 ± 0.246 | 4.300 ± 0.164 | 4.116 ± 0.040 | 0.905 ± 0.200 | 0.872 ± 0.015 | 0.908 ± 0.019 | 0.494 ± 0.032 |
>
> The hybrid selector achieves the highest values on Top-1 F, Top-10 F, AUC-Top10, and hypervolume, demonstrating more stable optimization and better convergence across objectives. Fitness-only selection tends to over-exploit and shows unstable hypervolume, while Pareto-only selection maintains higher diversity but converges more slowly. Combining the two strategies avoids the weaknesses of each individual selector and consistently produces the best overall multi-objective optimization performance.
> **Revision:** The results and discussion is added in Sect 6.3.3
>
> ---
>
> **Reviewer Question:** *"beneficial effects of the hybrid selection strategy "*
>
> **Response:**
> Using both selection mechanisms together is necessary because each single selector removes valuable molecules for different structural reasons. A strict Pareto filter discards any molecule that is slightly dominated, even when its chemistry is very different from the dominating molecule and therefore still informative for search. This leads to the loss of structurally diverse candidates that share similar objective performance. Conversely, fitness-only selection keeps only the highest-scoring molecules and gradually collapses toward a narrow region of chemical space, removing many molecules that are non-dominated in the multi-objective sense but simply have slightly lower fitness. Such molecules are important for preserving alternative optimization pathways and preventing mode collapse. The hybrid selector retains globally competitive molecules while maintaining sufficient structural variety, leading to a more stable and well-balanced optimization trajectory.
>
> **Revision:** This discussion is added in Sect 6.3.3

---

> ### Author Response · Authors · 2025-11-28
> **Response to Reviewer Kohi (Part 3)**
>
> **Reviewer Question:** *" no quantitative analysis of memory consumption, prompt token counts over time"*
>
> **Response:**
> We thank the reviewer for pointing this out. We have now added a dedicated analysis of overall cost and experience consumption in Section 6.5. The table below summarizes the results:
> | Method                     | Total input tokens (M) | Total output tokens (M) | Input tokens for experience (M) | Output tokens for experience (M) | Total cost (USD) | Running time (h) |
> |----------------------------|-------------------------|--------------------------|----------------------------------|-----------------------------------|-------------------|-------------------|
> | MOLLEO (GPT-4o)            | 2.991 ± 0.246           | 2.191 ± 0.201            | N/A                              | N/A                               | 19.126 ± 1.698    | 6.029 ± 1.281     |
> | ExLLM (GPT-4o)             | 2.857 ± 0.338           | 0.204 ± 0.020            | 0.253 ± 0.031                    | 0.031 ± 0.005                     | 6.938 ± 0.796     | 0.393 ± 0.114     |
> | ExLLM (Gemini-2.5-Flash)   | 3.799 ± 0.456           | 0.279 ± 0.036            | 0.365 ± 0.029                    | 0.027 ± 0.004                     | 1.102 ± 0.136     | 0.495 ± 0.133     |
> | ExLLM (DeepSeek-V3.1)      | 4.145 ± 0.364           | 0.283 ± 0.024            | 0.372 ± 0.033                    | 0.037 ± 0.004                     | 2.816 ± 0.246     | 1.703 ± 0.341     |
> | ExLLM (Qwen3-Max)          | 4.316 ± 1.687           | 0.285 ± 0.110            | 0.408 ± 0.160                    | 0.051 ± 0.020                     | 4.604 ± 0.389     | 1.575 ± 0.796     |
>
> The quantitative analysis in Sec 6.5 shows that ExLLM has substantially low memory-related token usage and drastically reduced runtime compared to MOLLEO when using the same API model. In particular, ExLLM produces far fewer output tokens due to the compact evolving-experience mechanism, and the experience module accounts for only a small fraction of the total token consumption. As a result, ExLLM achieves over an order of magnitude speedup (more than 15× faster) while also reducing overall API cost. The comparison across different proprietary and open-source LLMs further confirms that ExLLM’s memory mechanism remains efficient and lightweight regardless of the underlying model.
> **Revision:** These results and discussion are added in Sect 6.5
>
> ---
>
> **Reviewer Question:** *"Would the proposed framework work with open-source general-purpose LLMs "*
>
> **Response:**
> Yes. We have added a new ablation in Sec 6.3.4 evaluating ExLLM with both proprietary LLMs (GPT-4o and Gemini-2.5-Flash) and open-source LLMs (DeepSeek-V3.1 and Qwen3-Max). The results show that ExLLM consistently maintains strong optimization performance across all models, demonstrating that the framework is not tied to a particular backbone and generalizes well to open-source LLMs.
> The full comparison is provided below:
> | LLM in ExLLM | Top1 F | Top10 F | AUC-Top10 | Hypervolume | Uniqueness | Validity | Diversity |
> |--------------|--------|---------|-----------|-------------|------------|----------|-----------|
> | GPT-4o | **4.336 ± 0.246** | **4.300 ± 0.164** | **4.116 ± 0.040** | **0.905 ± 0.200** | **0.872 ± 0.015** | 0.908 ± 0.019 | 0.494 ± 0.032 |
> | Gemini-2.5-Flash | 4.208 ± 0.011 | 4.191 ± 0.009 | 4.070 ± 0.026 | 0.750 ± 0.007 | 0.615 ± 0.035 | 0.950 ± 0.006 | **0.521 ± 0.013** |
> | DeepSeek-V3.1 | 4.175 ± 0.039 | 4.152 ± 0.040 | 3.994 ± 0.039 | 0.763 ± 0.100 | 0.544 ± 0.072 | 0.959 ± 0.009 | 0.496 ± 0.020 |
> | Qwen3-Max | 4.075 ± 0.123 | 4.056 ± 0.115 | 3.968 ± 0.085 | 0.587 ± 0.215 | 0.316 ± 0.041 | **0.974 ± 0.011** | 0.500 ± 0.045 |
>
> **Revision:** These results and discussion are added in Sect 6.3.4

---

> ### Author Response · Authors · 2025-11-28
> **Response to Reviewer Kohi (Part 4)**
>
> **Reviewer Question:** *" what situation would correspond to "worst-init" where ... "*
>
> **Response:**
> The motivation for the three initialization schemes is to study how sensitive different optimization methods are to the quality of the starting population under a fixed oracle budget. In practice, the initial set of molecules in a real molecular discovery campaign can vary widely depending on how much prior information is available, and evolutionary optimization methods are known to be sensitive to such variability.
> Best-init and random-init reflect the two most common real-world scenarios:
> - **Best-init** models a setting where researchers have prior predictors, literature candidates, or earlier high-quality hits to initialize the search.
> - **Random-init** corresponds to the common case in de novo tasks where no prior structural biases exist and the initial library is obtained by uniform sampling or scaffold enumeration.
> The **worst-init** setting is not intended to represent a realistic choice made by practitioners, but rather serves as a stress test that evaluates *robustness to poor initialization*. In realistic campaigns, poorly chosen starting points can arise unintentionally due to limitations of surrogate predictors, scaffold biases, or dataset-specific artifacts, especially when working with noisy or unreliable scores. Evolutionary pipelines can fail catastrophically in these cases, collapsing early and wasting the entire evaluation budget. Including worst-init therefore reveals whether a method can recover from unfavorable starting points and still navigate toward high-quality regions of chemical space.
>
> **Revision:** This detailed discussion are added in Sect 6.8
>
> ---
>
> **Reviewer Question:** *"How do the authors justify sacrificing molecular diversity for higher fitness in the context of real-world applications where diversity is often explicitly required? "*
>
> **Response:**
> We agree with the reviewer that molecular diversity is essential in real-world drug discovery. The lower diversity reported in Table 2 is a consequence of a **controlled exploitation shift** that occurs in the later stages of ExLLM. As the search approaches high-value regions in multi-objective space, the algorithm intentionally focuses on more fine-grained optimization, which naturally reduces dispersion in the top-100 set. Importantly, this reduction does **not** imply inadequate exploration capacity. Evidence from PMO strongly supports this: ExLLM achieves near-maximal performance on structure-sensitive tasks such as **deco_hop**, **scaffold_hop**, and **valsartan_smarts**, all of which explicitly require scaffold-level exploration and long-range structural edits. These results indicate that the framework is fully capable of exploring diverse chemical regions before converging.
> Furthermore, ExLLM maintains **competitive global coverage**, as reflected by consistently strong **hypervolume** across all initialization schemes in Table 2. Hypervolume captures the spread of the Pareto front rather than only intra-top-100 diversity, showing that ExLLM preserves the overall breadth of solutions despite exhibiting tighter clustering among the highest-fitness molecules.
> Finally, the exploration–exploitation balance in ExLLM is not fixed: adjusting the hybrid selector ratio or the k-offspring sampling provides direct control over diversity. We are extending the framework with explicit diversity-aware selectors as a promising direction for future work.

---

> ### Author Response · Authors · 2025-11-28
> **Response to Reviewer Kohi (Part 5)**
>
> **Reviewer Question:** *"Would the framework be unsuitable for scaffold-hopping or exploring novel chemical space? "*
>
> **Response:**
> No. The structure-based PMO tasks already *directly* test scaffold hopping, long-range edits, and navigation of novel chemical regions. ExLLM achieves **near–maximal** scores on all three such tasks (From Table 2 and Table 16 ):
> | Task type & Objective (↑) | REINVENT | Aug. Memory | Graph GA | GP BO | MOLLEO | Genetic GFN | **ExLLM (Ours)** | LICO | Chemma |
> |---------------------------|----------|-------------|----------|-------|--------|--------------|------------------|-------|--------|
> | **deco_hop**             | 0.666 | 0.688 | 0.619 | 0.629 | _0.942_ | _0.733_ | **0.956** | 0.619 | 0.831 |
> | **scaffold_hop**         | 0.560 | 0.565 | 0.517 | 0.548 | **0.971** | _0.615_ | _0.916_ | 0.547 | 0.669 |
> | **valsartan_smarts**     | 0.000 | 0.000 | 0.000 | 0.000 | **0.867** | _0.135_ | _0.831_ | 0.000 | 0.000 |
> | **Total (↑)**            | 14.036 | 15.356 | 13.823 | 13.182 | 17.862 | 16.213 | **19.165** | 14.708 | 17.534 |
> | **Rank (↓)**             | 7 | 5 | 8 | 9 | 2 | 4 | **1** | 6 | 3 |
>
> All three tasks require **explicit scaffold replacement**, **SMARTS-level constraints**, and **exploration away from parent scaffolds**. ExLLM not only solves them but reaches **best or second-best scores** across the board.
>
> Moreover, on PMO ExLLM ranks **#1 overall (19.165)** and converges on **deco_hop / scaffold_hop within <1000 evaluations** (Figure 9), demonstrating that the framework does *not* get stuck in local basins and can reliably explore novel chemical space.
> Thus, empirical results strongly support that ExLLM is suitable for scaffold-hopping and broad chemical-space exploration.
>
> ---
>
> **Reviewer Question:** *"The paper provides no mechanism to prevent or detect such forgetting."*
>
> **Response:**
> The evolving experience mechanism is specifically designed to avoid catastrophic forgetting. Although only one memo is maintained, each update incorporates information from *old experience* and *both* the top-performing molecules and uniformly sampled low-performing molecules drawn from the entire history up to generation $t$. This bidirectional sampling ensures that the memo repeatedly reintroduces diverse structural patterns, including those that appeared early but may become important later. Because the update summarizes the full evidence set rather than only the most recent generation, useful patterns cannot disappear simply due to local shifts in the search region. Empirically, this design produces higher hypervolume and AUC than both “no memory’’ and retrieval-style memories, indicating that the experience remains stable and informative across the entire optimization trajectory (Table 1 & Sec 4.2).

---

### Official Review · Reviewer_Yxrt · 2025-10-31

**Soundness:** 2
**Presentation:** 2
**Contribution:** 1
**Rating:** 2
**Confidence:** 4

**Summary:**

This paper closely follows the overall workflow of MOLLEO for multi-objective molecular optimization. The main contribution lies in introducing several practical techniques for updating the memory pool and reducing the computational cost associated with feedback processing.

**Strengths:**

The related work section is well-organized and provides a clear overview of prior research. The paper introduces several strategies to reduce the computational cost in LLM-based molecular optimization, particularly within the MOLLEO framework.

**Weaknesses:**

Overall, the paper does not appear to provide a method with substantial novelty. The proposed approach largely follows the MOLLEO process, and the modifications in the memory update and feedback stages seem a little incremental rather than fundamentally new. The contribution is closer to presenting practical tips for reducing LLM API calling costs and prompt engineering strategies, rather than introducing a novel algorithmic framework.

Regarding the presentation of results, the main tables could be improved for clarity. The current main table contains many blank entries, which hurts readability. I recommend condensing the main table by focusing on a subset of objectives (e.g., 3–6 or 4–6 objectives) and reporting full results in the Appendix. Since the paper also includes experiments with 1–6 objectives, it would be more informative to include strong baselines such as MOLLEO, DyMol, and Genetic-GFN for the higher-dimensional objective settings in the main comparison.

Additionally, the result tables would benefit from further refinement. The “Worst” and “Best initial” columns contain many empty cells, and it is unclear whether they need to be part of the main results. These analyses feel closer to ablation-level studies and may be more suitable for the Appendix rather than presented as primary results.

**Questions:**

See weakness

---

> ### Author Response · Authors · 2025-11-28
> **Response to Reviewer Yxrt (Part 1)**
>
> **Reviewer Question:** *“Particularly within the MOLLEO framework. The contribution is closer to presenting practical tips for reducing LLM API calling costs and prompt engineering strategies, rather than introducing a novel algorithmic framework.”*
>
> **Response:**  We thank the reviewer for the feedback and wish to clarify a central misunderstanding. **ExLLM does not follow the MOLLEO workflow**, nor is it a collection of practical tips, but on a powerful optimizer for large discrete space search with significant generalizability. Our method introduces **novel algorithmic components not present in MOLLEO or prior LLM-based molecular optimization pipelines**, and these components are essential for the performance and cross-domain generalization demonstrated in the paper. We also added a new method overview diagram in **Figure 1**, which also shows our contributions.
>
> ## 1. ExLLM is not based on MOLLEO
> LLM-as-optimizer has become a well-established direction after OPRO, LMEA, and related work. Our method is developed within this broader paradigm, but it is **not** derived from MOLLEO. We start from the challenges of large discrete optimization and design an end-to-end framework from scratch. ExLLM introduces two core algorithmic components: **the first experience memory designed specifically for large discrete search** and an **autoregressive k-offspring exploration mechanism**. These enable ExLLM to achieve new state-of-the-art performance in multi-objective molecular optimization, while reducing runtime and cost by an order of magnitude, and generalizing across domains with a unified complex feedback adaptor.
> In contrast, MOLLEO is a GA–LLM hybrid tailored to molecular optimization, **with no memory mechanism, no k-offspring strategy, and no cross-domain generalizability**.
> Our contributions are algorithmic rather than “practical tips.” Each component has a clear theoretical motivation and demonstrates measurable impact:
> ### (a) First experience memory for large discrete optimization (Sec. 3.1)
> While memory is important in LLM-as-optimizer frameworks, conventional retrieval-based memory causes **information and storage explosion**, and leads to **significant runtime and cost increases** (Table 1).
> We propose an evolving experience pool with a curriculum-style update scheme, enabling a **lightweight memory** to remain consistently effective. On PMO, removing the experience module yields 18.165 (already state-of-the-art); adding it improves performance to 19.165 (Sec. 4.2), with full ablations in Sec. 6.3.2.
> We further incorporate probabilistic memory injection to balance exploration and exploitation (Sec. 6.3.2).
> ### (b) Autoregressive k-offspring exploration (Sec. 3.2)
> We leverage the autoregressive factorization of LLMs to generate k diverse offspring per call, increasing exploratory breadth while reducing cost and runtime.
> Comprehensive ablations (Sec. 6.3.1) show consistent gains when k>1.
> Runtime and cost comparisons are added in Sec. 6.5 and summarized below:
>
> | Method            | ExLLM (GPT-4o) | MOLLEO (GPT-4o) | Graph GA      | Gp-BO         | Genetic-GFN   | MARS          | JT-VAE        | DyMol         | REINVENT      |
> |-------------------|----------------|------------------|----------------|----------------|----------------|----------------|----------------|----------------|----------------|
> | Running time (hours) | **0.393 ± 0.114** | 6.029 ± 1.281    | 1.041 ± 0.156 | 0.683 ± 0.100 | 0.453 ± 0.071 | 0.692 ± 0.129 | 3.082 ± 0.362 | **0.289 ± 0.037** | 0.522 ± 0.088 |
> ---
>
>
> | Method                   | Total input tokens | Total output tokens | Input tokens for experience | Output tokens for experience | Total cost (USD) | Running time (h) |
> |--------------------------|--------------------|----------------------|------------------------------|-------------------------------|-------------------|------------------|
> | MOLLEO (GPT-4o)          | 2.991 ± 0.246      | 2.191 ± 0.201        | N/A                          | N/A                         | 19.126 ± 1.698    | 6.029 ± 1.281    |
> | ExLLM (GPT-4o)           | 2.857 ± 0.338      | 0.204 ± 0.020        | 0.253 ± 0.031                | 0.031 ± 0.005                 | 6.938 ± 0.796     | 0.393 ± 0.114    |
> | ExLLM (Gemini-2.5-Flash) | 3.799 ± 0.456      | 0.279 ± 0.036        | 0.365 ± 0.029                | 0.027 ± 0.004                 | 1.102 ± 0.136     | 0.495 ± 0.133    |
> | ExLLM (DeepSeek-V3.1)    | 4.145 ± 0.364      | 0.283 ± 0.024        | 0.372 ± 0.033                | 0.037 ± 0.004                 | 2.816 ± 0.246     | 1.703 ± 0.341    |
> | ExLLM (Qwen3-Max)        | 4.316 ± 1.687      | 0.285 ± 0.110        | 0.408 ± 0.160                | 0.051 ± 0.020                 | 4.604 ± 0.389     | 1.575 ± 0.796    |

---

> ### Author Response · Authors · 2025-11-28
> **Response to Reviewer Yxrt (Part 2)**
>
> ### (c) Unified feedback adapter (Sec. 3.3)
> A structured adapter that normalizes multi-objective signals, incorporates constraint violations, and integrates expert textual hints. The feedback adapter enables plug-and-play multi-objective and constraint integration.  MOLLEO does not support such unified complex feedback handling.
>
> ---
>
> ## 2. Strong evidence of novelty and impact
> If the method were merely incremental, it would not achieve:
> - **New SOTA on PMO** (19.165 total score, +7.3% over prior best),
> - **1st place on 17/23 PMO tasks**,
> - **Robust cross-domain generalization** across physics (stellarator design), geometry (circle packing), engineering (offshore jackets), routing problems (MOCVRP), peptide design, and GPU kernel optimization. We have moved the results to Sec 4.3 in the main text instead of only being in appendix.
> These domains differ fundamentally from MOLLEO’s scope.
> ---
> ## 3. Summary
> ExLLM provides a **new LLM-as-optimizer framework for optimization problems with large discrete search space** with:
> - evolving experience memory specifically for optimization with large discrete search space,
> - autoregressive multi-offspring exploration,
> - unified multi-objective & constraint-aware feedback handling,
> - and demonstrated generalization across 7 diverse domains.
> Therefore, the method is **not derived from MOLLEO**, and the contributions go significantly beyond prompt engineering or cost reduction.
> We hope this resolves the misunderstanding and accurately reflects the novelty of our work.
>
> ---
>
> **Reviewer Question:** *"Regarding the presentation of results, the main tables contain blank entries..."*
>
> **Response:**
> Thank you for the detailed suggestions. We clarify below why the current table structure is intentional, and how it aligns with fair comparison.
> **(1) Blank entries are not missing results, but reflect the constraints of the baselines.**
> Several RL-based baselines (REINVENT, DyMol, Genetic-GFN) **do not expose a mechanism to fix the initial population** and therefore cannot be evaluated under the *Worst* and *Best* initialization schemes. Rather than omitting those settings entirely, which would make the comparison incomplete, we explicitly mark missing entries to maintain transparency. We have changed the blank entries into “N/A” (Not applicable) in the revised version to avoid confusion.
>
> **(2) Worst / Random / Best initializations are not ablations; they are part of the core fairness protocol.**
> Evolution-based methods are sensitive to initial populations under fixed budget, and prior work has not standardized how these are chosen. To ensure **fair and reproducible** comparison, we fix three populations (Worst, Random, Best) for *all methods*, following the PMO recommendation to control for initialization effects. These settings provide complementary difficulty regimes and reveal robustness beyond random initialization use cases. Because this directly affects benchmark validity, we believe these columns belong in the main table.
>
> **(3) On condensing the main table:**
> The **5-objective setting is the canonical benchmark** used in MOLLEO and the broader PMO literature, so the main table is fixed for 5 objectives.
>
> **(4) Inclusion of strong baselines in higher-dimensional settings:**
> MOLLEO, DyMol, and Genetic-GFN are already included in the main table. For the experiment of 1 to 6 objectives, we already put that in Sec 6.3.5, and also compared with MOLLEO.
>
> We hope our explaination fully addresses your concerns.

---

### Author Response · Authors · 2025-11-28
**Authors’ Summary of Revisions and Clarifications**

# **Authors’ Summary of Revisions and Clarifications**

We thank all reviewers for their detailed and constructive feedback. In the revised version uploaded we have carefully addressed every comment and significantly strengthened the paper both technically and experimentally. Below is a summary of major revisions and clarifications:

### **Generalization Results Relocated from Appendix to Main Text**:
A summary of results of transfering to other domains added in the main text (Sec 4.3) Involving the strong generalizability to across physics (stellarator design), geometry (circle packing), engineering (offshore jackets), routing problems (MOCVRP), peptide design, and GPU kernel optimization.

### **1. Clarified novelty and contributions**
We clarified that **ExLLM is not based on MOLLEO**, but introduces a new LLM-as-optimizer framework for large discrete optimization. Our original Sec. 3 to detail the methodological innovations, including:
- a **compact evolving experience memory** designed specifically for long-horizon discrete optimization,
- an **autoregressive k-offspring exploration strategy**,
- a **unified feedback adapter** integrating multi-objective, constraint, and expert textual hints.

### **2. Added extensive new ablation studies**
We added multiple ablations requested by reviewers:
- **Hybrid selector ablation** (Sec. 6.3.3)
- **k-offspring ablation across 4 LLMs** (GPT-4o, Gemini-2.5-Flash, DeepSeek-V3.1, Qwen3-Max) (Sec. 6.3.1)
- **Open-source LLM evaluation** (Sec. 6.3.4)
- **Per-objective improvement table**, Pareto-front plots, and parallel coordinate plots (Sec. 6.6)

All ablations consistently demonstrate that each component of ExLLM provides meaningful contributions.

### **3. Added strong new LLM-as-optimizer baselines**
Following the reviewers’ suggestions, we included two recent SOTA LLM molecular optimization systems: **LICO (2025)** and **Chemma (2024)**.  (Sec 6.4)
ExLLM achieves the best performance on both PMO and PMO-1K benchmarks.

### **4. Added detailed cost, memory, and runtime analysis**
We added a full quantitative comparison of token usage, memory-related costs, and runtime in Sec. 6.5.
Results show that ExLLM provides **>15× speedup**, significantly fewer output tokens, and lower overall cost than MOLLEO when using the same API model.

### **5. Improved clarity of methodology and overall structure**
- Added a **new Figure 1** with an end-to-end overview of ExLLM, demonstrating the contributions.
- Restored and expanded **Sec. 2.3** to the main text.
- Added task templates for multiple domains (Sec. 6.7).
- Added prompt templates for other domains, showing the simplicity of using our system.

### **6. Improved result tables and fixed editorial issues**
- Replaced blank entries with **N/A** to reflect baseline constraints.
- Added oracle-call curves in Sec. 6.4.
- Corrected all editorial and formatting issues.

### **7. Added deeper analyses for diversity, initialization, and forgetting**
- Clarified the **fitness–diversity trade-off** and why ExLLM clusters more tightly at high-value regions.
- Explained the purpose of **Worst/Random/Best initialization** as evaluating robustness (Sec. 6.8).
- Discussed how the evolving experience mechanism avoids **catastrophic forgetting** by repeatedly incorporating historical information.

---

*In conclusion the revision includes new experiments, new analyses, clearer methodological descriptions, improved figures and tables,  expanded comparisons with state-of-the-art methods, and fixes to all editorial issues. We hope this revision addresses all the concerns.*

---

### Author Response · Authors · 2025-12-03
**For AC: summary of the paper, reviews and our responses**

Dear AC,

Due to the updated rebuttal procedure, we would like to offer a concise summary of our work and our responses to the reviewers’ comments. We hope this overview will assist in your final decision.
## 1. Summary of Our Work

-  **What we did (two-sentence summary):**
We propose **ExLLM**, a new LLM-as-optimizer framework specifically designed for *large discrete search spaces*, featuring an evolving experience mechanism, an autoregressive k-offspring strategy, and a unified feedback adapter. With only a task description and an evaluation function, ExLLM achieves **state-of-the-art results across seven heterogeneous domains**, including molecular design, geometry, physics, engineering, routing, peptides, and GPU kernels.

-  **Motivation & Novelty:**
Existing LLM-as-optimizer systems, such as those used in dialogue coding solvers or retrieval-based prompt memories, are **not designed for optimization with large discrete search space**, where common retrieval-based memories produces severe *information redundancy, memory growth, prompt explosion, significant runtime, and exploration collapse* (Table 1). In addition, current LLM-as-optimizer systems also tend to be limited to a specified domain such as reasoning, molecular design and do not generalize across heterogeneous tasks. Although AlphaEvolve generalizes across several domains, it is mainly applied to tasks where an explicit code solver can be implemented, which defines the limits of its current problem scope. Finally, exploration in LLM-as-optimizer frameworks is under-studied especially for problems with large discrete search space.

**ExLLM is the first framework to explicitly target these unmet needs**, and introduces mechanisms tailored to discrete search.

- **Our Contributions**
1. **First Evolving Experience Mechanism for Large Discrete Optimization**
   - A compact, curriculum-style, non-redundant experience snippet that prevents memory explosion and maintains exploration, explicitly addressing failure modes of retrieval memories in large discrete search. Furthermore, the over-conditioning of experience is explored, and a hyperparameter probability of experience injection is introduce to mitigate this effect while taking advantage of experience.

2. **Systematic Exploration of the Autoregressive k-Offspring Mechanism**
   - A principled analysis of how autoregressive sampling can expand search breadth, reduce API calls, improve convergence stability, and consistently outperform k=1 across *four different LLMs*.

3. **Unified Multi-objective / Constraint / Expert Feedback Adapter**
   - A general adapter that normalizes objectives, encodes constraints and expert hints, and enables **plug-and-play cross-domain transfer**.
   - Demonstrated SOTA results on **seven fundamentally different optimization problems**, all with the *same* hyperparameters.
   - Anonymous GitHub repository includes full reproducibility scripts and tutorials for transferring ExLLM to any new task. Link: [https://anonymous.4open.science/r/ExLLM-AD78/README.md](https://anonymous.4open.science/r/ExLLM-AD78/README.md)

---

## 2. Summary of Reviewers’ Recognized Strengths

Across reviewers, the following strengths were consistently highlighted:

- **Clear organization of related work(Yxrt)**

- **The reviewer highlights that many molecular generation works omit or superficially treat multi-objective optimization, and appreciates that our paper tackles it rigorously.(Kohi)**

- **Training-free & computationally lightweight approach( Kohi)**
  The reviewer considers the ability to avoid task-specific training a notable advantage, reducing computational burden relative to prior LLM optimization approaches.

- **k-offspring strategy reduces API cost (Kohi)**

- **Strong empirical performance on PMO and additional tasks (including peptides)(ds92)**
  The reviewer finds our results strong, especially the novel six-objective experiments and peptide optimization.

- **Mostly clear writing and presentation of experiments(ds92)**

- **Compelling framework leveraging pretrained LLMs without re-engineering(CwLi)**
  The reviewer considers ExLLM an effective and practical framework for using large pretrained models directly as optimizers.

- **Demonstrated improvement in both single-objective and multi-objective tasks(CwLi)**

- **Novel exploration mechanism and experience design recognized as valuable(CwLi)**
  The reviewer sees the contributions (k-offspring + evolving experience) as meaningful innovations that improve exploration efficiency.

These strengths reflect that ExLLM offers both **practical impact** and **algorithmic contributions**, with clear writing and presentations. For more details, you are more than welcome to check our revised paper submission with changes in blue font.

---

> ### Author Response · Authors · 2025-12-03
> **Summary of Our Responses to Reviewer Comments**
>
> #### **1. Clarifying the Main Contribution and Correcting the Core Misunderstanding**
>
> A central point that required clarification was how our key contributions were understood by some reviewers. We clarified emphatically that:
>
> - **Our contribution is *not* applying LLM-as-optimizer to molecular design**, nor focusing on reducing cost.
> - Instead, we highlight that **LLM-as-optimizer is an emerging research direction**, and our work **does not follow MOLLEO**, but **introduces a completely new optimizer designed from scratch for large discrete search spaces**.
>
> Our framework includes several **novel algorithmic components**:
>
> 1. **Experience Pool for Large Discrete Search Spaces**
>    - The first experience-learning mechanism designed specifically for problems with large discrete search space.
>    - Addresses the *information redundancy and memory explosion* caused by retrieval-based systems (**Table 1**).
>    - We also study the limitations of *overusing* experience in such problems.
>    - Our experience pool is **entirely newly designed**, **not adapted** from any prior implementation. The phrase *"inspired by"* referred only to the *conceptual idea of summarizing experience*, not to code or mechanism reuse. We have now clarified this misunderstanding explicitly in the paper (**middle of Sec. 3.1**).
>
> 2. **k-Offspring Mechanism and Its Cross-Model Trade-offs**
>    - Although simple, we conducted an in-depth investigation of its exploration–exploitation trade-offs and its gains across different k values and LLMs.
>    - We evaluated it across multiple LLMs and obtained **consistent, general conclusions** that are informative for future LLM-as-optimizer work especially for large discrete searching.
>
> 3. **Hybrid Selector and Unified Complex Feedback Handling**
>    - Enables the optimizer to handle complex feedback signals, including muti-objectives, textual feedback, and constraints, allowing plug-and-play transfer to new domains at minimal cost (requiring only a task description and a scorer) while maintaining** SOTA** performance.
>    - We provide fully open-sourced code for reproduction.
>
> **Source of the misunderstanding.**
> We think that the lack of an end-to-end overview diagram(*ds92*) and placing all cross-domain transfer results in the appendix may have contributed to confusion.
> We have now:
> - added a new **main method figure (Figure 1)**
> - added a summary of the cross-domain generalization results into **Sec. 4.3** in main text.
>
> ---
>
> #### **2. Clarifying Baseline Choices and Adding New Baselines for PMO benchmark**
>
> We clarified which baselines were **not appropriate** for our setting due to incompatible assumptions. We also **added additional LLM-as-optimizer baselines** (suggested by *Kohi*, *CwLi*), making comparisons more comprehensive.
>
> With these additions, **ExLLM still outperforms all baselines substantially** on final performance and efficiency (e.g., **PMO-1K, Sec. 6.4**).
>
> ---
>
> #### **3. Additional Ablation Experiments Added**
>
> Beyond our already extensive ablations, we added several new studies requested by reviewers.
> Each addition is listed below with its corresponding section and reviewer(s):
> - **Hybrid selector ablation**
>   *Sec. 6.3.3 — for Kohi, ds92, CwLi*
>
> - **k-offspring trade-off ablation across more different LLMs**
>   *Sec. 6.3.1 — for ds92, Kohi*
>
> - **Additional open-source LLMs on 5 objective optimization**
>   *Sec. 6.3.4 — for ds92, Kohi*
>
> - **Detailed experience and overall cost, running time analysis 6.5**
>   *Sec. 6.5 — for Yxrt,Kohi,ds92*
>
>
> These ablations directly answer reviewer questions about mechanism necessity, component contribution, and stability across domains.
>
> ---
>
>
> #### **4. Additional Clarifications Across the Paper**
>
> We added several important conceptual and methodological clarifications:
>
> - **Fitness–diversity trade-off explanation**
>   *Sec. 6.4 — for Kohi, CwLi*
>
> - **Expanded description of the three initialization schemes**
>   *Sec. 3.3 — for Kohi*
>
> - **Clarifying how the experience pool avoids catastrophic forgetting**
>   *Sec. 3.1 — for CwLi*
>
> ---
>
> #### **5. Editorial Fixes and Added Figures**
>
> We corrected all editorial errors reported by reviewers (*CwLi, ds92*) and updated the manuscript with a new **main overview Figure 1** (*ds92*).
>
> ---
>
> **We have provided clear clarifications and responses to every question raised by the reviewers. For each requested ablation, we have either pointed to the existing experiments in the paper or added new ones accordingly. We have also addressed all concerns regarding presentation clarity, as reflected in our revised submission. We hope that the strengthened quality, solid results, and the substantial effort invested in the revisions adequately address all remaining concerns.**

---

### Meta-Review · Area_Chair_q4ds · 2026-01-07

**Summary:**

The paper uses LLMs with an evolutionary search loop for optimization in large discrete search spaces with a primary focus on molecular design. The paper introduces prompting based mechanisms: a summarizer pool that summarizes historical high- and low-performing candidates, an offspring sampling strategy and a feedback adapter to handle constraints and multiple objectives.

The primary consensus among multiple reviewers (Yxrt, Kohi, ds92) is that the paper comes across as incremental prompt engineering improvement for a known idea. Reviewer Yxrt characterized the work as practical tips for reducing LLM API costs. The key idea relies mostly on standard prompt engineering techniques (summarization and few-shot prompting) applied to an evolutionary search loop. The terminology ("autoregressive exploration operator", "controlled exploitation shift") and its usage is a bit too hyperbolic for a scientific paper. I also didn't appreciate the use of LLM driven figures in the main paper. Therefore, I recommend rejecting the paper.

**Reviewer Concerns:**

The authors put effort in the rebuttal but it came across again written in a marketing/hyperbolic manner.

**Reviewer Scores:**

I do not think the reviewers would have changed their score.

---

### Decision · Program_Chairs · 2026-01-26

Reject